# Tightening the Score Matching Gap for Diffusion Models

**Benjamin Dupuis** [* 1]  **Tyler Farghly** [* 1]  **Maxime Haddouche** [1]  **Alain Durmus** [† 2]  **Umut Simsekli** [† 1]

## Abstract

Diffusion models (DMs) are a state-of-the-art generative method to approximately sample from an unknown distribution. Their training and evaluation primarily rely on an Evidence Lower Bound (ELBO), which relates the Kullback-Leibler (KL) divergence of model samples to the score matching loss along the path, which serves as a tractable surrogate. The difference between sample quality and the score matching loss produced by this bound leads to the *score matching gap*, which is known to be tight in the worst-case but not descriptive of sample quality in general. In this work, we provide a theoretical analysis of this gap, developing tighter bounds for three metrics: KL divergence, reverse KL divergence, and Wasserstein distance, effectively exploiting the regularity of the class of score estimators. Our results suggest that the quality of the score approximation has more impact on closing the score matching gap for low noise scales. To obtain these bounds, our key technical insight is to exploit the contraction properties of the backward processes. In particular, we rely on entropy flows, logarithmic Sobolev inequalities and reflection couplings, rigorously linking the ergodicity of the Langevin diffusion to the score matching gap problem.

## 1. Introduction

Diffusion, or score-based generative models (DMs, SGMs) have shown remarkable performances in recent years (Song et al., 2021b; Karras et al., 2022; Esser et al., 2024), with applications ranging from computer vision (Ho et al., 2020; Dhariwal & Nichol, 2021) and medicine (Kazerouni et al., 2023) to natural language processing (Yang et al., 2024).

[1]INRIA - CNRS - Département d'Informatique de l'Ecole Normale Supérieure - PSL Research University, France [2]École Polytechnique - CMAP, IP Paris, Palaiseau, France. Correspondence to: Benjamin Dupuis <benjamin.dupuis@inria.fr>, Tyler Farghly <tyler.farghly@inria.fr>.

*Proceedings of the $43^{rd}$ International Conference on Machine Learning*, Seoul, South Korea. PMLR 306, 2026. Copyright 2026 by the author(s).

DMs aim to estimate a data distribution $\mu$, with access only to a finite set of samples. They do this by constructing a stochastic process $(\overrightarrow{X}_t)_{t \in [0,T]}$ (the forward process), being the solution of a stochastic differential equation (SDE) over the time interval $[0, T]$ and initialized from the data distribution $\mu$. Two classical instantiations of $(\overrightarrow{X}_t)_{t \in [0,T]}$ are either the $d$-dimensional Brownian motion or the Ornstein–Uhlenbeck process (Uhlenbeck & Ornstein, 1930). In our work, we adopt the latter, which admits the standard Gaussian, denoted by $\gamma^d$, as stationary distribution. This construction defines a path measure connecting $\mu$ to $\gamma^d$, and thus an ideal generative model can be formed by taking a large value of $T$ and considering the time-reversed (or backward) process associated with $(\overrightarrow{X}_t)_{t \in [0,T]}$, defined for any $t \in [0, T]$ as $\overleftarrow{X}_t := \overrightarrow{X}_{T-t}$. It can be shown that this too is a diffusion process, whose drift depends on the Stein scores $(t, x) \mapsto s(t, x)$ of the forward marginals (Haussmann & Pardoux, 1986; Millet et al., 1989) which are unknown in practice. This score can be estimated using a family of neural networks $s^\theta : (t, x) \mapsto s^\theta(t, x)$ parameterized by $\theta \in \Theta$ (Song et al., 2021a) which can then be used to approximate the backward process with $(\overleftarrow{X}_t^\theta)_{t \in [0,T]}$ using the network $s^\theta$ in place of $s$. In practice, the backward process is further approximated by initializing from $\gamma^d$ and applying a numerical scheme to discretize the corresponding SDE. Formally, this is often framed as minimizing the Kullback-Leibler (KL) divergence between the data and model distributions.

**Learning the score.** Training and evaluating directly using the KL divergence is intractable as we do not have direct access to the density of the model distribution. To circumvent this, we typically rely on a tractable upper bound, given by,

$$\mathrm{KL}\left(\mu | \overleftarrow{\mu}_T^\theta\right) \leqslant \frac{1}{4} \int_0^T \mathbb{E}[\varepsilon_t^\theta(\overrightarrow{X}_t)^2] \mathrm{d}t + \mathrm{KL}\left(\overrightarrow{\mu}_T | \gamma^d\right), \quad (1)$$

with $\varepsilon_t^\theta(x)^2 := \|s(t, x) - s^\theta(t, x)\|^2$ and $\overrightarrow{\mu}_T := \mathrm{Law}(\overrightarrow{X}_T)$. The first term on the right-hand side is the *score matching loss*, defined as a weighted $\mathrm{L}^2$ loss between the true and approximate score functions (Song et al., 2021a). The second term is usually intractable, but does not depend on the learned score $s^\theta$ and decays exponentially fast as $T$ grows (Bakry et al., 2014). This loss function is further approximated using the denoising score matching loss, efficiently optimized using stochastic gradient methods (Vin-

cent, 2011; Song & Ermon, 2019). The theoretical justification for Equation (1) is provided by Song et al. (2021a), using a data processing inequality that bounds the KL between the marginals, $\mu$ and $\overleftarrow{\mu}_T^\theta$, by the KL between their path measures, which is further expressed via Girsanov's theorem (Øksendal, 2003). Complementing this perspective, Kingma et al. (2021); Huang et al. (2021); Kingma & Gao (2023); Vahdat et al. (2021); Luo (2022) show that this bound can be understood as a variational evidence lower bound (ELBO) on the log-likelihood, grounding diffusion models within the variational inference framework.

**The score-matching gap.** Given that the score matching loss is an upper bound, with no guarantee of closely approximating the KL divergence, it might seem surprising that this has so effectively served as a theoretical foundation of the diffusion model framework. The use of a path measure data processing inequality produces an unpredictable looseness in the bound which we refer to as the *score matching gap*. This observation motivates our fundamental question:

*Is there any way to make the score matching gap tighter?*

The existence of this gap has non-trivial consequences. In particular, it renders the score matching objective an unreliable proxy for model evaluation; a lower score matching loss does not necessarily imply a better approximation. Consequently, evaluation typically utilizes sample-based metrics, which require many function evaluations and fail to disentangle the errors arising from score approximation with those arising from the discretization of the reverse process (Song et al., 2021b; Karras et al., 2022).

Even without being formally defined, the score matching gap has been implicitly involved in the convergence bounds literature (Bortoli, 2022), motivated by the study of the strong generalization ability of DMs (Bonnaire et al., 2025). Such results consist in upper-bounding a discrepancy[1] $\mathrm{D}\left(\mu, \overleftarrow{\mu}_T^\theta\right)$ by the score matching loss and other terms accounting for the initialization and discretization error. A common practice in this literature is to relate the error to a discretized version of $\int_{[0,T]} \varepsilon_t^\theta(\overrightarrow{X}_t)^2 \mathrm{d}t$ or assume an uniform bound of $\varepsilon_t^\theta(\overrightarrow{X}_t)^2$ over $[0,T]$, meaning implicitly that the score matching loss is the right objective to evaluate the method. Thus, the potential looseness of the score matching gap is discarded from the convergence analysis. However, it allows to retrieve discretized versions of (1) for various choices of D, including the KL divergence case (Chen et al., 2023a; Benton et al., 2024a; Strasman et al., 2025; Conforti et al., 2025) as an improvement of previous results for the Total Variation (TV) distance (Lee et al., 2022a; Chen et al., 2023b). To address certain limitations of KL and TV, Wasserstein convergence bounds have also

---

[1]D typically denotes a divergence or a Wasserstein distance.

been considered (Beyler & Bach, 2025), often involving a uniform bound on the score matching loss (Lee et al., 2022b; Gentiloni-Silveri & Ocello, 2025; Bruno et al., 2025; Gao et al., 2025).

**Contributions.** In this work, we provide a comprehensive analysis of the score matching gap with a focus on tightening score matching loss bounds like Equation (1). We begin Section 3 with a result suggesting that for $T$ large, the score matching gap is tight in the worst case: it can only be tightened given additional information. In Section 3.1, we tighten (1) using a pseudo Lipschitz assumption on the score network $s^\theta$, along with optional dissipativity and obtain,

$$\mathrm{KL}\left(\mu | \overleftarrow{\mu}_T^\theta\right) \lesssim \int_0^T \lambda_T(t)\mathbb{E}[\varepsilon_t^\theta(\overrightarrow{X}_t)^2]\mathrm{d}t + \mathrm{K}_T \ , \quad (2)$$

with $\mathrm{K}_T := C_T \mathrm{KL}\left(\overrightarrow{\mu}_T | \gamma^d\right)$ for some $C_T < 1$. The time weighting $\lambda_T(t) < 1$ is decreasing in $t$ (see Table 1 for explicit rates), prioritising the score matching loss near convergence. This also tightens, as a byproduct, the classical convergence bounds of (Conforti et al., 2025; Benton et al., 2024a) in the time-continuous setting.

Inspired by the convergence analysis literature, we investigate the score matching gap for alternative topologies. In Section 4.1, we derive bounds of type (1) for the reversed KL divergence $\mathrm{KL}\left(\overleftarrow{\mu}_T^\theta | \mu\right)$ with a time-decaying weighting that depends only on the concentration properties of the data distribution $\mu$. We also extend our theory to Wasserstein distances in Section 4.2, where we show that under dissipativity and smoothness conditions, it holds that,

$$\mathscr{W}_1(\mu, \overleftarrow{\mu}_T^\theta) \lesssim \int_0^T C_T(t)\mathbb{E}[\varepsilon_t^\theta(\overrightarrow{X}_t)]\mathrm{d}t + \mathscr{W}_1(\overrightarrow{\mu}_T, \gamma^d) \ ,$$

where, again, $C_T(t)$ is a decay function detailed in Table 1.

Finally, Section 5 presents some experimental results on a low-dimensional setting showing that our time-decaying weightings provide better estimates of sample quality that are more useful for model evaluation.

**Technical innovations.** Our core technical innovation is to exploit the contraction properties of the backward processes, connecting the score-matching gap to ergodicity of the underlying diffusion process. For our KL bounds, we exploit an 'entropy flow' analysis (Section 3.1) alongside logarithmic Sobolev inequalities (Gross, 1975). These ideas appeared partially in the convergence bounds literature, focusing on time-discrete settings. In particular, (Lee et al., 2022a;b) used "$\chi^2$-flows" to reach TV and Wasserstein bounds, but only considered reversed $\chi^2$ flows, without exhibiting an explicit decay like $\lambda_T(t)$. For our Wasserstein bounds, we use a reflection coupling technique to tighten the time-dependence of the bounds. While using this technique is new in the context of DMs, some Wasserstein convergence

| Divergence | Assumptions | Decay - $\lambda_T(t)$ or $C_T(t)$ |
|---|---|---|
| Forward KL | $M$-P.L., $M < 1$ | $\sqrt{\frac{e^{2(1-M)(T-t)}-M}{e^{2(1-M)T}-M}}$ |
| Forward KL | $M$-P.L., $M > 1$ | $\sqrt{\frac{M-e^{-2(M-1)(T-t)}}{M-e^{-2(M-1)T}}}$ |
| Forward KL | $(1+\frac{c}{t})$-P.L. | $\sqrt{1 - \frac{2t^{2c+1}}{(2c+1)T^{2c}+2T^{2c+1}}}$ |
| Forward KL | P.L., D. | $\exp\left(-t/C_{\mu,\theta,d}\right)$ |
| Reverse KL | $\rho_0$-LSI | $\sqrt{\frac{\rho_0}{\rho_0+e^{2t}-1}}$ |
| Wass. $\mathscr{W}_2$ | $\rho_0$-LSI | $\sqrt{\frac{\rho_0^3}{\rho_0+e^{2t}-1}}$ |
| Wass. $\mathscr{W}_f, \mathscr{W}_1$ | P.L., D. | $\exp\left\{-\int_0^t c_s \mathrm{d}s\right\}$ |

*Table 1.* $M$-P.L. ($s^\theta$ is $M$-one-sided Lipschitz as in Theorem 3.7), $\rho_0$-LSI ($\mu$ satisfies the log-Sobolev inequality with constant $\rho_0$), D. (Dissipativity). We refer to the corresponding statements for the constants appearing in the dissipative cases.

bounds tightening the classical $\mathcal{O}(T)$ factor also recently emerged (Wang & Wang, 2025; Bruno & Sabanis, 2025), as well as time-uniform results (Beyler & Bach, 2025).

All omitted proofs can be found in the appendix.

**Notation.** We denote $a \wedge b := \min(a, b)$. The set of Borel probability measures on $\mathbb{R}^d$ is written $\mathcal{P}(\mathbb{R}^d)$. Given $\rho, \pi \in \mathcal{P}(\mathbb{R}^d)$, their Kullback-Leibler (KL) divergence is $\mathrm{KL}(\rho|\pi) := \int \log(\mathrm{d}\rho/\mathrm{d}\pi)\mathrm{d}\rho$ and the relative Fisher information is $\mathscr{I}(\rho|\pi) := \int \|\nabla \log(\mathrm{d}\rho/\mathrm{d}\pi)\|^2 \mathrm{d}\rho$. We use $\lesssim$ for inequality up to an absolute constant. Given $I \subset \mathbb{R}^d$, we note by $\mathrm{C}^{1,2}(I \times \mathbb{R}^d)$ the functions $f(t,x)$ that are $\mathrm{C}^1$ in $t$ and $\mathrm{C}^2$ in $x$. Given $\mu, \nu \in \mathcal{P}(\mathbb{R}^d)$, $\Pi(\mu, \nu) \subset \mathcal{P}(\mathbb{R}^d \times \mathbb{R}^d)$ is the set of couplings between $\mu$ and $\nu$. For a distance $d$ on $\mathbb{R}^d$, the Wasserstein distance is $\mathscr{W}_d(\mu, \nu) := \inf_{\pi \in \Pi(\mu,\nu)} \int d(x,y)\mathrm{d}\pi(x,y)$. For $p \geqslant 1$, $\mathscr{W}_p$ denotes $\mathscr{W}_d$ with the $p$-norm $d(x,y) = \|x-y\|_p$.

## 2. Setup and Technical Background

### 2.1. Background on Diffusion Models

In this paper, we consider the forward process to be an Ornstein-Uhlenbeck process on $\mathbb{R}^d$, *i.e.*,

$$\mathrm{d}\overrightarrow{X}_t = -\overrightarrow{X}_t \mathrm{d}t + \sqrt{2}\mathrm{dB}_t , \quad \overrightarrow{X}_0 \sim \mu , \quad (3)$$

where $\mu$ is the data distribution. This process admits the standard Gaussian $\gamma^d := \mathrm{N}(0, \mathrm{I}_d)$ as invariant distribution. We denote $\overrightarrow{\mu}_t := \mathrm{Law}(\overrightarrow{X}_t)$, $\overrightarrow{p}_t$ its Lebesgue density, for $t > 0$, and by $\tilde{p}_t := \overrightarrow{p}_t/\gamma^d$ the renormalized density.

Let $T > 0$ and define the time-reversal process $\overleftarrow{X}_t := \overrightarrow{X}_{T-t}$, which under mild conditions is shown to be a weak solution of the SDE (Anderson, 1982; Föllmer, 1985),

$$\mathrm{d}\overleftarrow{X}_t = \left(-\overleftarrow{X}_t + 2\nabla \log \tilde{p}_{T-t}(\overleftarrow{X}_t)\right)\mathrm{d}t + \sqrt{2}\mathrm{d}\overline{\mathrm{B}}_t , \quad (4)$$

with $\overleftarrow{X}_0 \sim \overrightarrow{\mu}_T$ and $(\overline{\mathrm{B}}_t)_{t \geqslant 0}$ a standard Brownian mo-

tion. Similarly, we denote $\overleftarrow{\mu}_t := \mathrm{Law}(\overleftarrow{X}_t)$ and by $\overleftarrow{p}_t$ its Lebesgue density for $t > 0$. The score $s(t, x) := 2\nabla \log \tilde{p}_t(x)$ is usually estimated through a parametric family of *score networks* $\{s^\theta : \mathbb{R}_+ \times \mathbb{R}^d \to \mathbb{R}^d, \theta \in \Theta\}$, where $\Theta$ is a given hypothesis class. Given such a parameter $\theta \in \Theta$, the backward process (4) is approximated by the following SDE, starting from $\overleftarrow{X}_0 \sim \gamma^d$ (instead of the unknown $\overrightarrow{\mu}_T$) and defined by

$$\mathrm{d}\overleftarrow{X}_t^\theta = \{-\overleftarrow{X}_t^\theta + s^\theta(T-t, \overleftarrow{X}_t^\theta)\}\mathrm{d}t + \sqrt{2}\mathrm{dB}_t . \quad (5)$$

Popular discretization schemes to simulate Equation (5) include the Euler-Maruyama and exponential Euler integrator schemes (Durmus & Moulines, 2014). We denote $\overleftarrow{\mu}_t^\theta := \mathrm{Law}(\overleftarrow{X}_t^\theta)$ and $\overleftarrow{p}_t^\theta$ its Lebesgue density. Throughout, we assume that $s^\theta$ is locally bounded measurable.

In practice, $\mu$ is unknown and the score has to be estimated from i.i.d.samples $(Z_1, \ldots, Z_n) \sim \mu^{\otimes n}$. It has been shown by Vincent (2011); Hyvärinen (2005) that $\theta$ can be estimated by minimizing the *denoising score matching (DSM) loss*,

$$\frac{1}{n}\sum_{i=1}^n \int \mathbb{E}[\|s^\theta(t, \overrightarrow{X}_t^{Z_i}) - 2\nabla \log \tilde{p}_{t|0}(\overrightarrow{X}_t^{Z_i}|Z_i)\|^2]\mathrm{d}\varpi(t) ,$$

where $\varpi$ is a probability distribution on $[0, T]$, $\overrightarrow{X}_t^x$ corresponds to (3) initialized at $\overrightarrow{X}_0^x := x$, and $\tilde{p}_{t|0}(\cdot|x)$ is the conditional density of $\overrightarrow{X}_t$ given $\overrightarrow{X}_0 = x$. This is motivated by the identity $\nabla \log \tilde{p}_t(\overrightarrow{X}_t) = \mathbb{E}[\nabla \log \tilde{p}_{t|0}(\overrightarrow{X}_t|\overrightarrow{X}_0)]$, coming from the Fisher's identity (Efron, 2011). Modern practices often involve a time-weighted version of this loss (Ho et al., 2020; Dhariwal & Nichol, 2021).

### 2.2. Logarithmic Sobolev Inequalities

Let $\Phi(x) := x\log(x)$, with $\Phi(0) := 0$. Let $\pi \in \mathcal{P}(\mathbb{R}^d)$, the entropy functional associated to $\pi$ is defined as $\mathrm{Ent}_\pi(f) := \mathbb{E}[\Phi(f(X))] - \Phi(\mathbb{E}[f(X)])$, for $X \sim \pi$ and $f \in \mathrm{L}^1(\pi)$. Some of our contributions are based on the logarithmic Sobolev inequalities (LSI) associated with Equations (4) and (5) (Bakry et al., 2014; Chafaï & Lehec, 2017).

**Definition 2.1.** $\nu \in \mathcal{P}(\mathbb{R}^d)$ satisfies the LSI with constant $\rho$ (denoted $\rho$-LSI) if for all differentiable positive $f \in \mathrm{L}^1(\nu)$,

$$\mathrm{Ent}_\nu(f) \leqslant \frac{\rho}{2}\int \frac{\|\nabla f\|^2}{f}\mathrm{d}\nu .$$

Let $\mu, \nu \in \mathcal{P}(\mathbb{R}^d)$ such that $\mu \ll \nu$, the LSI for $\mu$ can be rewritten as $2\mathrm{KL}(\nu|\mu) \leqslant \rho\mathscr{I}(\nu|\mu)$. The typical example is the standard Gaussian $\gamma^d$, which satisfies the LSI with constant 1 (Gross, 1975). LSIs classically appear in the ergodic theory of diffusion processes (Bakry et al., 2014). In particular, the fact that the invariant distribution of a

given diffusion process (such as $(\overrightarrow{X}_t)_{t \geqslant 0}$) satisfies an LSI is equivalent to the exponential convergence of the entropy along the associated semigroup. In our case, by reversibility of the invariant measure $\gamma^d$, this translates to

$$\mathrm{KL}\left(\overrightarrow{\mu}_t | \gamma^d\right) \leqslant e^{-2t} \mathrm{KL}\left(\mu | \gamma^d\right) . \qquad (6)$$

In our paper, we exploit these convergence properties of the forward and backward processes to obtain sharper bounds.

## 3. Theoretical Study of Score Matching Gaps

To motivate our work, we first investigate the question: *can the score matching bound be tightened in general?* In the proposition below, we provide a negative result.

**Proposition 3.1.** *Suppose that* $\mathrm{KL}\left(\mu | \nu\right), \mathrm{KL}\left(\nu | \gamma^d\right) < \infty$, $T \geqslant \log(d) \vee 1$ *and* $\mathrm{supp}(\mu) \subseteq B_R(\mathbf{0})$ *for some* $R < \infty$, *there exists a score* $s^\theta$ *such that* $\overleftarrow{\mu}_T^\theta = \nu$, *and*

$$\mathrm{KL}(\mu | \overleftarrow{\mu}_T^\theta) \geqslant \frac{1}{4} \int_0^T \mathbb{E}[\varepsilon_t^\theta(\overrightarrow{X}_t)^2] \mathrm{d}t + \mathrm{KL}(\overrightarrow{\mu}_T | \gamma^d)$$
$$- 2e^{\frac{1+R^2}{2} - 2T}(1 + \mathrm{KL}\left(\nu | \gamma^d\right)) .$$

Here, $\nu$ represents any given distribution produced by a diffusion model and $\mu$ is the target. The proposition shows that by increasing $T$, the score matching gap can be made arbitrarily tight with the right choice of score. The construction is based on the solution to a dynamic Schrödinger bridge problem (Léonard, 2013) and can be seen as a non-typical, irregular score function. This suggests that, in general, one cannot tighten the score matching gap without specifying particular score estimators. This motivates our direction of improving the score matching bound through the use of additional regularity assumptions. To this end, we first describe the entropy flow method for DMs.

### 3.1. Adapting the Entropy Flow Technique to DMs

Equation (6) is based on the celebrated De-Bruijn identity (Stam, 1959), $\frac{\mathrm{d}}{\mathrm{d}t}\mathrm{KL}\left(\overrightarrow{\mu}_t | \gamma^d\right) = -\mathscr{I}\left(\overrightarrow{\mu}_t | \gamma^d\right)$ and follows directly from the LSI for $\gamma^d$ and Grönwall's lemma. Such "entropy flow" computations have also been used to bound the distance between the marginal distributions of Langevin processes Bogachev et al. (2016), and in various subfields of machine learning (Mou et al., 2018; Chourasia et al., 2021; Borovykh et al., 2023; Dupuis & Simsekli, 2024; Dupuis et al., 2026). The interest of this technique in our context is to provide natural upper bounds on the forward and reverse KL divergences between $\mu$ and $\overleftarrow{\mu}_T^\theta$.

When $\mu$ has a Lebesgue density, the map $(t, x) \mapsto \overleftarrow{p}_t(x)$ is in $\mathrm{C}^{1,2}([0, T] \times \mathbb{R}^d)$ and the spatial derivatives up to order 2 are bounded (see Section B.2).

*Remark* 3.2. It is known that, under mild regularity assumptions, Equation (5) has an almost-surely continuous

solution on $[0, T]$ for all $\theta \in \Theta$, and that the probability density function $\overleftarrow{p}_t^\theta$ of $\overleftarrow{X}_t^\theta$ is a solution of the Fokker-Planck equation associated with Equation (5) (Umarov et al., 2018; Bogachev et al., 2015). In this work, as it is classically done in the literature, we implicitly assume that $(t, x) \mapsto \overleftarrow{p}_t^\theta(x)$ belongs to $\mathrm{C}^{1,2}([0, T] \times \mathbb{R}^d)$, and it is a solution of the Fokker-Planck equation

Background on Fokker-Planck equations is given in Section A.1. Let us recall the notation,

$$\varepsilon_t^\theta(x)^2 := \|2\nabla \log \tilde{p}_t(x) - s^\theta(t, x)\|^2 , \quad x \in \mathbb{R}^d . \quad (7)$$

When stated, we make the following assumption, ensuring that several terms appearing in our theory are finite.

**Assumption 3.3.** For any $t \in (0, T)$, $\mathscr{I}\left(\overleftarrow{\mu}_t | \overleftarrow{\mu}_t^\theta\right) < +\infty$, and the map $t \mapsto \mathbb{E}[\varepsilon_t^\theta(\overleftarrow{X}_t)^2 + (1 + \|\overleftarrow{X}_t\|)^{-1} s^\theta(T-t, \overleftarrow{X}_t)]$ is finite and integrable on $(0, T)$.

Note that the first part of the assumption only holds for $t \in (0, T)$, in particular, it does *not* require that $\mathscr{I}\left(\mu | \gamma^d\right)$ or that $\mu$ is absolutely continuous *w.r.t.* $\gamma^d$. The second part of the assumption is mild. In particular, square-integrability of $\varepsilon_t^\theta(\overleftarrow{X}_t)$ is necessary for the score matching framework to be well-posed and the last part is typically implied by Hölder continuity of $s^\theta$ when $\mu$ has compact support.

The following proposition is an upper bound on the relative entropy between the true and estimated backward processes.

**Lemma 3.4.** *Suppose that Assumption 3.3 holds. For any map* $\alpha : \mathbb{R}_+ \to [0, 1)$ *and for all* $t \in (0, T)$, *we have*

$$\frac{\mathrm{d}}{\mathrm{d}t}\mathrm{KL}\left(\overleftarrow{\mu}_t | \overleftarrow{\mu}_t^\theta\right) \leqslant -\alpha(t)\mathscr{I}\left(\overleftarrow{\mu}_t | \overleftarrow{\mu}_t^\theta\right) + \frac{\mathbb{E}[\varepsilon_{T-t}^\theta(\overleftarrow{X}_t)^2]}{4(1 - \alpha(t))} .$$

This result is an adaptation of the entropy flow method for DMs and follows from the Fokker-Planck equations satisfied by the backward processes (Section A.1). It is a natural generalization of the De-Bruijn identity for DMs. Whenever $\overleftarrow{\mu}_t^\theta$ satisfies a LSI, *i.e.*, the Fisher information can be replaced by a KL divergence, and Lemma 3.4 leads to a contraction, improving the bound.

In Sections 3.2 and 4.1 we will apply respectively Lemma 3.4 to tighten the score matching gap via time-decaying weightings (see Table 1, Figures 1 and 2).

### 3.2. Tightening the bound under LSI

The next theorem is a generic bound on the forward KL $\mathrm{KL}\left(\mu | \overleftarrow{\mu}_T^\theta\right)$ expressed in terms of the (possibly infinite) time-dependent LSI constant of $\overleftarrow{\mu}_t^\theta$.

**Theorem 3.5.** *Under the conditions of Lemma 3.4, assume that there exists a map* $\rho : [0, T] \to \mathbb{R}_+^\star \cup \{+\infty\}$ *such that*

$1/\rho$ is continuous and $\overleftarrow{\mu}_t^\theta$ satisfies the $\rho(t)$-LSI. Then, for any continuous map $\alpha : \mathbb{R}_+ \to [0,1)$, we have

$$\mathrm{KL}\left(\mu|\overleftarrow{\mu}_T^\theta\right) \leqslant \mathrm{K}_T + \int_0^T \frac{e^{-\int_0^t \frac{2\alpha(T-u)}{\rho(T-u)}\mathrm{d}u}}{4(1-\alpha(t))}\mathbb{E}[\varepsilon_t^\theta(\overrightarrow{X}_t)^2]\mathrm{d}t \,,$$

with the constant $\mathrm{K}_T$ given by

$$\mathrm{K}_T := \exp\left\{-\int_0^T \frac{2\alpha(u)}{\rho(u)}\mathrm{d}u\right\}\mathrm{KL}\left(\overrightarrow{\mu}_T|\gamma^d\right) \,.$$

*Remark* 3.6. By the logarithmic Sobolev inequality satisfied by $\gamma^d$, it is known that $\mathrm{KL}\left(\overrightarrow{\mu}_T|\gamma^d\right) \leqslant e^{-2T}\mathrm{KL}\left(\mu|\gamma^d\right)$, as soon as $\mu \ll \gamma^d$. Even for singular measures $\mu$, we can show with classical arguments that

$$\mathrm{KL}\left(\overrightarrow{\mu}_T|\gamma^d\right) \leqslant \frac{1}{2\left(e^{2T}-1\right)}\mathscr{W}_2(\mu,\gamma^d)^2 \,,$$

as soon as $\mu$ has finite moments of order 2.

This theorem improves over Equation (1) obtained by Song et al. (2021a) using Girsanov's theorem, which it recovers in the case of infinite LSI constants by choosing $\alpha \equiv 0$ (roughly matching the analysis of Lyu (2009)). However, as we always have $\rho(0) = 1$ (as $\overleftarrow{\mu}_0^\theta = \gamma^d$), we can *systematically* improve the time-dependence of Equation (1) as soon as $\rho$ is continuous.

We note that to benefit from an exponential decay Theorem 3.5, we need to choose $\alpha$ such that $\alpha(t) \in (0,1)$ at least on a subset of $[0,T]$ with non-zero Lebesgue measure. While this leads to improvements in the time-dependence of the bound, it might also lead to worsening the absolute constant $1/4$ in front of the integral. In the following results, we make the choice $\alpha \equiv 1/2$ to simplify the exposition. As we will see, this leads to non-trivial exponential decay in the bound, improving the time dependence at the cost of potentially worsening the absolute constants in the bound.

Now, our main challenge is to estimate the log-Sobolev constant of $\overleftarrow{\mu}_t^\theta$, for all $t \in [0,T]$. We note that Equation (5) can be seen as a perturbation of the Ornstein-Uhlenbeck process (3), with the perturbation drift $s^\theta(t,\cdot)$. This observation allows us to exploit the rich literature on perturbative analysis of LSIs (Malrieu, 2001; Monmarché et al., 2024) for Langevin processes. In the next theorem, we obtain score matching bounds based on a one-sided Lipschitz condition.

**Theorem 3.7.** *In the same setting as Lemma 3.4, suppose that there exists $M_t \in \mathbb{R}$ s.t. for all $t \in [0,T]$, $x,y \in \mathbb{R}^d$, $\langle s^\theta(t,x) - s^\theta(t,y), x-y \rangle \leqslant M_t \|x-y\|^2$ (one-sided Lipschitz condition). If $M_t \equiv M$ is constant, we have*

$$\mathrm{KL}\left(\mu|\overleftarrow{\mu}_T^\theta\right) \leqslant \mathrm{K}_T + \int_0^T \frac{\lambda_T(t)}{2}\mathbb{E}[\varepsilon_t^\theta(\overrightarrow{X}_t)^2]\mathrm{d}t \,,$$

with $\mathrm{K}_T := \lambda_T(T)\mathrm{KL}\left(\overrightarrow{\mu}_T|\gamma^d\right)$, and, for $t \in [0,T]$,

$$\begin{cases} \lambda_T(t) := \sqrt{\frac{M-e^{-2(M-1)(T-t)}}{M-e^{-2(M-1)T}}} \,, & \text{if } M > 1 \,, \\ \lambda_T(t) := \sqrt{\frac{1+2(T-t)}{1+2T}} \,, & \text{if } M = 1 \,, \\ \lambda_T(t) := \sqrt{\frac{e^{2(1-M)(T-t)}-M}{e^{2(1-M)T}-M}} \leqslant e^{-\frac{1-M}{2-M}t} \,, & \text{if } M < 1 \,. \end{cases}$$

*If we have $M_t = 1 + c/t$, with $c > 1$ a constant, then,*

$$\lambda_T(t) := \sqrt{1 - \frac{2t^{2c+1}}{(2c+1)T^{2c} + 2T^{2c+1}}} \,.$$

We refer to Remark 3.6 for further upper bounds on the constant $\mathrm{K}_T$ in Theorem 3.7. Interestingly, our theory links time-weightings improvements in score-matching bounds with regularity conditions on $s^\theta$. We consider two particular cases: a one-sided Lipschitz constant uniformly bounded by $M$ over $[0,T]$ and a weaker assumption of a one-sided Lipschitz constant $M_t$ that can diverge in $\mathcal{O}(1/t)$ as $t \to 0^+$. In particular, if $M < 1$, we obtain a time-uniform bound of order $\mathcal{O}(\sup_t \mathbb{E}[\varepsilon_t^\theta(\overrightarrow{X}_t)^2])$. Moreover, note that the initialization error term, $e^{-2T}\lambda_T(T)\mathrm{KL}\left(\mu|\gamma^d\right)$, is also improved over the classical analysis. In the idealized continuous-time setting, this result improves over the classical KL convergence analysis (Conforti et al., 2025; Benton et al., 2024b). Additionally, our proof technique can accommodate any formula for $t \mapsto M_t$ (e.g. $M_t = \mathcal{O}(1/t^p), p > 0$), even though it may lead to intricate expressions (see Section B.4).

**Time decay analysis.** The map $\lambda_T(t)$ is represented on Figure 1 for various parameter values and are always decreasing and $\lambda_T(0) = 1$. Qualitatively, this shows that the score approximation has more impact on the score matching gap for small forward times, *i.e.*, low noise levels. This is to be compared with the standard scheduling practices when training DMs (Dhariwal & Nichol, 2021; Kingma & Gao, 2023). Indeed, as noted by Ho et al. (2020), the standard $\epsilon$-prediction objective implies to downweight the small noise scales. With our notation, a weighted DSM loss,

$$\frac{1}{n}\sum_{i=1}^n \int_0^T \omega_t \mathbb{E}[\|s^\theta(t,\overrightarrow{X}_t^{Z_i}) - 2\nabla\log\tilde{p}_{t|0}(\overrightarrow{X}_t^{Z_i}|Z_i)\|^2]\mathrm{d}t$$

is minimized with a finite dataset $(Z_1,\ldots,Z_n) \sim \mu^{\otimes n}$, and $t \mapsto \omega_t$ is increasing, *i.e.*, it is smaller for small times scales. A byproduct of this procedure might be to reduce the overfitting at small noise scales and lead to a better score approximation at these small scales, as overfitting leads to generating only images close to the training dataset. While this procedure is initially motivated by reducing the variance of the objective (Ho et al., 2020), we argue that our theory may serve as a partial explanation of the good generalization error observed in practice, as it suggests that the small noise scales matter more for generalization than the large

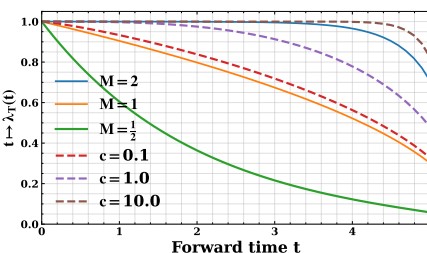

*Figure 1.* Decay term $\lambda_T(t)$ of Theorem 3.7 for $M_t \equiv M$ (full line) and for $M_t = 1 + c/t$ (dotted line). Here $T = 5$.

ones. We empirically investigate this in Section 5. Most importantly, our bounds decouple the time-weighting that should be used for training (Ho et al., 2020) and evaluation. *Remark* 3.8 (Connection to generalization bounds). Recently, Farghly et al. (2026); Dupuis et al. (2025) uncovered links between the denoising score matching loss and statistical learning. Our results are compatible with their approach, and we improve their results in the time-continuous settings in Section D, as an additional contribution.

In Theorem 3.7, we require a small pseudo-Lipschitz constant ($M < 1$) to obtain an exponential decay $\eta_T$ and, hence, a time-uniform bound. This is due to exponentially growing estimates of the LSI constant of $\overleftarrow{\mu}_t^\theta$ when $M > 1$. As shown by Monmarché et al. (2024), it is possible to improve these estimates by adding a dissipativity assumption, leading to the following condition.

**Assumption 3.9** (Dissipativity at distance). We say that a map $b : \mathbb{R}^d \to \mathbb{R}^d$ satisfies $(L, \rho, R)$-dissipativity at distance when, for all $t \geqslant 0$ and all $x, y \in \mathbb{R}^d$, we have

$$\begin{cases} \langle b(x) - b(y), x - y \rangle \leqslant L \|x - y\|^2 , \\ \langle b(x) - b(y), x - y \rangle \leqslant -\rho \|x - y\|^2 , \ \|x\| \geqslant R . \end{cases}$$

which we exploit in the following theorem.

**Theorem 3.10.** *Let $b_t^\theta(x) := s^\theta(t, x) - x$ and $b_t(x) := 2\nabla \log \tilde{p}_t(x) - x$. Assume that the conditions of Lemma 3.4 hold and that there exist, $\rho, L, R, K > 0$ such that $b_t^\theta$ satisfies $(L, \rho, R)$-dissipativity at distance for all $t \geqslant 0$. Also assume that $\langle x - s^\theta(t, x), x \rangle \leqslant A$ for all $\|x\| \leqslant R_\star := R(2 + 2L/\rho)^{1/d}$, and $2(2L + \rho)(L + \frac{\rho}{4})R_\star^2 + A \leqslant \rho d$. Then, there exists a constant $C_{\mu,d,\theta} > 0$ such that*

$$\mathrm{KL}\left(\mu | \overleftarrow{\mu}_T^\theta\right) \leqslant \frac{1}{2} \int_0^T e^{-\frac{t}{C_{\mu,\theta,d}}} \mathbb{E}[\varepsilon_t^\theta(\overrightarrow{X}_t)^2] \mathrm{d}t + \mathrm{K}_T' ,$$

*with* $\mathrm{K}_T' := e^{-T/C_{\mu,\theta,d}} \mathrm{KL}\left(\overrightarrow{\mu}_T | \gamma^d\right).$

Theorem 3.10 improves the classical ELBO of (1) and tightens the score matching gap. It requires the drifts of the backward equation to be one-sided Lipschitz (as in Theorem B.6) and Assumption 3.9 adds a dissipativity-like

condition, which we call "one-point dissipativity". Similar dissipativity conditions classically appear in the literature (Raginsky et al., 2017; Monmarché et al., 2024).

## 4. Extending the Score Matching Gap to Alternative Topologies

We now investigate extensions of the score matching gap for the reverse KL and Wasserstein topologies. While losing the link with variational inference, this approach circumvents limitations of Section 3.2, in particular, the reverse KL results of Section 4.1 avoid regularity assumptions on $s^\theta$. Alternatively, and inspired by the convergence analysis literature, Wasserstein distances studied in Section 4.2 avoid the absolute continuity constraints of KL-based metrics.

### 4.1. Reverse KL Bounds

Despite the fact that the KL divergence is not symmetric (van Erven & Harremoës, 2014), the entropy flow toolbox is flexible enough to obtain a flow for the reverse divergence $\mathrm{KL}\left(\overleftarrow{\mu}_t^\theta | \overleftarrow{\mu}_t\right)$. To state this result, we need an assumption, which is the reverse counterpart of Assumption 3.3.

**Assumption 4.1.** *For any $t \in (0, T)$, $\mathscr{I}\left(\overleftarrow{\mu}_t^\theta | \overleftarrow{\mu}_t\right) < +\infty$, and the map $t \mapsto \mathbb{E}[\varepsilon_t^\theta(\overleftarrow{X}_t^\theta)^2 + (1 + \|\overleftarrow{X}_t^\theta\|)^{-1} s(T - t, \overleftarrow{X}_t^\theta)]$ is finite and integrable on $[0, T)$.*

We then have the following reverse KL bound.

**Lemma 4.2.** *Suppose that Assumption 4.1 holds. We have, for all $t \in (0, T)$,*

$$\frac{\mathrm{d}}{\mathrm{d}t} \mathrm{KL}\left(\overleftarrow{\mu}_t^\theta | \overleftarrow{\mu}_t\right) \leqslant -\frac{1}{2} \mathscr{I}\left(\overleftarrow{\mu}_t^\theta | \overleftarrow{\mu}_t\right) + \frac{1}{2} \mathbb{E}[\varepsilon_{T-t}^\theta(\overleftarrow{X}_t^\theta)^2] .$$

Similarly, Lee et al. (2022a;b) analyzed the time-derivative of the reverse $\chi^2$-divergence $\chi^2\left(\overleftarrow{\mu}_t^\theta | \overleftarrow{\mu}_t\right)$ to get TV and Wasserstein convergence bounds. Lemma 4.2 may be seen as a generalization of their technique to forward and reverse KL divergences. Beyond yielding a stronger result, our approach provides explicit decay rates (see Theorem 4.3), which are hard to obtain in the $\chi^2$ case.

The results of Section 3.2 involve estimating log-Sobolev properties of $\overleftarrow{\mu}_T^\theta$, which requires regularity assumptions on $s^\theta$. Interestingly, we can avoid such an assumption by considering the reverse KL divergence $\mathrm{KL}\left(\overleftarrow{\mu}_T^\theta | \mu\right)$ which already appeared in the message passing literature (Minka, 2005) or in relation to Rényi variational inference (Li & Turner, 2016). To do so, we involve instead a log-Sobolev property for $\mu$, as detailed below.

**Theorem 4.3.** *Let $\rho_0 > 0$. Suppose that the conditions of Lemma 4.2 hold and that $\mu$ satisfies the $\rho_0$-LSI. Then,*

$$\mathrm{KL}\left(\overleftarrow{\mu}_T^\theta | \mu\right) \leqslant \bar{\mathrm{K}}_T + \int_0^T \frac{\bar{\lambda}(t)}{2} \mathbb{E}[\varepsilon_t^\theta(\overleftarrow{X}_{T-t}^\theta)^2] \mathrm{d}t ,$$

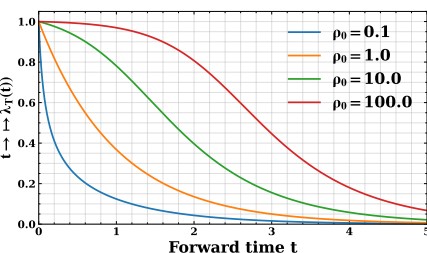

*Figure 2.* Value of the decay term $t \mapsto \lambda_T(t)$ appearing in Theorem 4.3 for different values of $\rho_0$.

with $\bar{\mathrm{K}}_T := \bar{\lambda}(T)\mathrm{KL}\left(\gamma^d | \overrightarrow{\mu}_T\right)$, and, for $t \in [0, T]$,

$$\bar{\lambda}(t) := \sqrt{\frac{\rho_0}{\rho_0 + e^{2t} - 1}} \leqslant e^{-t/\max(1, \rho_0)} . \quad (8)$$

*Remark* 4.4. Similar to Remark 3.6 in the forward KL case, it follows from the joint convexity of the KL divergence that

$$\mathrm{KL}\left(\gamma^d | \overrightarrow{\mu}_T\right) \leqslant \frac{1}{2\left(e^{2T} - 1\right)}\mathscr{W}_2(\gamma^d, \mu)^2 ,$$

when $\mu$ has finite moments of order 2. The data processing inequality also implies that $\mathrm{KL}\left(\gamma^d | \overrightarrow{\mu}_T\right) \leqslant \mathrm{KL}\left(\gamma^d | \mu\right)$ .

Theorem 4.3 contains an explicit decay $\bar{\lambda}$ (see Figure 2) independent of $s^\theta$. Equation (8) shows that this decay is at least exponential, making the bound time-uniform, *i.e.*, of order $\sup_t \mathbb{E}[\varepsilon_t^\theta(\overleftarrow{X}_{T-t}^\theta)]$. Such LSI assumptions have already been considered in the literature (Lee et al., 2022a;b). It allows us to estimate the LSI constant of $\overleftarrow{\mu}_t$ (see Section B.6).

*Remark* 4.5. The assumption that $\mu$ satisfies a LSI in Theorem 4.3 can be relaxed. Indeed, we only use it to estimate the log-Sobolev constant of $\overrightarrow{\mu}_t$, which (up to rescaling) is a convolution of $\mu$ with a Gaussian distribution. Recent results have estimated the LSI constant of such convolution-type measures under mild assumption on $\mu$, such as compact support (Zimmermann, 2013; Bardet et al., 2018) or sub-gaussian behavior (Wang & Wang, 2016) (at the cost of potentially dimension-dependent estimates). See also (Cattiaux & Guillin, 2022) for recent results on this matter.

As mentioned earlier, such a modified score matching gap cannot be linked to a classical ELBO (Song et al., 2021a), as minimizing the reverse KL divergence is not equivalent to maximizing the likelihood of the model. Despite this, we argue that the reverse KL divergence $\mathrm{KL}\left(\overleftarrow{\mu}_T^\theta | \mu\right)$ also translates a facet of the problem by providing new insights on the evaluation of DMs: concentration properties of the data distribution can be exploited in the absence of exploitable regularity of the score network and we observe again that small noise scales are key to evaluate overfitting in DMs. Theorem 4.3 can also be seen as a reverse KL convergence bound for continuous-time DMs, which is new to the best

of our knowledge. Interestingly, the term $\mathbb{E}[\varepsilon_t^\theta(\overleftarrow{X}_{T-t}^\theta)^2]$ already appears in the literature on Wasserstein convergence bounds for DMs, but with a worse dependence on $T$ (Silveri & Ocello, 2025; Bruno & Sabanis, 2025).

Note that Theorem 4.3 involves the score approximation *w.r.t.* $\overleftarrow{X}_t^\theta$, which cannot be estimated from samples. We take a first step towards a more classical score approximation term involving $X_t$ in Corollary 4.6 via Donsker-Varadhan's change of measure (Donsker & Varadhan, 1983).

**Corollary 4.6.** *In the setting of Theorem 4.3,*

$$\mathrm{KL}\left(\overleftarrow{\mu}_T^\theta | \mu\right) \leqslant \tilde{\mathrm{K}}_T + \int_0^T \frac{\tilde{\lambda}_T(t)}{2\rho_t} \log \mathbb{E}\left[e^{\rho_t \varepsilon_t^\theta(\overrightarrow{X}_t)^2}\right] \mathrm{d}t ,$$

*with $\rho_t = e^{-2t}\rho_0 + 1 - e^{-2t}$, $\tilde{\mathrm{K}}_T = \tilde{\lambda}_T(T)\mathrm{KL}\left(\gamma^d | \mu\right)$ and*

$$\tilde{\lambda}_T(t)^4 := \frac{\rho_0}{\rho_0 + e^{2t} - 1} .$$

### 4.2. Wasserstein Bounds

**Wasserstein bound via concentration of measure.** A corollary of Theorem 4.3 is a new bound in the $\mathscr{W}_2$ distance. Indeed, it has been shown by Otto & Villani (2000); Bobkov et al. (2001) that the LSI with constant $\rho_0$ implies the Talagrand inequality $\rho_0 \mathscr{W}_2(\mu, \nu) \leqslant \mathrm{KL}\left(\nu | \mu\right)$, for all $\nu \in \mathcal{P}(\mathbb{R}^d)$. Combined with Theorem 4.3, this proves that

$$\frac{\mathscr{W}_2(\mu, \overleftarrow{\mu}_T^\theta)^2}{\rho_0} \leqslant \bar{\mathrm{K}}_T + \int_0^T \frac{\bar{\lambda}(t)}{2}\mathbb{E}[\varepsilon_t^\theta(\overleftarrow{X}_{T-t}^\theta)^2]\mathrm{d}t . \quad (9)$$

This is of order $\mathcal{O}((\rho_0^2 \sup_t \mathbb{E}[\varepsilon_t^\theta(\overleftarrow{X}_{T-t}^\theta)^2])^{1/2})$, thus time-uniform. However, this term still cannot be estimated from samples. In the rest of this section, we alleviate this issue by exploiting the reflection coupling technique.

**Wasserstein Bounds via reflection couplings.** Analogously to the KL case, we exploit Wasserstein contraction properties. Historically, exponential contraction in $\mathscr{W}_p$ ($p \in [1, +\infty)$) were obtained for some Langevin SDEs (Bolley et al., 2012; Von Renesse & Sturm, 2005) via synchronous couplings. Major progress was achieved by Eberle (2016), using *reflection couplings* (Lindvall & Rogers, 1986) to obtain Wasserstein contraction rates under more generic assumptions (see Section A.3). Inspired by these works, we rely on a time-dependent dissipativity at distance condition as defined in Assumption 3.9. Let us introduce the notation

$$\kappa_b(r) := \inf_{\|x - y\| = r} \frac{\langle x - y, b(y) - b(x) \rangle}{\|x - y\|^2} .$$

We observe that the $(L, K, R/2)$-dissipativity at distance condition of Assumption 3.9 implies that

$$\kappa_b(r) \geqslant \begin{cases} -L, & r \leqslant R, \\ K, & r \geqslant R . \end{cases}$$

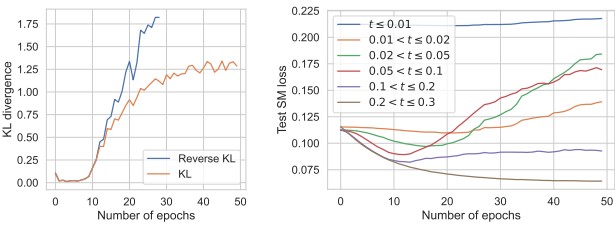

*Figure 3.* KL divergence and test score matching loss over training, with the SM loss split by forward-time interval. Earlier (low-noise) intervals track the KL divergence trajectory most closely.

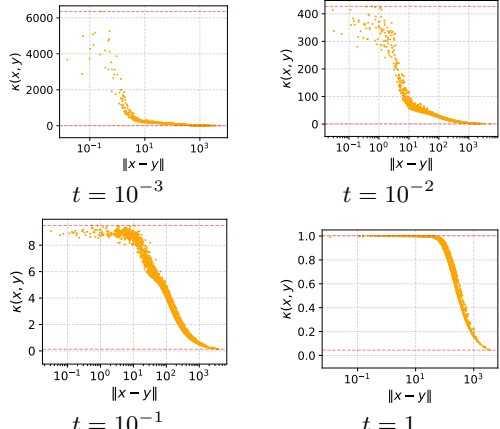

*Figure 4.* Empirical check of the smoothness and dissipativity assumptions for a U-Net trained on CIFAR-10: $\kappa(x, y)$ (see (10)) plotted against $\|x - y\|$ at different noise levels.

This is instrumental in our proofs, where it is used to control the distance between the two coupled SDEs (see Section C).

To obtain Wasserstein contraction, a key insight from (Eberle, 2016) is to analyze the quantity $f(r_t)$, where $r_t$ is the distance between the coupled processes and $f : \mathbb{R}_+ \to \mathbb{R}_+$ is a function constructed in the following lemma.

**Lemma 4.7** ((Eberle, 2016)). *Let $b : \mathbb{R}^d \to \mathbb{R}^d$ satisfying $(L, K, R/2)$-dissipativity at distance. Then there exists an increasing concave function $f := f_{L,R,K} : \mathbb{R}_+ \to \mathbb{R}_+$ and $c := c_{L,R,K} > 0$ such that $4f''(r) - r\kappa_b(r)f'(r) \leqslant -c$, for all $r > 0$. Moreover, we can take*

$$\frac{2}{c} \leqslant \int_0^R (r \wedge \frac{\sqrt{2\pi}}{\sqrt{L}})e^{\frac{Lr^2}{8}}\mathrm{d}r + \frac{4}{K} + \sqrt{\frac{8}{K}}Re^{\frac{LR^2}{8}} \ .$$

Additional details are gathered in Section A.3. Eberle (2016) showed that contraction in the distance $\mathscr{W}_f$ induced by the distance $(x, y) \mapsto f(\|x - y\|)$ requires the existence of $f$ as given by Lemma 4.7. For this reason, we focus in our main result on $\mathscr{W}_f(\mu, \overleftarrow{\mu}_T^\theta)$. Note however that $\mathscr{W}_f$ distance is equivalent to $\mathscr{W}_1$. Indeed, we prove in Section C that $\mathscr{W}_1 \leqslant \mathscr{W}_f \leqslant 2e^{\frac{LR^2}{8}}\mathscr{W}_1$, with $f$ given by Lemma 4.7,

Let us recall the notations $b_t(x) := s(t, x) - x$ and $b_t^\theta(x) := s^\theta(t, x) - x$, with $s(t, x) := 2\nabla \log \tilde{p}_t(x)$. Since we wish to derive contractions for two SDEs with different drifts, we must make a technical but important modification to the method of Eberle (2016). In the present case, borrowing the methodology of (Durmus et al., 2020), we couple the processes using a reflection coupling at distance and with a synchronous coupling when close. All details are rigorously stated in Section C. This leads to the following theorem.

**Theorem 4.8.** *Suppose that there is a $C^1$ non-increasing function, $L_t : [0, T] \to \mathbb{R}_+$, such that $b_t^\theta$ satisfies $(L, K, R_t/2)$-dissipativity at distance, and $t \mapsto \mathbb{E}[\varepsilon_t^\theta(\overrightarrow{X}_t)]$ is integrable on $[0, T]$. Let $f := f_{R,L_0,K}$, we have,*

$$\mathscr{W}_f(\mu, \overleftarrow{\mu}_T^\theta) \leqslant e^{-T}C_T\mathscr{W}_1(\mu, \gamma^d) + \int_0^T C_T(t)\mathbb{E}[\varepsilon_t^\theta(\overrightarrow{X}_t)]\mathrm{d}t,$$

*with $c_t := c_{R,L_t,K}$ and*

$$C_T(t) := \exp\left\{-\int_0^t c_s \mathrm{d}s\right\} \ .$$

The main advantage of this result compared to Equation (9) is that it directly involves the forward process $\overrightarrow{X}_t$, which can be estimated from samples as in the forward case. Similar to Sections 3.2 and 4.1, we observe that the $C_T$ factor comes from the regularity assumption on $s^\theta$. As above, we observe that $C_T$ is a decreasing function of $t$, showing that the score approximation is more critical to the performance for small noise scales.

Using the Lemma A.7 in appendix, we deduce the following corollary, which is a bound in the distance $\mathscr{W}_1$.

**Corollary 4.9.** *Under the same conditions as Theorem 4.8,*

$$\frac{\mathscr{W}_1(\mu, \overleftarrow{\mu}_T^\theta)}{2} \leqslant e^{\frac{L_0 R^2}{8}}\left(\mathrm{W}_T + \int_0^T C_T(t)\mathbb{E}[\varepsilon_t^\theta(\overrightarrow{X}_t)]\mathrm{d}t\right) \ .$$

This bound might also be interpreted as a convergence bound, in the idealized setting of a continuous-time backward process. Then, Corollary 4.9 has improved dependence on $T$ compared to the recent results of (Gentiloni-Silveri & Ocello, 2025; Bruno & Sabanis, 2025; Wang & Wang, 2025). From this convergence analysis perspective, the closest work from ours is (Beyler & Bach, 2025), which, despite using a slightly different setting, can obtain time-uniform bounds when transferred to our setup. Their setup, proof technique, and assumptions differ from our work and may be seen as a complementary approach.

## 5. Empirical Study

We investigate the validity of our theoretical results and their potential practical utility on toy experiments and with a DDPM model on CIFAR-10. See Section E for further details regarding experiments.

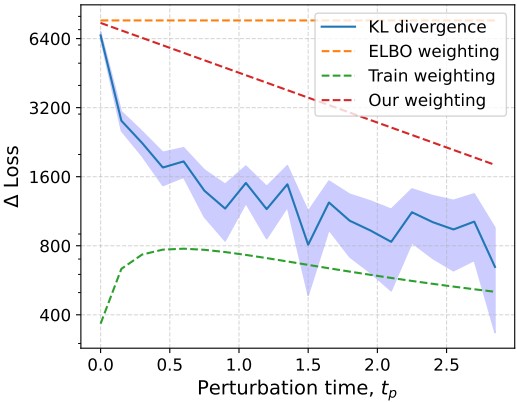

*Figure 5.* Sensitivity of the KL divergence and SM losses to score perturbations at different noise levels. Our weighting tracks the KL sensitivity more closely than the ELBO weighting while remaining an upper bound.

**Toy experiment.** We consider a toy setting where the data distribution is uniform on a unit circle embedded in $\mathbb{R}^2$ with a training set of equally spaced points around the circle. We train a small DM and since the setting is low dimensional, we can directly estimate the KL divergence of model samples via a histogram approximation. In Figure 3, we plot the test curves at different timesteps (*i.e.*, the test denoising score matching loss for some times $t \in [0, T]$, which is, up to an additive constant, an estimation of $\mathbb{E}[\varepsilon_t^\theta]$), compared against the KL divergence over the training trajectory. Our results are gathered in Figure 3. We find that the test curves at different timesteps are quite different: at larger timesteps, the test score matching error reaches its minimum further into training, or even never reaches it. Furthermore, we find that the test curves at smaller timesteps more closely track the KL divergence over the training trajectory, identifying a clear point of inflection. This agrees with our theory that smaller noise scales are more critical for model evaluation.

**Evaluating regularity properties of the U-Net.** Next, we assess the validity of the Lipschitz and dissipativity assumptions used throughout our work in a high dimensional realistic setting. We use a DDPM U-Net model trained on CIFAR-10. In Figure 4, we compute the quantity,

$$\kappa(x,y) = \frac{\langle s(y,t) - s(x,t), x-y \rangle}{\|x-y\|^2}, \tag{10}$$

for different values of $x$ and $y$ and plot it against $\|x - y\|$. This allows us to evaluate the value of the smoothness constant, $M_t = \sup_{x,y} \kappa(x,y)$. We find that $\kappa(x,y)$ is determined largely by $\|x-y\|$ and find a similar thing when $x$ and $y$ are based on separate test points. For larger $t$ it is clear that the quantity is both upper and lower bounded, verifying the assumptions used in this work. Across timesteps, it seems that $\kappa(x,y)$ is largest when $\|x - y\|$ is small with values growing as $t$ is taken small.

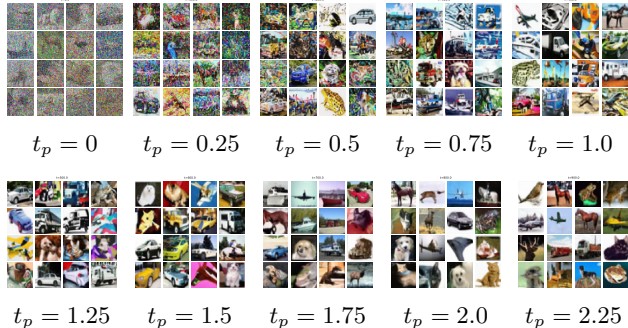

$t_p = 0 \qquad t_p = 0.25 \qquad t_p = 0.5 \qquad t_p = 0.75 \qquad t_p = 1.0$

$t_p = 1.25 \qquad t_p = 1.5 \qquad t_p = 1.75 \qquad t_p = 2.0 \qquad t_p = 2.25$

*Figure 6.* CIFAR-10 samples with score perturbed at different times, $t_p$. Small $t_p$ yields grainy, locally corrupted images and large $t_p$ alters global content.

**Score perturbations on CIFAR-10.** Remaining in the same setting as the previous section, we consider a perturbed version of the score: $\tilde{s}(x,t) = s(x,t) + 1_{t_p \leqslant t \leqslant t_p + \delta} \xi$ , where $\xi$ is a Gaussian vector. We consider how changing $t_p$ impacts the KL divergence and score matching losses. In Figure 6, we see that changing $t_p$ greatly changes the character of samples, with low values of $t_p$ producing grainy images and large values of $t_p$ creating subtle but more global, semantically significant changes. In Figure 5 we plot how the KL divergence of samples changes as $t_p$ is increased and we compare this with how different weightings of score matching losses change. We find that our weighting, with a crudely estimated regularity constant, better predicts the sensitivity of perturbations than the ELBO loss, whilst faithfully remaining an upper bound.

## 6. Conclusion

In this paper, we explored the score matching gap for DMs, *i.e.* the fundamental discrepancy between the learning goal used in practice and the theoretical quantity we aim to control. We first provided theoretical support for the need of additional structural assumptions to improve the classical score matching bound. We then revisited the ELBO and obtained a time-decaying version through the regularity properties of the score estimators. Then, we extended the score matching gap to two alternative topologies: the Wasserstein one, leading to similar qualitative conclusions, and the reverse KL, where regularity of the data distribution is needed instead of regularity of the score network. Finally, we empirically investigated our theory on a toy setting.

**Limitations & future works.** Several directions remain to be studied. In particular, tightening the log-Sobolev estimates of Section 3.2 might lead to improved bounds and even more informative decay terms. Alternatively, we believe that our technical insights might be exploited in the convergence analysis literature and tighten the time-dependence of several existing results.

## Impact Statement

Our work is largely theoretical, and we do not expect it to have any particular ethical or societal impact.

## Acknowledgements

U.S. is partially supported by the French government under the management of Agence Nationale de la Recherche as part of the "Investissements d'avenir" program, reference ANR-19-P3IA-0001 (PRAIRIE 3IA Institute). B.D., T.F., M.H, and U.S. are supported by the European Research Council Starting Grant DYNASTY – 101039676. A.D. is supported by the France 2030 program with the reference ANR-25-PEIA-0001 (THEOREM project). A.D. is funded by the European Union (ERC-2022-SYG-OCEAN-101071601). Views and opinions expressed are however those of the author(s) only and do not necessarily reflect those of the European Union or the European Research Council Executive Agency. Neither the European Union nor the granting authority can be held responsible for them. A.D. is supported by Hi! Paris and Agence Nationale de la Recherche (Grant 11-LABX-0047). This work received government funding administered by the National Research Agency (ANR) under the France 2030 program "Hi! PARIS", grant number ANR-23-IACL-0005. The authors would like to thank Giovanni Conforti for valuable comments and stimulating discussions.

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

## A. Additional technical background

In this section, we provide some additional technical background related to diffusion processes, namely, Fokker-Planck equations, stability properties of the log-Sobolev inequalities, and the reflection couplings as presented in (Eberle, 2016).

### A.1. Fokker-Planck Equations

We give below a brief technical background on the Fokker-Planck equations satisfied by the backward processes. The notations are the same as in Section 2. Note that, in the whole paper, we use $\partial_t$ for $\partial/\partial t$.

Whenever the data distribution satisfies $\mu \ll \gamma^d$, the Lebesgue density $(\overleftarrow{p}_t)_{t \geqslant 0}$ satisfies the Fokker-Planck (or forward Kolmogorov) equation (Barbu & Röckner, 2024; Risken, 1996; Bogachev et al., 2015),

$$\partial_t \overleftarrow{p}_t = \Delta \overleftarrow{p}_t - \nabla \cdot (\overleftarrow{p}_t b_{T-t}) , \quad t \in [0, T) , \tag{11}$$

where $b_t(x) := 2\nabla \log \tilde{p}(t, x) - x$ for all $x \in \mathbb{R}^d$.

Similar to Equation (11), under mild conditions, $(\overleftarrow{p}_t^\theta)_{t \geqslant 0}$ satisfies the Fokker-Planck equation, in the sense of distributions,

$$\partial_t \overleftarrow{p}_t^\theta = \Delta \overleftarrow{p}_t^\theta - \nabla \cdot (\overleftarrow{p}_t^\theta b_{T-t}^\theta) , \quad \overleftarrow{p}_0^\theta := \mathrm{d}\gamma^d/\mathrm{d}x , \tag{12}$$

with $b_t^\theta(x) := s^\theta(t, x) - x$. In all the following, we assume that the score network $(t, x) \mapsto s^\theta(t, x)$ is jointly continuous for all $\theta \in \Theta$.

These equations play a central role in our analysis, as they serve as the basis of the entropy flow computations, as it already appeard in existing works (Bogachev et al., 2016; Borovykh et al., 2023).

### A.2. Estimation of log-Sobolev constants

We recall the following classical stability properties of the logarithmic Sobolev inequality (Bakry et al., 2014; Chourasia et al., 2021). We include proofs for the sake of completeness.

**Lemma A.1** (Stability by Lipschitz mappings)**.** *Assume that $\mu$ satisfies the log-Sobolev inequality with constant $\rho$ and let $T : \mathbb{R}^d \to \mathbb{R}^d$ be a $L$-Lipschitz-continuous mapping, then the pushforward measure $T_{\#}\mu$ satisfies the log-Sobolev inequality with constant $L^2\rho$.*

*Proof.* Let $f$ be a smooth function and let $\Phi(x) := x \log(x)$.

$$\begin{aligned}
\mathrm{Ent}_{T_{\#}\mu}(f) &= \int \Phi(f \circ T)\mathrm{d}\mu - \Phi\left(\int f \circ T \mathrm{d}\mu\right) \\
&\leqslant \frac{\rho}{2} \int \frac{\|\nabla(f \circ T)\|^2}{f \circ T}\mathrm{d}\mu \\
&\leqslant \frac{L^2\rho}{2} \int \frac{\|\nabla f\|^2}{f} \circ T \mathrm{d}\mu ,
\end{aligned}$$

where we used that the spectral norm of $\mathrm{Jac}(T)$ is bounded by $L$. $\qquad\square$

**Lemma A.2** (Stability by convolution). *Let $\mu_1$ and $\mu_2$ be two Borel probability distributions on $\mathbb{R}^d$ that satisfy the log-Sobolev inequality with constants $\rho_1$ and $\rho_2$, respectively. Then $\mu_1 * \mu_2$ satisfies the log-Sobolev with constant $\rho_1 + \rho_2$.*

*Proof.* Let $f$ be positive and smooth enough, we have

$$\mathrm{Ent}_{\mu_1 * \mu_2}(f) = \mathrm{Ent}_{\mu_1 \otimes \mu_2}(\tilde{f}),$$

where $\tilde{f} : \mathbb{R}^d \times \mathbb{R}^d \to \mathbb{R}^d$ is defined by $\tilde{f}(x, y) = f(x + y)$. By tensorization of the entropy, we obtain that

$$\mathrm{Ent}_{\mu_1 * \mu_2}(f) \leqslant \int \left( \mathrm{Ent}_{\mu_1}^{(1)}(\tilde{f}) + \mathrm{Ent}_{\mu_2}^{(2)}(\tilde{f}) \right) \mathrm{d}(\mu_1 \otimes \mu_2),$$

where $\mathrm{Ent}_{\mu_1}^{(i)}(\tilde{f})$ means that we only integrate with respect to the $i$-th variable. By the log-Sobolev inequalities for $\mu_1$ and $\mu_2$, we immediately obtain the result. □

### A.3. Reflection couplings and Eberle's result

Consider an SDE of the form $\mathrm{d}X_t = b(X_t)\mathrm{d}t + \sqrt{2}\mathrm{d}B_t$. To obtain a Wasserstein contraction, Eberle (2016) uses a reflection coupling of the form,

$$\begin{aligned}
\mathrm{d}X_t &= b(X_t)\mathrm{d}t + \sqrt{2}\mathrm{d}B_t, \\
\mathrm{d}Y_t &= b(Y_t)\mathrm{d}t + \sqrt{2}(I - 2e_t e_t^T)\mathrm{d}B_t, \qquad \forall t < T, \quad X_t = Y_t, \forall t \geqslant T,
\end{aligned}$$

where $e_t = (X_t - Y_t)/\|X_t - Y_t\|$ and $T = \inf\{t \geqslant 0 : X_t = Y_t\}$. This coupling reflects the Brownian motion along the direction connecting the two processes, which helps control their distance. Using Itô's lemma, they show that for any nice function $f : \mathbb{R}_+ \to \mathbb{R}_+$,

$$\mathrm{d}f(r_t) = f'(r_t) \frac{\langle X_t - Y_t, b(X_t) - b(Y_t) \rangle}{\|X_t - Y_t\|} \mathrm{d}t + 4f''(r_t)\mathrm{d}t + 2\sqrt{2}f'(r_t)\mathrm{d}B_t,$$

where $r_t = \|X_t - Y_t\|$. Defining $\kappa$ as,

$$\kappa(r) = \inf \left\{ -\frac{\langle x - y, b(x) - b(y) \rangle}{\|x - y\|^2}, \ x, y \in \mathbb{R}^d, \ \|x - y\| = r \right\}, \tag{13}$$

we have that, for contractions in the Wasserstein metric $\mathscr{W}_f$ with rate $c$, it is sufficient to have

$$f''(r) - \frac{1}{4}r\kappa(r)f'(r) \leqslant -\frac{c}{4}f(r). \tag{14}$$

They then prove the following lemma.

**Lemma A.3** ((Eberle, 2016)). *Given any function $\kappa : \mathbb{R}_+ \to \mathbb{R}$ satisfying,*

$$\liminf_{r \to \infty} \kappa(r) \geqslant 0, \qquad \int_0^1 r\kappa(r)^- \mathrm{d}r < \infty,$$

*there exists a twice-differentiable, strictly-concave, increasing function $f : \mathbb{R}_+ \to \mathbb{R}_+$ and a constant $c > 0$ such that for a.e. $r > 0$,*

$$f''(r) - \frac{1}{4}r\kappa(r)f'(r) \leqslant -\frac{c}{4}f(r).$$

The construction of the function $f$ and contraction rate $c$ proceeds as follows. First, define the parameters:

$$R_0 = \inf\{R \geqslant 0 : \kappa(r) \geqslant 0 \ \forall r \geqslant R\}, \qquad R_1 = \inf\{R \geqslant R_0 : \kappa(r)R(R - R_0) \geqslant 8 \ \forall r \leqslant R\} \tag{15}$$

Next, define the auxiliary functions,

$$\varphi(r) = \exp\left(-\frac{1}{4}\int_0^r s\kappa^-(s)\mathrm{d}s\right), \qquad \Phi(r) = \int_0^r \varphi(s)\mathrm{d}s,$$

where $\kappa^-(s) = \max\{-\kappa(s), 0\}$ denotes the negative part. The contraction rate $c > 0$ is then given by

$$\frac{1}{c} = \frac{1}{2} \int_0^{R_1} \frac{\Phi(s)}{\varphi(s)} \mathrm{d}s.$$

To construct the distance function $f$, first define $g : [0, \infty) \to \mathbb{R}_+$ by

$$g(r) = 1 - \frac{c}{4} \int_0^{r \wedge R_1} \frac{\Phi(s)}{\varphi(s)} \mathrm{d}s.$$

Finally, the strictly concave, increasing function $f : [0, \infty) \to [0, \infty)$ is defined by,

$$f(r) = \int_0^r \varphi(s) g(s) \mathrm{d}s.$$

### A.3.1. SPECIAL CASE: DISSIPATIVITY AT DISTANCE AND LEMMA 4.7

In this subsection, we explicit the constants and functions above in the case of dissipativity at distance (Assumption 3.9), following the derivations of Eberle (2016).

Assume that the drift $b : \mathbb{R}^d \to \mathbb{R}^d$ satisfies the $(L, K, R/2)$-dissipativity at distance condition, in the sense of Assumption 3.9. In particular, this implies that there are non-negative constants, $L, K, R \geqslant 0$ such that

$$\frac{\langle x - y, b(x) - b(y) \rangle}{\|x - y\|^2} \leqslant \begin{cases} L, & \|x - y\| \leqslant R, \\ -K, & \|x - y\| \geqslant R. \end{cases}$$

Note that this implies, in particular, that

$$\kappa(r) \geqslant \begin{cases} -L, & r \leqslant R, \\ K, & r \geqslant R. \end{cases}$$

The next lemma provides the explicit rate of contraction for the distance $\mathscr{W}_f$, under dissipativity at distance.

Eberle (2016) provides a simplification of the contraction rate in the Lemma below.

**Lemma A.4.** *(Eberle, 2016) Assume that*

$$\kappa(r) \geqslant \begin{cases} -L, & r \leqslant R, \\ K, & r \geqslant R. \end{cases}$$

*We have that*

$$\frac{2}{c} \leqslant \int_0^{R_0} \left( r \wedge \sqrt{\frac{2\pi}{L}} \right) e^{\frac{Lr^2}{8}} \mathrm{d}r + \frac{4}{K} + \sqrt{\frac{8}{K}} R_0 e^{\frac{LR_0^2}{8}}.$$

*In particular, if $LR_0^2 \leqslant 8$, we have that*

$$\frac{2}{c} \leqslant \frac{e - 1}{2} R^2 + e\sqrt{\frac{8}{K}} R + \frac{4}{K}.$$

*and if $LR_0^2 > 8$, we have that*

$$\frac{2}{c} \leqslant \frac{8\sqrt{2\pi}}{R\sqrt{L}} \left( \frac{1}{L} + \frac{1}{K} \right) e^{\frac{LR^2}{8}} + \frac{32}{R^2 K^2}.$$

**Constants appearing in Lemma 4.7.** We can simplify the metric function by setting

$$\kappa_0(r) := \begin{cases} -L, & r \leqslant R, \\ K, & r \geqslant R, \end{cases}$$

so that $\kappa \geqslant \kappa_0$. By noting that $R_0$, $R_1$, and $1/c$ are decreasing functions of $\kappa$, we can make the following choices for the functions and constants.

- $R_0 = R$.

- $R_1$ is the smallest value greater than $R_0$ that satisfies $KR_1(R_1 - R) \geqslant 8$, *i.e.*,

$$R_1 = \frac{R}{2} + \sqrt{\frac{R^2}{4} + \frac{8}{K}} \ .$$

- The functions $\varphi$ and $\Phi$ are then given by,

$$\varphi(r) = \exp\left(-\frac{L}{8}(r \wedge R)^2\right), \qquad \Phi(r) = \sqrt{\frac{2\pi}{L}} \operatorname{erf}\left((r \wedge R)\sqrt{\frac{L}{8}}\right) + (r - R)_+ \exp\left(-\frac{LR^2}{8}\right). \quad (16)$$

The resulting function $f$ and the contraction rate $c$ are then given as in Section A.3. In particular, we can take

$$\frac{2}{c} \leqslant \int_0^{R_0} \left(r \wedge \sqrt{\frac{2\pi}{L}}\right) e^{\frac{Lr^2}{8}} \mathrm{d}r + \frac{4}{K} + \sqrt{\frac{8}{K}} R_0 e^{\frac{LR_0^2}{8}} \ .$$

Then, the formulas of Lemma A.4 are valid after replacing $R_0$ by $R$ in all the expressions.

We use the notation $f_{L,R,K}$ and $c_{L,R,K}$ for the resulting function and contraction rate, respectively.

In the following lemmas, we prove some additional properties of the functions constructed above. These results complement the construction of Eberle (2016) and will be particularly useful for our analysis. They may be seen as an additional technical contribution.

**Lemma A.5.** *The function $f_{L,R,K} : \mathbb{R}_+ \to \mathbb{R}_+$ is everywhere non-increasing in $L$.*

*Proof.* As we have

$$f(r) := \int_0^r \varphi(s)g(s)\mathrm{d}s \ ,$$

our proof strategy is to analyze the positive functions $\varphi$ and $g$ separately. Let us define

$$\mathcal{R}(r; R', R, L) := \frac{\int_0^{r \wedge R'} \frac{\Phi_{L,R}(s)}{\varphi_{L,R}(s)}\mathrm{d}s}{\int_0^{R'} \frac{\Phi_{L,R}(s)}{\varphi_{L,R}(s)}\mathrm{d}s} \ .$$

To show that $f_{L,R,K}(r)$ is non-increasing in $L$, we note that $\varphi_{L,R}(s)$ is non-increasing in $L$, and so it is sufficient to show that $\frac{\partial}{\partial R}(g_{L,R,K}(s)) \leqslant 0$ almost everywhere in $s$. Since $R_1$ is increasing in $L$, it is sufficient to show that $\frac{\partial \mathcal{R}}{\partial L}(r; R', R, L), \geqslant 0$ almost everywhere in $r$. We have

$$\frac{\frac{\partial}{\partial L}\varphi_{L,R}(r)}{\varphi_{L,R}(r)} = -\frac{1}{8}(r \wedge R)^2 \ .$$

Therefore,

$$\begin{aligned}
\frac{\partial}{\partial L}\left(\frac{\Phi_{L,R}(r)}{\varphi_{L,R}(r)}\right) &= \frac{\int_0^r \frac{\partial}{\partial L}\varphi_{L,R}(s)\mathrm{d}s}{\varphi_{L,R}(r)} - \frac{\Phi_{L,R}(r)}{\varphi_{L,R}(r)^2}\frac{\partial}{\partial L}\varphi_{L,R}(r) \\
&= \frac{\Phi_{L,R}(r)}{\varphi_{L,R}(r)}\left(\frac{\int_0^r \frac{\partial}{\partial L}\varphi_{L,R}(s)\mathrm{d}s}{\int_0^r \varphi_{L,R}(s)\mathrm{d}s} - \frac{\frac{\partial}{\partial L}\varphi_{L,R}(r)}{\varphi_{L,R}(r)}\right) \\
&= \frac{\Phi_{L,R}(r)}{\varphi_{L,R}(r)}\left(\int_0^r \frac{\frac{\partial}{\partial L}\varphi_{L,R}(s)}{\varphi_{L,R}(s)}q(\mathrm{d}s) - \frac{\frac{\partial}{\partial L}\varphi_{L,R}(r)}{\varphi_{L,R}(r)}\right) \ ,
\end{aligned}$$

where $q$ is a probability measure with support in $[0, r]$ satisfying $q(ds) \propto \varphi_{L,R}(s)\mathrm{d}s$. Thus, it follows from Jensen's inequality that,

$$\frac{\partial}{\partial L}\left(\frac{\Phi_{L,R}(r)}{\varphi_{L,R}(r)}\right) \geqslant \frac{\Phi_{L,R}(r)}{\varphi_{L,R}(r)}\left(\inf_{s \in [0,r]}\left\{\frac{\frac{\partial}{\partial L}\varphi_{L,R}(s)}{\varphi_{L,R}(s)}\right\} - \frac{\frac{\partial}{\partial L}\varphi_{L,R}(r)}{\varphi_{L,R}(r)}\right) = 0 \ ,$$

where the equality follows from the fact that $(\frac{\partial}{\partial L}\varphi_{L,R}(r))/\varphi_{L,R}(r)$ is decreasing. This concludes the proof. $\square$

**Lemma A.6.** *For any $R, L, K \geqslant 0$, we have constants $C_{L,R,K}, \delta_{L,R,K} > 0$ such that for all $r \leqslant \delta_{R,L,K}$,*

$$-C_{L,R,K}r \leqslant \frac{\mathrm{d}^2 f_{L,R,K}}{\mathrm{d}r^2}(r) \leqslant 0 \ .$$

*Proof.* The inequality on the right-hand side of the statement follows immediately from the concavity of $f_{L,R,K}$. For the left-hand side inequality, we note that, when $r < R$, we have

$$\frac{\mathrm{d}^2 f_{L,R,K}}{\mathrm{d}r^2}(r) = \varphi'_{L,R,K}(r)g_{L,R,K}(r) + \varphi_{L,R,K}(r)g'_{L,R,K}(r)$$

$$= -\frac{L}{4}r\varphi_{L,R,K}(r)g_{L,R,K}(r) - \frac{c_{L,R,K}}{4}\Phi(r) \ .$$

By using that $\varphi_{L,R,K}, g_{L,R,K} \leqslant 1$, we obtain

$$\frac{\mathrm{d}^2 f_{L,R,K}}{\mathrm{d}r^2}(r) \geqslant -r\left(c_{L,R,K} + \frac{L}{4}\right) \ .$$

This concludes the proof. $\qquad\square$

The following lemma allows to compare the Wasserstein distances $\mathscr{W}_f$ and $\mathscr{W}_1$.

**Lemma A.7.** *For all $r \geqslant 0$, we have $f_{L,R,K}(r) \leqslant r$ and $r \leqslant 2e^{\frac{LR^2}{8}} f_{L,R,K}(r)$.*

*Proof.* In the proof, we omit the dependence of $f$ on $(L, R, K)$. By construction, $f$ is concave and $f(0) = 0$, $f'(0) = 1$. Therefore, by concavity we have $f(r) \leqslant r$.

For the second part of the statement, we note that $\varphi$ and $g$ are decreasing. Moreover, $\varphi$ is constant for $r \geqslant R$ and $g$ is constant for $r \geqslant R_1$. Therefore, we have, for all $r \geqslant 0$

$$\varphi(r) \geqslant \varphi(R) = e^{-\frac{LR^2}{8}} \ , \quad , g(r) \geqslant g(R_1) = \frac{1}{2} \ .$$

Thus,

$$f(r) := \int_0^r g(s)\varphi(s)\mathrm{d}s \geqslant e^{-\frac{LR^2}{8}} \int_0^r g(s)\mathrm{d}s \geqslant e^{-\frac{LR^2}{8}} \frac{r}{2} \ .$$

This concludes the proof. $\qquad\square$

# B. Omitted proofs of Section 3

## B.1. Proofs for the negative result

In this section, we present the proof of the negative result in Proposition 3.1. We begin by recalling the necessary prerequisites for the dynamical Schrödinger bridge (SB) problem in the Ornstein–Uhlenbeck setting and provide two technical lemmas.

For any $\rho \in \mathcal{P}(\mathbb{R}^d)$, let $R^\rho$ denote the path measure of the Ornstein–Uhlenbeck process on $[0, T]$ with initial distribution $\rho$. For probability measures $\rho_0, \rho_1 \in \mathcal{P}(\mathbb{R}^d)$, we define the dynamical Schrödinger bridge problem,

$$\text{Minimize KL}\left(Q|R^\rho\right) \ , \qquad \text{subject to } Q_0 = \rho_0, \ Q_T = \rho_1 \ . \tag{17}$$

Under the conditions $\text{KL}\left(\rho_0|\gamma^d\right), \text{KL}\left(\rho_1|\gamma^d\right) < \infty$, it is known that it admits a unique solution (Léonard, 2013, Theorem 2.12). Furthermore, due to the decomposition,

$$\text{KL}\left(Q|R^\rho\right) = \mathbb{E}_{Q_0}\left[\text{KL}\left(Q_{0:T|0}|R^\rho_{0:T|0}\right)\right] + \text{KL}\left(Q_0|R^\rho_0\right) = \mathbb{E}_{\rho_0}\left[\text{KL}\left(Q_{0:T|0}|R_{0:T|0}\right)\right] + \text{KL}\left(\rho_0|\rho\right) , \tag{18}$$

we see that the solution $Q$, will not depend on the initial law of the reference, $\rho$: any two OU references $R^\rho, R^{\rho'}$ with $\text{KL}\left(\rho_0|\rho\right), \text{KL}\left(\rho_0|\rho'\right) < \infty$ share the same Markov kernel and hence, the two resulting SB problems share the same

solution, which in this section, we denote by $Q^*$. Taking $\rho = \gamma^d$ for convenience and $R := R^{\gamma^d}$, we recall the classical result that the path measure $Q^*$ factorizes as,

$$dQ^*(\omega) = f(\omega_0)\, g(\omega_T)\, dR(\omega)\,, \tag{19}$$

for measurable Schrödinger potentials $f, g : \mathbb{R}^d \to \mathbb{R}_+$ (Conforti, 2021). Since $R$ has marginal $\gamma^d$ at every time, the pair of Schrödinger equations characterizing $f, g$ reduces to

$$\frac{d\rho_0}{d\gamma^d}(z) = f(z) \cdot (P_T g)(z)\,, \qquad \frac{d\rho_1}{d\gamma^d}(x) = (P_T f)(x) \cdot g(x)\,. \tag{20}$$

Moreover, the optimal $Q^*$ is Markov (Léonard, 2013, Proposition 2.10) and inherits from $R$ the structure of a diffusion process on $[0, T]$ with the same diffusion coefficient. Furthermore, its drift is given by the Doob $h$-transform,

$$b^Q(t, x) = -x + 2\nabla \log(P_{T-t} g)(x)\,, \tag{21}$$

which is well defined for almost all $t$. For a more substantial introduction to Schrödinger bridge problems, we refer the read to the notes of (Nutz, 2021) and (Conforti, 2021) as well as the survey of (Léonard, 2013).

The first lemma is a technical property showing that the conditional distribution $Q^*_{T|0}$ is 1-strongly log-concave.

**Lemma B.1.** *Suppose that $\rho_1 = \gamma^d$ and $\mathrm{KL}\left(\rho_0 | \gamma^d\right) < \infty$, then for any $z \in \mathbb{R}^d$ and $T > 0$, the measure $Q^*_{T|0}(\cdot | z)$ is 1-strongly log-concave.*

*Proof.* By (19) and (20), we obtain the density,

$$\frac{dQ^*_{T|0}}{dR_{T|0}}(x \mid z) = \frac{(dQ^*_{0,T}/dR_{0,T})(z, x)}{(dQ^*_0/dR_0)(z)} = \frac{f(z)g(x)}{f(z)\, P_T g(z)} = \frac{1}{P_T f(x)\, P_T g(z)}\,, \tag{22}$$

where the final equality follows from (20) and the fact that $\rho_1 = \gamma^d$. Furthermore, the conditional distribution $dR_{T|0}(\cdot \mid z)$ coincides exactly with the forward transition kernel started at $z$, and therefore has Lebesgue density, $\overrightarrow{p}_{T|0}(\cdot \mid z)$. Thus, by the chain rule, we have that $Q^*_{T|0}$ is absolutely continuous with respect to the Lebesgue measure and has density,

$$q^*_{T|0}(x \mid z) = \frac{\overrightarrow{p}_{T|0}(x \mid z)}{P_T f(x)\, P_T g(z)}\,.$$

Since $T > 0$, $P_T f$ and $\overrightarrow{p}_{T|0}(\cdot \mid z)$ are both smooth functions bounded away from $0$ and hence the logarithm is almost-everywhere twice-differentiable and satisfies,

$$\nabla^2_x \log q^*_{T|0}(x \mid z) = \nabla^2_x \log \overrightarrow{p}_{T|0}(x|z) - \nabla^2_x \log P_T f(x) = -\frac{1}{1 - e^{-2T}} I_d - \nabla^2_x \log P_T f(x)\,.$$

To control $\nabla^2_x \log P_T f$ we note that $P_T f(x) = (f * \varphi_{1-e^{-2T}})(e^{-T} x)$, where $\varphi_s$ denotes the centered Gaussian density of variance $s$. Using the log-semi-convexity property of the Ornstein-Uhlenbeck semigroup (see, e.g., (Gozlan et al., 2023)), we have

$$\nabla^2_x \log P_T f(x) = e^{-2T} \nabla^2 \log(f * \varphi_{1-e^{-2T}})(e^{-T} x) \succcurlyeq -\frac{e^{-2T}}{1 - e^{-2T}} I_d\,.$$

From which, we obtain the final bound,

$$\nabla^2_x \log q^*_{T|0}(x \mid z) \preccurlyeq -I_d\,,$$

and that $Q^*_{T|0}$ is 1-strongly log-concave. $\qquad\square$

The second lemma bounds the gradient of the Schrödinger potential and follows from Proposition 2.4 of (Chiarini et al., 2022).

**Lemma B.2.** *Let $f$ be as in Equation (20). Suppose that $\rho_1 = \gamma^d$, $\mathrm{KL}\left(\rho_0 | \gamma^d\right) < \infty$ and $T \geqslant \frac{1}{2}\log(4)$, then we have that,*

$$\int \|\nabla \log P_T f\|^2 d\gamma^d \leqslant \frac{4}{e^{2T} - 1}\left(\mathrm{KL}\left(\rho_0 | \gamma^d\right) + e^{-2T} d\right)\,. \tag{23}$$

*Proof.* From Proposition 2.4 of (Chiarini et al., 2022), we obtain the bound,

$$\int \|\nabla \log P_T f\|^2 \mathrm{d}\gamma^d \leqslant \frac{2}{e^{2T}-1} \inf_{\pi} \mathrm{KL}\left(\pi | R_{0,T}\right) , \tag{24}$$

where the infimum is taken over all couplings between $\rho_0$ and $\gamma^d$. We upper bound this infimum by choosing the naïve coupling, $\pi = \rho_0 \otimes \gamma^d$, producing the bound,

$$\inf_{\pi} \mathrm{KL}\left(\pi | R_{0,T}\right) \leqslant \mathrm{KL}\left(\rho_0 \otimes \gamma^d | R_{0,T}\right) = \mathbb{E}_{X \sim \gamma^d}\left[\mathrm{KL}\left(\rho_0 | R_{0|T}(\cdot \mid X)\right)\right] . \tag{25}$$

By reversibility of the Ornstein–Uhlenbeck process with respect to $\gamma^d$, we have that,

$$R_{0,T}(\mathrm{d}z, \mathrm{d}x) = \overrightarrow{p}_{T|0}(x \mid z)\gamma^d(z)\,\mathrm{d}z\,\mathrm{d}x = \overrightarrow{p}_{T|0}(z \mid x)\gamma^d(x)\,\mathrm{d}z\,\mathrm{d}x ,$$

and consequently, the conditional distribution $\mathrm{d}R_{0|T}(z \mid x)$ coincides exactly with the forward transition kernel started at $x$, and therefore has Lebesgue density, $\overrightarrow{p}_{T|0}(z \mid x)$. We then relate this to the quantity $\mathrm{KL}\left(\rho_0 | \gamma^d\right)$ by first developing the identity,

$$\mathrm{KL}\left(\rho_0 | \overrightarrow{p}_{T|0}(\cdot \mid x)\right) - \mathrm{KL}\left(\rho_0 | \gamma^d\right) = -\mathbb{E}_{Z \sim \rho_0}\left[\log \overrightarrow{p}_{T|0}(Z|x)\right] + \mathbb{E}_{Z \sim \rho_0}\left[\log \gamma^d(Z)\right]$$

$$= \frac{d}{2}\log(1 - e^{-2T}) + \frac{1}{2(1 - e^{-2T})}\mathbb{E}_{Z \sim \rho_0}[\|Z - e^{-T}x\|^2] - \frac{1}{2}\mathbb{E}_{Z \sim \rho_0}[\|Z\|^2]$$

$$= \frac{d}{2}\log(1 - e^{-2T}) + \frac{e^{-2T}}{2(1 - e^{-2T})}(\mathbb{E}_{Z \sim \rho_0}[\|Z\|^2] + \|x\|^2) - \frac{e^{-T}}{1 - e^{-2T}}\mathbb{E}_{Z \sim \rho_0}[\langle Z, x\rangle].$$

To control the second moment $\mathbb{E}_{Z \sim \rho_0}[\|Z\|^2]$, we apply the Donsker-Varadhan variational principle to obtain,

$$\frac{1}{4}\mathbb{E}_{Z \sim \rho_0}[\|Z\|^2] \leqslant \mathrm{KL}\left(\rho_0 | \gamma^d\right) + \log \mathbb{E}_{Z \sim \gamma^d}[e^{\|Z\|^2/4}]$$

$$\leqslant \mathrm{KL}\left(\rho_0 | \gamma^d\right) + \frac{d}{2}.$$

With this, we obtain the upper bound,

$$\mathbb{E}_{X \sim \gamma^d}\left[\mathrm{KL}\left(\rho_0 | R_{0|T}(\cdot \mid X)\right)\right] \leqslant \mathrm{KL}\left(\rho_0 | \gamma^d\right) + \frac{e^{-2T}}{2(1 - e^{-2T})}(4\mathrm{KL}\left(\rho_0 | \gamma^d\right) + 3d) \tag{26}$$

$$\leqslant 2\mathrm{KL}\left(\rho_0 | \gamma^d\right) + 2e^{-2T}d, \tag{27}$$

where the final inequality follows from $T \geqslant \frac{1}{2}\log(4)$. Substituting (25) and (27) into (24), we obtain the bound in the statement. $\square$

### B.1.1. PROOF OF PROPOSITION 3.1

We instantiate the Schrödinger bridge problem (17) with $\rho_0 = \nu$ and $\rho_1 = \gamma^d$. Since $\mathrm{KL}\left(\nu | \gamma^d\right) < \infty$, it follows that a unique solution exists and we denote the resulting bridge by $Q^*$. We also denote the Schrödinger potentials by $f, g$ which satisfy (19) and, as a result of (20), satisfy $P_T f \cdot g \equiv 1$. It is this optimal path measure $Q^*$ which we use to choose the score function that closes the score matching gap.

By the bridging property (Léonard, 2013, Proposition 2.3), the path-space KL divergence reduces to the joint endpoints,

$$\mathrm{KL}\left(R|Q^*\right) = \mathrm{KL}\left(R_{0,T}|Q^*_{0,T}\right) + \int \mathrm{KL}\left(R_{\cdot|0,T}|Q^*_{\cdot|0,T}\right)\mathrm{d}R_{0,T}$$

$$= \mathrm{KL}\left(R_{0,T}|Q^*_{0,T}\right) ,$$

or equivalently, $\mathrm{KL}\left(R_{\cdot|0,T}|Q^*_{\cdot|0,T}\right) = 0$ almost everywhere. Similarly, we can apply this property to the reference measure $R^\mu$ starting at initial distribution $\mu$ to obtain,

$$\mathrm{KL}\left(R^\mu|Q^*\right) = \mathrm{KL}\left(R^\mu_{0,T}|Q^*_{0,T}\right) + \int \mathrm{KL}\left(R_{\cdot|0,T}|Q^*_{\cdot|0,T}\right)\mathrm{d}R^\mu_{0,T}$$

$$= \mathrm{KL}\left(R^\mu_{0,T}|Q^*_{0,T}\right) , \tag{28}$$

where we use that $R^\mu_{:|0,T} = R_{:|0,T}$. Furthermore, the chain rule for the KL divergence yields,

$$\mathrm{KL}\left(R^\mu_{0,T}|Q^*_{0,T}\right) = \mathrm{KL}\left(\mu|\nu\right) + \Delta_T, \qquad \Delta_T := \int \mathrm{KL}\left(R^\mu_{T|0}(\cdot\mid x)|Q^*_{T|0}(\cdot\mid x)\right)\mathrm{d}\mu(x), \tag{29}$$

where we use that $Q^*_0 = \nu$. Combining (28) and (29), we obtain a correspondence between the marginal and path measure KL divergence:

$$\mathrm{KL}\left(\mu|\nu\right) = \mathrm{KL}\left(R^\mu|Q^*\right) - \Delta_T. \tag{30}$$

Next, we control the quantity $\Delta_T$. From Lemma B.1, we have that for any $z \in \mathbb{R}^d$, $Q^*_{T|0}(\cdot|z)$ is 1-strongly log-concave. Hence by the Bakry–Émery criterion (Bakry et al., 2014), it must satisfy a logarithmic Sobolev inequality (2.1), with constant 1. By (22), we have

$$\frac{\mathrm{d}R^\mu_{T|0}}{\mathrm{d}Q^*_{T|0}}(x\mid z) = P_T f(x)\, P_T g(z).$$

Thus, the logarithmic Sobolev inequality produces the bound,

$$\Delta_T \leqslant \frac{1}{2}\int\int \|\nabla \log P_T f\|^2\,\mathrm{d}R^\mu_{T|0}(\cdot|z)\,\mathrm{d}\mu(z) = \frac{1}{2}\int \|\nabla \log P_T f\|^2\,\mathrm{d}\overrightarrow{\mu}_T. \tag{31}$$

To apply the bound in Lemma B.2, we must replace $\overrightarrow{\mu}_T$ by $\gamma^d$. For this, we observe that for any $y \in \mathbb{R}^d$,

$$\tilde{p}_{T|0}(x|y) = (1 - e^{-2T})^{-d/2}\exp\left(\frac{\|y\|^2}{2} - \frac{e^{-2T}\|x - e^T y\|^2}{2(1 - e^{-2T})}\right) \leqslant (1 - e^{-2T})^{-d/2}\, e^{\|y\|^2/2}.$$

Since $\mathrm{d}\overrightarrow{\mu}_T/\mathrm{d}\gamma^d(x) = \int \tilde{p}_{T|0}(x|y)\,\mu(\mathrm{d}y)$, we thus obtain,

$$\frac{\mathrm{d}\overrightarrow{\mu}_T}{\mathrm{d}\gamma^d}(x) \leqslant (1 - e^{-2T})^{-d/2}\, e^{R^2/2}, \tag{32}$$

where we use that $\mathrm{supp}(\mu) \subseteq B_R(\mathbf{0})$. Thus, we conclude that combining (31) and (32) and applying Lemma B.2, we obtain,

$$\Delta_T \leqslant \frac{1}{2}(1 - e^{-2T})^{-d/2}\, e^{R^2/2}\int \|\nabla \log P_T f\|^2\,\mathrm{d}\gamma^d$$

$$\leqslant \frac{2(1 - e^{-2T})^{-d/2}\, e^{R^2/2}}{e^{2T} - 1}\left(\mathrm{KL}\left(\nu|\gamma^d\right) + e^{-2T}d\right).$$

Thus, whenever $T \geqslant \log d$, we have that

$$\Delta_T \leqslant 2e^{\frac{1+R^2}{2} - 2T}(1 + \mathrm{KL}\left(\nu|\gamma^d\right)),$$

and in particular, $\mathrm{KL}\left(R^\mu|Q^*\right) < \infty$.

By the argument of (21), the path measure $Q^*$ coincides with the forward SDE,

$$\mathrm{d}X_t = \left(-X_t + 2\nabla\log(P_{T-t}g)(X_t)\right)\mathrm{d}t + \sqrt{2}\,\mathrm{dB}_t, \qquad X_0 \sim \nu.$$

We now reverse $Q^*$ in time. Since $\mathrm{KL}\left(R^\mu|Q^*\right) < \infty$, it follows from (Cattiaux et al., 2023) that the reversed path measure $\overleftarrow{Q^*}$ is the law of the diffusion started at $\overleftarrow{Q^*}_0 = Q^*_T = \gamma^d$ with drift,

$$\overleftarrow{b}^{Q^*}(t, y) = y - 2\nabla\log(P_t g)(y) + 2\nabla\log q^*_{T-t}(y).$$

In fact, we can state the drift more directly in terms of the $fg$-decomposition. From (19), it follows that the marginals have density,

$$q^*_t(x) = (P_t f)(x)\,(P_{T-t}g)(x)\,\gamma^d(x), \tag{33}$$

where we use that, by reversibility, $\mathbb{E}_R[f(X_0) \mid X_t = x] = P_t f(x)$. Consequently, using $\nabla \log \gamma^d(x) = -x$,

$$\overleftarrow{b}^{Q^*}(t, y) = y - 2\nabla \log(P_t g)(y) + 2\big(\nabla \log(P_{T-t}f)(y) + \nabla \log(P_t g)(y) - y\big)$$
$$= -y + 2\nabla \log(P_{T-t}f)(y) \,.$$

Thus, by defining the score as,

$$s^\theta(\tau, x) := 2\nabla \log(P_\tau f)(x) \,, \tag{34}$$

the reverse process $\overleftarrow{X}_t^\theta$ coincides with the reverse path measure $\overleftarrow{Q}^*$.

Furthemore, the time-reversal $\overleftarrow{R}^\mu$ of the path measure $R^\mu$ also coincides with the true reverse process $\overleftarrow{X}_t$ defined in (4). Since the KL divergence is invariant under time reversal, we have that the reversal $\overleftarrow{R}^\mu$ of the path measure $R^\mu$ satisfies, $\mathrm{KL}\left(\overleftarrow{R}^\mu|\overleftarrow{Q}^*\right) = \mathrm{KL}\left(R^\mu|Q^*\right) < \infty$. Therefore, it follows from (Léonard, 2012) that a Girsanov-type identity holds:

$$\mathrm{KL}\left(\overleftarrow{R}^\mu|\overleftarrow{Q}^*\right) \geqslant \frac{1}{4}\int_0^T \mathbb{E}\left[\varepsilon_t^\theta(\overrightarrow{X}_t)^2\right] \mathrm{d}t + \mathrm{KL}\left(\overleftarrow{\mu}_0|Q_0\right) = \frac{1}{4}\int_0^T \mathbb{E}\left[\varepsilon_t^\theta(\overrightarrow{X}_t)^2\right] \mathrm{d}t + \mathrm{KL}\left(\overrightarrow{\mu}_T|\gamma^d\right) \,. \tag{35}$$

Combining the two inequalities (30) and (35) yields the bound in the statement. $\qquad\square$

### B.2. Technical lemmas

The next lemma regards the regularity of the true backward density $\overleftarrow{p}^\theta$ and is inspired from (Conforti et al., 2025).

**Lemma B.3.** *Assume that $\mu$ is absolutely continuous with respect to the Lebesgue measure. Then the density $(t, x) \mapsto \overleftarrow{p}_t$ of the true backward process is in $\mathrm{C}^{1,2}([0, T] \times \mathbb{R}^d)$.*

*Proof.* This follows from classical results (Bogachev et al., 2015). In particular, when $\mu$ is absolutely continuous with respect to the Lebesgue measure, it was proven by Conforti et al. (2025) that the map $(t, x) \mapsto \overrightarrow{p}_t(x)$ is in $\mathrm{C}^{1,2}((0, T] \times \mathbb{R}^d)$, where $\overrightarrow{p}_t$ is the density of the forward process (3). The results follows immediately from the relation $\overleftarrow{p}_t(x) := \overrightarrow{p}_{T-t}(x)$. $\qquad\square$

### B.3. Proof of Lemma 3.4

In this section, we prove Lemma 3.4. First, we recall that we use the notation $\partial_t$ for $\partial/\partial_t$, and

$$b_t(x) := 2\nabla \log \tilde{p}(t, x) - x \,, \qquad b_t^\theta(x) := s^\theta(t, x) - x \,.$$

*Proof.* (of Lemma 3.4) Note that our assumptions imply that the quantities appearing below are finite.

Let $\Phi(u) := u \log(u)$ with $\Phi(0) = 0$, and denote

$$v_t := \frac{\overleftarrow{p}_t}{\overleftarrow{p}_t^\theta} \,.$$

At least formally, we have for $t \in (0, T)$

$$\frac{\mathrm{d}}{\mathrm{d}t}\mathrm{KL}\left(\overleftarrow{\mu}_t|\overleftarrow{\mu}_t^\theta\right) = \frac{\mathrm{d}}{\mathrm{d}t}\int \Phi(v_t)\overleftarrow{p}_t^\theta \mathrm{d}x$$
$$= \int \Phi'(v_t)\left(\partial_t\overleftarrow{p}_t - v_t\partial_t\overleftarrow{p}_t^\theta\right)\mathrm{d}x + \int \Phi(v_t)\partial_t\overleftarrow{p}_t^\theta \mathrm{d}x$$
$$=: \mathrm{C}_{\mathrm{diffusion}} + \mathrm{C}_{\mathrm{score}} \,,$$

where we use the Fokker-Planck equations for both distributions and separate the contributions of the Laplace operators and the potential terms. This gives

$$\mathrm{C}_{\mathrm{diffusion}} := \int \Phi'(v_t)\left(\Delta\overleftarrow{p}_t - v_t\Delta\overleftarrow{p}_t^\theta\right)\mathrm{d}x + \int \Phi(v_t)\Delta\overleftarrow{p}_t^\theta \mathrm{d}x \,.$$

By integrating by parts, we obtain that

$$
\begin{aligned}
\mathrm{C}_{\mathrm{diffusion}} &= \int \left\{ v_t \Delta(\Phi'(v_t)) - \Delta(v_t \Phi'(v_t)) + \Delta(\Phi(v_t)) \right\} \overleftarrow{p}_t^\theta \mathrm{d}x \\
&= \int \left\{ -\Phi'(v_t) \Delta v_t - 2\Phi''(v_t) \|\nabla v_t\|^2 + \Delta(\Phi(v_t)) \right\} \overleftarrow{p}_t^\theta \mathrm{d}x \\
&= -\int \Phi''(v_t) \|\nabla v_t\|^2 \overleftarrow{p}_t^\theta \mathrm{d}x \\
&= -\mathscr{I}\left( \overleftarrow{\mu}_t | \overleftarrow{\mu}_t^\theta \right) \ ,
\end{aligned}
$$

where we noted that $\Phi''(x) := 1/x$. For the second term, we have

$$
\mathrm{C}_{\mathrm{score}} := -\int \Phi'(v_t) \left( \nabla \cdot (\overleftarrow{p}_t b_{T-t}) - v_t \nabla \cdot (\overleftarrow{p}_t^\theta b_{T-t}^\theta) \right) \mathrm{d}x - \int \Phi(v_t) \nabla \cdot (\overleftarrow{p}_t^\theta b_{T-t}^\theta) \mathrm{d}x \ .
$$

By integrating by parts again, we have

$$
\mathrm{C}_{\mathrm{score}} = \int \left\{ \Phi''(v_t) v_t \langle \nabla v_t, b_{T-t} \rangle - \Psi'(v_t) \langle \nabla v_t, b_{T-t}^\theta \rangle \right\} \overleftarrow{p}_t^\theta \mathrm{d}x \ ,
$$

with $\Psi(x) := x\Phi'(x) - \Phi(x)$ (in our case, $\Psi(x) = x$). This gives

$$
\mathrm{C}_{\mathrm{score}} = \int v_t \Phi''(v_t) \langle \nabla v_t, b_{T-t} - b_{T-t}^\theta \rangle \overleftarrow{p}_t^\theta \mathrm{d}x \ .
$$

Consider an arbitrary mapping $a : \mathbb{R}_+ \to (0, 1]$. Using the expression of $\Phi$ and the Cauchy-Schwarz and Young inequalities, we have

$$
\mathrm{C}_{\mathrm{score}} \leqslant a(t) \int \frac{\|\nabla v_t\|^2}{v_t} \overleftarrow{p}_t^\theta \mathrm{d}x + \frac{1}{4a(t)} \int \left\| b_{T-t} - b_{T-t}^\theta \right\|^2 v_t \overleftarrow{p}_t^\theta \mathrm{d}x \tag{36}
$$

$$
= a(t) \mathscr{I}\left( \overleftarrow{\mu}_t | \overleftarrow{\mu}_t^\theta \right) + \frac{1}{4a(t)} \int \left\| 2\nabla \log \tilde{p}_{T-t}(x) - s^\theta(T-t, x) \right\|^2 \overleftarrow{p}_t(x) \mathrm{d}x \ . \tag{37}
$$

The conclusion immediately follows by the change of variable $\alpha(t) := 1 - a(t)$. $\qquad\square$

## B.4. Proofs of Section 3.2

### B.4.1. PROOF OF THEOREM 3.5

*Proof.* (of Theorem 3.5) Let $t \in (0, T)$, by Lemma 3.4, we have, for any continuous mapping $C : \mathbb{R}_+ \to [0, 1)$,

$$
\frac{\mathrm{d}}{\mathrm{d}t} \mathrm{KL}\left( \overleftarrow{\mu}_t | \overleftarrow{\mu}_t^\theta \right) \leqslant -\alpha(t) \mathscr{I}\left( \overleftarrow{\mu}_t | \overleftarrow{\mu}_t^\theta \right) + \frac{1}{4(1-\alpha(t))} \mathbb{E}\left[ \left\| 2\nabla \log \tilde{p}_{T-t}(\overrightarrow{X}_{T-t}) - s^\theta(T-t, \overrightarrow{X}_{T-t}) \right\|^2 \right] \ .
$$

By the assumed logarithmic Sobolev inequality for $\overleftarrow{\mu}_t^\theta$, we have

$$
\frac{\mathrm{d}}{\mathrm{d}t} \mathrm{KL}\left( \overleftarrow{\mu}_t | \overleftarrow{\mu}_t^\theta \right) \leqslant -\frac{2\alpha(t)}{\rho(t)} \mathrm{KL}\left( \overleftarrow{\mu}_t | \overleftarrow{\mu}_t^\theta \right) + \frac{1}{4(1-\alpha(t))} \mathbb{E}\left[ \left\| 2\nabla \log \tilde{p}_{T-t}(\overrightarrow{X}_{T-t}) - s^\theta(T-t, \overrightarrow{X}_{T-t}) \right\|^2 \right] \ .
$$

Consider $\epsilon, \delta \in (0, T)$ with $T - \epsilon > \delta$. As $1/\rho$ is assumed to be continuous, by Grönwall's lemma (Gronwall, 1919), we obtain

$$
\begin{aligned}
\mathrm{KL}\left( \overleftarrow{\mu}_{T-\epsilon} | \overleftarrow{\mu}_{T-\epsilon}^\theta \right) \leqslant &\ \mathrm{KL}\left( \overleftarrow{\mu}_\delta | \overleftarrow{\mu}_\delta^\theta \right) \exp\left\{ -\int_\delta^{T-\epsilon} \frac{2\alpha(u)}{\rho(u)} \mathrm{d}u \right\} \\
&+ \int_\delta^{T-\epsilon} \exp\left\{ -\int_t^{T-\epsilon} \frac{2\alpha(u)}{\rho(u)} \mathrm{d}u \right\} \frac{\mathbb{E}\left[ \left\| 2\nabla \log \tilde{p}_{T-t}(\overrightarrow{X}_{T-t}) - s^\theta(T-t, \overrightarrow{X}_{T-t}) \right\|^2 \right]}{4(1-\alpha(t))} \mathrm{d}t \ .
\end{aligned}
$$

We know that the $t \mapsto \overrightarrow{X}_t$ is almost-surely continuous at $t = 0$ and that $t \mapsto \overleftarrow{X}_t^\theta$ is almost surely continuous at $t = T$. As almost sure convergence implies convergence in distribution, this implies the convergences in distribution $\overleftarrow{\mu}_t \rightharpoonup \mu$ and $\overleftarrow{\mu}_{T-t}^\theta \rightharpoonup \overleftarrow{\mu}_T^\theta$ as $t \to 0^+$. By the joint lower semi-continuity of relative entropy (van Erven & Harremoës, 2014, Theorem 19) with respect to the weak convergence of distributions and up to taking a sequence $\epsilon_n \to 0$, we can take the inferior limit as $\epsilon \to 0^+$ to get that

$$\mathrm{KL}\left(\mu | \overleftarrow{\mu}_T^\theta\right) \leqslant \exp\left\{-\int_\delta^T \frac{2\alpha(s)}{\rho(s)}\mathrm{d}s\right\} \mathrm{KL}\left(\overleftarrow{\mu}_\delta | \overleftarrow{\mu}_\delta^\theta\right)$$

$$+ \int_\delta^T \exp\left\{-\int_t^T \frac{2\alpha(s)}{\rho(s)}\mathrm{d}s\right\} \frac{\mathbb{E}\left[\left\|2\nabla \log \tilde{p}_{T-t}(\overrightarrow{X}_{T-t}) - s^\theta(T-t, \overrightarrow{X}_{T-t})\right\|^2\right]}{4(1-\alpha(t))}\mathrm{d}t \,.$$

Now we take the superior limit as $\delta \to 0^+$ to obtain that

$$\mathrm{KL}\left(\mu | \overleftarrow{\mu}_T^\theta\right) \leqslant \exp\left\{-\int_0^T \frac{2\alpha(s)}{\rho(s)}\mathrm{d}s\right\} \limsup_{\delta \to 0^+} \mathrm{KL}\left(\overleftarrow{\mu}_\delta | \overleftarrow{\mu}_\delta^\theta\right)$$

$$+ \int_0^T \exp\left\{-\int_t^T \frac{2\alpha(s)}{\rho(s)}\mathrm{d}s\right\} \frac{\mathbb{E}\left[\left\|2\nabla \log \tilde{p}_{T-t}(\overrightarrow{X}_{T-t}) - s^\theta(T-t, \overrightarrow{X}_{T-t})\right\|^2\right]}{4(1-\alpha(t))}\mathrm{d}t \,.$$

For the term $\mathrm{KL}\left(\overleftarrow{\mu}_\delta | \overleftarrow{\mu}_\delta^\theta\right)$, we note that our assumptions are sufficient to apply (Bogachev et al., 2016, Theorem 1.1), in particular we apply their Remark 1.3, corresponding to the case where the initializations of the two processes differ. This gives us that

$$\mathrm{KL}\left(\overleftarrow{\mu}_\delta | \overleftarrow{\mu}_\delta^\theta\right) \leqslant \mathrm{KL}\left(\overrightarrow{\mu}_T | \gamma^d\right) + \int_0^\delta \mathbb{E}\left[\left\|2\nabla \log \tilde{p}_{T-t}(\overrightarrow{X}_{T-t}) - s^\theta(T-t, \overrightarrow{X}_{T-t})\right\|^2\right]\mathrm{d}t$$

$$\xrightarrow[\delta \to 0^+]{} \mathrm{KL}\left(\overrightarrow{\mu}_T | \gamma^d\right) \,.$$

where the convergence follows from Assumption 3.3. This concludes the proof.

$\square$

### B.4.2. PROOF OF THEOREM 3.7

Before presenting the proof of Theorem 3.7, we give the following technical lemma, which provides an estimate of the log-Sobolev constant of $\overleftarrow{\mu}_t^\theta$ under a one-sided Lipschitz condition on the score network. This is a direct corollary of Malrieu (2001) (cited by Monmarché et al. (2024)).

**Lemma B.4.** *Assume that there exists $M \in \mathbb{R}$ for all $t \in [0, T]$ and $x, y \in \mathbb{R}^d$, we have*

$$\langle s^\theta(t, x) - s^\theta(t, y), x - y \rangle \leqslant M \|x - y\|^2 \,, \tag{38}$$

*then $\overleftarrow{p}_t^\theta$ satisfies the log-Sobolev inequality with constant*

$$C_t := e^{2(M-1)t} + 2\int_0^t e^{2(M-1)s}\mathrm{d}s \,.$$

*Proof.* Let $b_t(x) := s^\theta(T-t, x) - x$, so that the estimated backward density follows the Fokker-Planck equation

$$\partial_t \overleftarrow{p}_t^\theta = \nabla \cdot \left(\nabla \overleftarrow{p}^\theta - \overleftarrow{p}^\theta b_t\right) \,,$$

which is exactly (Monmarché et al., 2024, Equation (1.1)) with $\sigma = 1$. Then, we have

$$\langle b_t(t, x) - b_t(t, y), x - y \rangle = \langle s^\theta(T-t, x) - s^\theta(T-t, y), x - y \rangle - \|x - y\|^2 \leqslant (M-1)\|x - y\|^2 \,.$$

The result follows by a direct application of (Monmarché et al., 2024, Proposition 1.1), after noting that $\gamma^d$ satisfies the log-Sobolev inequality with constant 1. This concludes the proof. $\square$

We can now present the proof of Theorem 3.7.

*Proof.* (of Theorem 3.7) Let $t \in (0, T)$, by Lemma 3.4 (with $\alpha(t) = 1/2$ for all $t$), we have

$$\frac{\mathrm{d}}{\mathrm{d}t} \mathrm{KL}\left(\overleftarrow{\mu}_t | \overleftarrow{\mu}_t^\theta\right) \leqslant -\frac{1}{2}\mathscr{I}\left(\overleftarrow{\mu}_t | \overleftarrow{\mu}_t^\theta\right) + \frac{1}{2}\mathbb{E}\left[\left\|2\nabla \log \tilde{p}_{T-t}(\overrightarrow{X}_{T-t}) - s^\theta(T-t, \overrightarrow{X}_{T-t})\right\|^2\right] .$$

We first consider the case where $M_t \equiv M$ is constant.

**Case 1: $M \neq 1$.** By Lemma B.4 and our assumptions, we know that $\overleftarrow{p}_t^\theta$, for $t \in [0, T]$, satisfies the log-Sobolev inequality with constant

$$C(t) := e^{2(M-1)t} + 2\int_0^t e^{2(M-1)s}\mathrm{d}s = e^{2(M-1)t} + \frac{e^{2(M-1)t} - 1}{M - 1} .$$

It is easily noted that $C(t)$ is non-negative for all $t \geqslant 0$ (for all values of $M \in \mathbb{R}$). Therefore, by the logarithmic Sobolev inequality, we have, for $t \in (0, t)$,

$$\frac{\mathrm{d}}{\mathrm{d}t} \mathrm{KL}\left(\overleftarrow{\mu}_t | \overleftarrow{\mu}_t^\theta\right) \leqslant -\frac{1}{C(t)} \mathrm{KL}\left(\overleftarrow{\mu}_t | \overleftarrow{\mu}_t^\theta\right) + \frac{1}{2}\mathbb{E}\left[\left\|2\nabla \log \tilde{p}_{T-t}(\overrightarrow{X}_{T-t}) - s^\theta(T-t, \overrightarrow{X}_{T-t})\right\|^2\right] .$$

By Theorem 3.5 (with $\alpha(t) = 1/2$ for all $t$), we obtain

$$\mathrm{KL}\left(\mu | \overleftarrow{\mu}_T^\theta\right) \leqslant \mathrm{KL}\left(\overrightarrow{\mu}_T | \gamma^d\right) \exp\left\{-\int_0^T \frac{\mathrm{d}u}{C(u)}\right\}$$
$$+ \frac{1}{2}\int_0^T \exp\left\{-\int_t^T \frac{\mathrm{d}u}{C(u)}\right\} \mathbb{E}\left[\left\|2\nabla \log \tilde{p}_{T-t}(\overrightarrow{X}_{T-t}) - s^\theta(T-t, \overrightarrow{X}_{T-t})\right\|^2\right] \mathrm{d}t .$$

Then, we need to distinguish two cases. If $M > 1$, we have, for $t \in (0, T)$,

$$\int_t^T \frac{\mathrm{d}u}{C(u)} = \frac{1}{2}\int_t^T \frac{(M-1)e^{-2(M-1)u}}{M - e^{-2(M-1)u}}\mathrm{d}u = \frac{1}{2}\log\left(\frac{M - e^{-2(M-1)T}}{M - e^{-2(M-1)t}}\right)$$

Therefore

$$\mathrm{KL}\left(\mu | \overleftarrow{\mu}_T^\theta\right) \leqslant \eta_T(0)\mathrm{KL}\left(\overrightarrow{\mu}_T | \gamma^d\right) + \int_0^T \frac{\eta_T(t)}{2}\mathbb{E}\left[\left\|2\nabla \log \tilde{p}_{T-t}(\overrightarrow{X}_{T-t}) - s^\theta(T-t, \overrightarrow{X}_{T-t})\right\|^2\right] \mathrm{d}t ,$$

with

$$\eta_T(t) := \exp\left\{-\int_t^T \frac{\mathrm{d}u}{C(u)}\right\} = \sqrt{\frac{M - e^{-2(M-1)t}}{M - e^{-2(M-1)T}}} .$$

Now if $M < 1$, we have

$$\exp\left\{-\int_t^T \frac{\mathrm{d}u}{C(u)}\right\} = \exp\left\{-\frac{1}{2}\int_t^T \frac{(1-M)e^{2(1-M)u}}{e^{2(1-M)u} - M}\right\} = \sqrt{\frac{e^{2(1-M)t} - M}{e^{2(1-M)T} - M}} .$$

We also observe that for all $t \geqslant 0$, we have

$$C(t) \leqslant 1 + 2\int_0^{+\infty} e^{-2(1-M)u}\mathrm{d}u = 1 + \frac{1}{1 - M} .$$

Therefore, we obtain the same bound with

$$\eta_T(t) := \exp\left\{-\frac{1-M}{2-M}(T - t)\right\} .$$

**Case 2:** $M = 1$. In this case, we have $C(t) = 1 + 2t$. Therefore, we have the same bound as before, but with

$$\eta_T(t) = \exp\left\{ -\int_t^T \frac{\mathrm{d}u}{C(u)} \right\} = \sqrt{\frac{1 + 2t}{1 + 2T}} \; .$$

By defining $\lambda_T(t) := \eta_T(T - t)$, this concludes the proof of the first part of the statement. The proof of the second part of Theorem 3.7 is deferred to Section B.4.3, where it is obtained in Corollary B.7. □

### B.4.3. TIME-DEPENDENT ONE-SIDED LIPSCHITZ CONSTANTS

In this subsection, we explore the case where the one-sided Lipschitz constant of the network is allowed to depend on time. This requires to adapt the log-Sobolev estimates of (Malrieu, 2001; Monmarché et al., 2024), which we first present.

**Proposition B.5.** *Consider a time-dependent drift $b_t : \mathbb{R}^d \to \mathbb{R}^d$ and the SDE*

$$\mathrm{d}X_t = b_t(X_t)\mathrm{d}t + \sqrt{2}\mathrm{d}B_t \; , \quad X_0 \sim \nu_0 \; , \tag{39}$$

*and assume that there exists a continuous function $L : \mathbb{R}_+ \to \mathbb{R}$ which is integrable on $[0, T]$ and such that, for all $x, y \in \mathbb{R}^d$ and $t > 0$,*

$$\langle b_t(x) - b_t(y), x - y \rangle \leqslant L(t) \|x - y\|^2 \; .$$

*We further assume that $\mu_0$ satisfies the log-Sobolev inequality with constant $C_0$, then the law $\nu_t$ of $X_t$ satisfies the log-Sobolev inequality with constant given by*

$$C_t = C_0 \exp\left\{ \int_0^t 2L(s)\mathrm{d}s \right\} + 2\int_0^t \exp\left\{ \int_s^t 2L(u)\mathrm{d}u \right\} \mathrm{d}s \; .$$

For the purpose of this operator, we introduce the operators

$$P_{s,t}f(x) := \mathbb{E}\left[ f(X_t)|X_s = x \right] \; .$$

*Proof.* The proof is inspired by the proof of (Monmarché et al., 2024, Proposition 1.1), which we adapt to handle time-dependent one-sided Lipschitz constants.

Let $x, h \in \mathbb{R}^d$, $s \geqslant 0$, and consider $f \in \mathcal{C}_b^2(\mathbb{R}^d)$, *i.e.*, $f$ is twice continuously differentiable and is bounded with bounded derivatives of order 1 and 2. Let $X_t$ and $X_t'$ be two solutions of the SDE, with the same Brownian motion and initialized at time $s$ such that $X_s = x + h$ and $X_s' = x$. Our assumption gives that, for all $t \geqslant s$, we have

$$\mathrm{d}\|X_t - X_t'\|^2 \leqslant 2L(t)\|X_t - X_t'\|^2 \; .$$

Hence, by Grönwall's lemma, we have

$$\|X_t - X_t'\|^2 \leqslant \exp\left\{ \int_s^t 2L(u)\mathrm{d}u \right\} \|X_s - X_s'\|^2 \; .$$

By the Taylor-Langrange formula, we have

$$f(X_t) - f(X_t') \leqslant \langle X_t - X_t', \nabla f(X_t') \rangle + \frac{1}{2} \left\| \nabla^2 f \right\|_\infty \|X_t - X_t'\|^2 \; .$$

Now we take the expectation and use $h := \epsilon u$ with $u := \nabla P_{s,t}f(x) / \|\nabla P_{s,t}f(x)\|$. By the Cauchy-Schwarz inequality,

$$\frac{1}{\epsilon}\left( P_{s,t}f(x + \epsilon u) - P_{s,t}f(x) \right) \leqslant \exp\left\{ \int_s^t L(u)\mathrm{d}u \right\} P_{s,t}(\|\nabla f\|)(x) + \mathcal{O}\left( \epsilon \right) \; ,$$

From which we deduce, by taking $\epsilon \to 0^+$, the following commutation property,

$$\|\nabla P_{s,t}f\| \leqslant \exp\left\{ \int_s^t L(u)\mathrm{d}u \right\} P_{s,t}(\|\nabla f\|) \; . \tag{40}$$

The rest of the proof follows the classical Bakry-Émery interpolation technique, as in (Monmarché et al., 2024, Proposition 1.1). More precisely, let $0 \leqslant s \leqslant t$ and $\Phi(x) := x \log(x)$. Let us introduce $\mathcal{L}_t \varphi := \Delta \varphi + \langle b_t, \nabla \varphi \rangle$ the time-dependent generator of Equation (39). It is known to satisfy the two Kolmogorov equations (Collet & Malrieu, 2008),

$$\partial_t P_{s,t} f = P_{s,t} \mathcal{L}_t f , \quad \partial_s P_{s,t} f = -\mathcal{L}_s P_{s,t} f .$$

By the Kolmogorov equations and the diffusion property for $\mathcal{L}_t$ (Bakry et al., 2014), we have

$$\partial_s \left( P_{0,s} \Phi(P_{s,t} f) \right) = P_{0,s} \left( \mathcal{L}_t \Phi(P_{s,t} f) - \Phi'(P_{s,t} f) \mathcal{L}_s P_{s,t} f \right) = P_{0,s} \left( \frac{\|P_{s,t} f\|^2}{P_{s,t} f} \right) .$$

By the commutation property and the Cauchy-Schwarz inequality, we have

$$\partial_s \left( P_{0,s} \Phi(P_{s,t} f) \right) \leqslant \exp \left\{ \int_s^t 2L(u) \mathrm{d}u \right\} P_{0,s} \left( \frac{(P_{s,t} \|f\|)^2}{P_{s,t} f} \right) \leqslant \exp \left\{ \int_s^t 2L(u) \mathrm{d}u \right\} P_{0,t} \left( \frac{\|f\|^2}{f} \right) .$$

By integrating between $0$ and $t$, we get

$$P_{0,t} \Phi(f) - \Phi(P_{0,t} f) \leqslant \int_0^t \exp \left\{ \int_s^t 2L(u) \mathrm{d}u \right\} \mathrm{d}s \int \frac{\|f\|^2}{f} \mathrm{d}\nu_t .$$

By integrating with respect to $\nu_0$ and using the log-Sobolev inequality for $\nu_0$, we have

$$\int \Phi(f) \mathrm{d}\nu_t \leqslant \int \Phi(P_{0,t} f) \mathrm{d}\nu_0 + \int_0^t \exp \left\{ \int_s^t 2L(u) \mathrm{d}u \right\} \mathrm{d}s \int \frac{\|f\|^2}{f} \mathrm{d}\nu_t$$

$$\leqslant \Phi \left( \int f \mathrm{d}\nu_t \right) + \left( \frac{C_0}{2} + \int_0^t \exp \left\{ \int_s^t 2L(u) \mathrm{d}u \right\} \mathrm{d}s \right) \int \frac{\|f\|^2}{f} \mathrm{d}\nu_t .$$

This concludes the proof by definition of the log-Sobolev inequality in Definition 2.1. $\qquad \square$

In the next theorem, we present the associated score-matching bound.

**Theorem B.6.** *Under the conditions of Lemma 3.4, suppose that there exists $M \in \mathbb{R}$ s.t. for all $t \in [0, T]$, $x, y \in \mathbb{R}^d$, $\langle s^\theta(t, x) - s^\theta(t, y), x - y \rangle \leqslant M(t) \|x - y\|^2$ such that the map $t \mapsto M_t$ is continuous on $[0, T]$. Then,*

$$\mathrm{KL} \left( \mu | \overleftarrow{\mu}^\theta_T \right) \leqslant \int_0^T \frac{\eta_T(t)}{2} \mathbb{E}[\varepsilon^\theta_{T-t}(\overleftarrow{X}_t)^2] \mathrm{d}t + \eta_T(0) \mathrm{KL} \left( \overrightarrow{\mu}_T | \gamma^d \right) ,$$

*where the time-dependent decay factor is*

$$\eta_T(t) := \exp \left\{ -\int_t^T \left( \exp \left\{ \int_0^s 2(M(T - u) - 1) \mathrm{d}u \right\} + 2 \int_0^s \exp \left\{ \int_u^s 2(M(T - v) - 1) \mathrm{d}v \right\} \mathrm{d}u \right)^{-1} \mathrm{d}s \right\} .$$

*We refer to Remark 3.6 for further upper bounds on $\mathrm{KL} \left( \overrightarrow{\mu}_T | \gamma^d \right)$ under generic assumptions.*

*Proof.* By reasoning as in the proof of Lemma B.4 and applying Proposition B.5, we obtain that $\overleftarrow{\mu}^\theta_t$ satisfies the log-Sobolev inequality with constant

$$\gamma(t) = \exp \left\{ \int_0^t 2(M(T - s) - 1) \mathrm{d}s \right\} + 2 \int_0^t \exp \left\{ \int_s^t 2(M(T - u) - 1) \mathrm{d}u \right\} \mathrm{d}s ,$$

where we noted that the initial distribution $(\gamma^d)$ satisfies the log-Sobolev inequality with constant 1. As $\gamma$ is positive and continuous on $[0, T]$, we can apply Theorem 3.5 (with $\alpha(t) = 1/2$ for all $t$) to obtain that

$$\mathrm{KL} \left( \mu | \overleftarrow{\mu}^\theta_T \right) \leqslant \int_0^T \frac{\eta_T(t)}{2} \mathbb{E}[\varepsilon^\theta_{T-t}(\overleftarrow{X}_t)] \mathrm{d}t + e^{-2T} \eta_T(0) \mathrm{KL} \left( \mu | \gamma^d \right) ,$$

with $C_T := e^{-2T}\eta_T(0)$ and

$$\eta_T(t) := \exp\left\{-\int_t^T \frac{\mathrm{d}s}{\gamma(s)}\right\}$$

$$= \exp\left\{-\int_t^T \left(\exp\left\{\int_0^s 2(M(T-u)-1)\mathrm{d}u\right\} + 2\int_0^s \exp\left\{\int_u^s 2(M(T-v)-1)\mathrm{d}v\right\}\mathrm{d}u\right)^{-1}\mathrm{d}s\right\}.$$

This concludes the proof. $\qquad\square$

**Corollary B.7.** *Under the same assumption as Theorem B.6, suppose that we can take $M(t) = 1 + c/t$, then, the result of Theorem B.6 holds with*

$$\eta_T(t) := \sqrt{1 - \frac{2(T-t)^{2c+1}}{(2c+1)T^{2c} + 2T^{2c+1}}}.$$

*Proof.* By Theorem B.6, we have

$$\eta_T(t) = \exp\left\{-\int_t^T \left(\exp\left\{\int_0^s 2(M(T-u)-1)\mathrm{d}u\right\} + 2\int_0^s \exp\left\{\int_u^s 2(M(T-v)-1)\mathrm{d}v\right\}\mathrm{d}u\right)^{-1}\mathrm{d}s\right\}$$

$$= \exp\left\{-\int_t^T \left(\exp\left\{\int_0^s \frac{2c}{T-u}\mathrm{d}u\right\} + 2\int_0^s \exp\left\{\int_u^s \frac{2c}{T-v}\mathrm{d}v\right\}\mathrm{d}u\right)^{-1}\mathrm{d}s\right\}$$

$$= \exp\left\{-\int_t^T \left(\left(\frac{T}{T-s}\right)^{2c} + 2\int_0^s \left(\frac{T-u}{T-s}\right)^{2c}\mathrm{d}u\right)^{-1}\mathrm{d}s\right\}$$

$$= \exp\left\{-\int_t^T \left(\left(\frac{T}{T-s}\right)^{2c} + \frac{2}{(2c+1)(T-s)^{2c}}\left(T^{2c+1} - (T-s)^{2c+1}\right)\right)^{-1}\mathrm{d}s\right\}$$

$$= \exp\left\{-\int_t^T \frac{(2c+1)(T-s)^{2c+1}}{(2c+1)T^{2c} + 2T^{2c+1} - 2(T-s)^{2c+1}}\mathrm{d}s\right\}$$

$$= \exp\left\{-\frac{1}{2}\left[\log\left((2c+1)T^{2c} + 2T^{2c+1} - 2(T-s)^{2c+1}\right)\right]_{s=t}^T\right\}$$

$$= \sqrt{\frac{(2c+1)T^{2c} + 2(T^{2c+1} - (T-t)^{2c+1})}{(2c+1)T^{2c} + 2T^{2c+1}}}$$

$$= \sqrt{1 - \frac{2(T-t)^{2c+1}}{(2c+1)T^{2c} + 2T^{2c+1}}}.$$

This concludes the proof. $\qquad\square$

The above corrolary also concludes the proof of Theorem 3.7 by setting $\lambda_T(t) := \eta_T(T-t)$.

### B.5. Proof of Theorem 3.10

*Proof.* (of Theorem 3.10) Let $t > 0$, by Lemma 3.4 (with $\alpha(t) = 1/2$ for all $t$), we have

$$\frac{\mathrm{d}}{\mathrm{d}t}\mathrm{KL}\left(\overleftarrow{\mu}_t | \overleftarrow{\mu}_t^\theta\right) \leqslant -\frac{1}{2}\mathscr{I}\left(\overleftarrow{\mu}_t | \overleftarrow{\mu}_t^\theta\right) + \frac{1}{2}\mathbb{E}\left[\left\|2\nabla\log\tilde{p}_{T-t}(\overrightarrow{X}_{T-t}) - s^\theta(T-t, \overrightarrow{X}_{T-t})\right\|^2\right].$$

The backward dynamics $(\overleftarrow{\mu}_t^\theta)_{t\in[0,T]}$ is initialized from the Gaussian distribution $\gamma^d$. Therefore, we can apply (Monmarché et al., 2024, Theorem 1.3) (the assumptions of Theorem 3.10 are exactly the assumptions that required to apply this result) to

see that there exists a constant $C_{\mu,\theta,d} > 0$ such that, for all $t \in [0,T]$, the probability distribution $\overleftarrow{\mu}_t^\theta$ satisfies a log-Sobolev inequality with constant $C_{\mu,\theta,d}$. Therefore, by the log-Sobolev inequality, we have that

$$\frac{\mathrm{d}}{\mathrm{d}t}\mathrm{KL}\left(\overleftarrow{\mu}_t|\overleftarrow{\mu}_t^\theta\right) \leqslant -\frac{1}{C_{\mu,\theta,d}}\mathrm{KL}\left(\overleftarrow{\mu}_t|\overleftarrow{\mu}_t^\theta\right) + \frac{1}{2}\mathbb{E}\left[\left\|2\nabla\log\tilde{p}_{T-t}(\overrightarrow{X}_{T-t}) - s^\theta(T-t,\overrightarrow{X}_{T-t})\right\|^2\right].$$

By Grönwall's lemma (and reasoning as in the proof of Theorem 4.3 to take limits), we have

$$\mathrm{KL}\left(\mu|\overleftarrow{\mu}_T^\theta\right) \leqslant e^{-\frac{T}{C_{\mu,\theta,d}}}\mathrm{KL}\left(\overrightarrow{\mu}_T|\gamma^d\right) + \frac{1}{2}\int_0^T e^{-\frac{T-t}{C_{\mu,\theta,d}}}\mathbb{E}\left[\left\|2\nabla\log\tilde{p}_{T-t}(\overrightarrow{X}_{T-t}) - s^\theta(T-t,\overrightarrow{X}_{T-t})\right\|^2\right]\mathrm{d}t.$$

Finally, we use the decay of relative entropy along the Ornstein-Uhlenbeck semigroup (Bakry et al., 2014) to get that

$$\mathrm{KL}\left(\mu|\overleftarrow{\mu}_T^\theta\right) \leqslant e^{-\frac{T}{C_{\mu,\theta,d}}}\mathrm{KL}\left(\overrightarrow{\mu}_T|\gamma^d\right) + \frac{1}{2}\int_0^T e^{-\frac{T-t}{C_{\mu,\theta,d}}}\mathbb{E}\left[\left\|2\nabla\log\tilde{p}_{T-t}(\overrightarrow{X}_{T-t}) - s^\theta(T-t,\overrightarrow{X}_{T-t})\right\|^2\right]\mathrm{d}t.$$

This concludes the proof. $\qquad\square$

## B.6. Proofs of Section 4.1

The proof of Lemma 4.2 is formally similar to the proof of Lemma 3.4, as explained below.

We give below the proof of Lemma 4.2.

*Proof.* (of Lemma 4.2) Our assumptions ensure that the quantities appearing below are finite. Let $\Phi(u) := u\log(u)$ and define

$$\bar{v}_t := \frac{\overleftarrow{p}_t^\theta}{\overleftarrow{p}_t}.$$

For $t \in (0,T)$, by the same computations as in the proof of Lemma 3.4 switching the roles of $v_t$ and $\bar{v}_t$, we have (noting that $\Phi''(u) = 1/u$)

$$\frac{\mathrm{d}}{\mathrm{d}t}\mathrm{KL}\left(\overleftarrow{\mu}_t^\theta|\overleftarrow{\mu}_t\right) = -\mathscr{I}\left(\overleftarrow{\mu}_t^\theta|\overleftarrow{\mu}_t\right) - \int\langle\nabla\bar{v}_t(x), b_{T-t}(x) - b_{T-t}^\theta(x)\rangle\overleftarrow{p}_t\mathrm{d}x.$$

Therefore, by Young's inequality, we have

$$w\frac{\mathrm{d}}{\mathrm{d}t}\mathrm{KL}\left(\overleftarrow{\mu}_t^\theta|\overleftarrow{\mu}_t\right) \leqslant -\frac{1}{2}\mathscr{I}\left(\overleftarrow{\mu}_t^\theta|\overleftarrow{\mu}_t\right) + \frac{1}{2}\int\left\|2\nabla\log\tilde{p}_{T-t}(x) - s^\theta(T-t,x)\right\|^2\overleftarrow{p}_t(x)\mathrm{d}x.$$

This concludes the proof. $\qquad\square$

We now present the following simple proposition, which is an estimate of the time-dependent log-Sobolev constant along the Ornstein-Uhlenbeck semigroup.

**Proposition B.8.** *Assume that $\mu$ satisfies the log-Sobolev inequality with constant $\rho_0$. Then $\overrightarrow{\mu}_t$ satisfies the log-Sobolev inequality with constant $e^{-2t}\rho_0 + (1 - e^{-2t})$, for all $t \geqslant 0$. As a consequence, $\overleftarrow{\mu}_t$ satisfies the log-Sobolev inequality with constant $e^{-2(T-t)}\rho_0 + (1 - e^{-2(T-t)})$.*

*Proof.* We note that

$$\overrightarrow{\mu}_t = \mathrm{Law}\left(e^{-t}Z + \sqrt{1 - e^{-2t}}\Xi\right),$$

where $Z \sim \mu$, $\Xi \sim \gamma^d$, and $Z$ and $\Xi$ are independent. Then, the results follows immediately from Lemmas A.1 and A.2. $\qquad\square$

### B.6.1. PROOF OF THEOREM 4.3

*Proof.* (of Theorem 4.3) Let us denote $\rho_t := e^{-2t}\rho_0 + 1 - e^{-2t}$ for all $t \geqslant 0$. We apply Lemma 4.2, it gives

$$\frac{\mathrm{d}}{\mathrm{d}t} \mathrm{KL}\left(\overleftarrow{\mu}_t^\theta | \overleftarrow{\mu}_t\right) \leqslant -\frac{1}{2}\mathscr{I}\left(\overleftarrow{\mu}_t^\theta | \overleftarrow{\mu}_t\right) + \frac{1}{2}\mathbb{E}\left[\left\|2\nabla\log\tilde{p}_{T-t}(\overleftarrow{X}_t^\theta) - s^\theta(T-t, \overleftarrow{X}_t^\theta)\right\|^2\right].$$

By Proposition B.8, we know that $\overleftarrow{\mu}_t$ satisfies the log-Sobolev inequality with constant $\rho_{T-t}$. Therefore, we have

$$\frac{\mathrm{d}}{\mathrm{d}t} \mathrm{KL}\left(\overleftarrow{\mu}_t^\theta | \overleftarrow{\mu}_t\right) \leqslant -\frac{1}{\rho_{T-t}}\mathrm{KL}\left(\overleftarrow{\mu}_t^\theta | \overleftarrow{\mu}_t\right) + \frac{1}{2}\mathbb{E}\left[\left\|2\nabla\log\tilde{p}_{T-t}(\overleftarrow{X}_t^\theta) - s^\theta(T-t, \overleftarrow{X}_t^\theta)\right\|^2\right].$$

By Grönwall's lemma (Gronwall, 1919), we obtain that

$$\mathrm{KL}\left(\overleftarrow{\mu}_{T-\epsilon}^\theta | \overrightarrow{\mu}_\epsilon\right) \leqslant \exp\left\{-\int_\delta^{T-\epsilon}\frac{\mathrm{d}s}{\rho_{T-s}}\right\}\mathrm{KL}\left(\overleftarrow{\mu}_\delta^\theta | \overleftarrow{\mu}_\delta\right)$$
$$+ \frac{1}{2}\int_\delta^{T-\epsilon}\exp\left\{-\int_t^{T-\epsilon}\frac{\mathrm{d}s}{\rho_{T-s}}\right\}\mathbb{E}\left[\left\|2\nabla\log\tilde{p}_{T-t}(\overleftarrow{X}_t^\theta) - s^\theta(T-t, \overleftarrow{X}_t^\theta)\right\|^2\right]\mathrm{d}t.$$

We know that the $t \mapsto \overrightarrow{X}_t$ is almost-surely continuous at $t = 0$ and that $t \mapsto \overleftarrow{X}_t^\theta$ is almost surely continuous at $t = T$. As almost sure convergence implies convergence in distribution, this implies the convergences in distribution $\overleftarrow{\mu}_t \rightharpoonup \mu$ and $\overleftarrow{\mu}_{T-t}^\theta \rightharpoonup \overleftarrow{\mu}_T^\theta$ as $t \to 0^+$. By the joint lower semi-continuity of relative entropy (van Erven & Harremoës, 2014) and up to taking a sequence $\epsilon_n \to 0$, we can take the inferior limit as $\epsilon \to 0^+$ to get that

$$\mathrm{KL}\left(\overleftarrow{\mu}_T^\theta | \mu\right) \leqslant \exp\left\{-\int_\delta^T\frac{\mathrm{d}s}{\rho_{T-s}}\right\}\mathrm{KL}\left(\overleftarrow{\mu}_\delta^\theta | \overleftarrow{\mu}_\delta\right)$$
$$+ \frac{1}{2}\int_\delta^T\exp\left\{-\int_t^T\frac{\mathrm{d}s}{\rho_{T-s}}\right\}\mathbb{E}\left[\left\|2\nabla\log\tilde{p}_{T-t}(\overleftarrow{X}_t^\theta) - s^\theta(T-t, \overleftarrow{X}_t^\theta)\right\|^2\right]\mathrm{d}t.$$

Now we take the superior limit as $\delta \to 0^+$ to obtain that

$$\mathrm{KL}\left(\overleftarrow{\mu}_T^\theta | \mu\right) \leqslant \exp\left\{-\int_0^T\frac{\mathrm{d}s}{\rho_{T-s}}\right\}\limsup_{\delta\to0^+}\mathrm{KL}\left(\overleftarrow{\mu}_\delta^\theta | \overleftarrow{\mu}_\delta\right)$$
$$+ \frac{1}{2}\int_0^T\exp\left\{-\int_t^T\frac{\mathrm{d}s}{\rho_{T-s}}\right\}\mathbb{E}\left[\left\|2\nabla\log\tilde{p}_{T-t}(\overleftarrow{X}_t^\theta) - s^\theta(T-t, \overleftarrow{X}_t^\theta)\right\|^2\right]\mathrm{d}t.$$

The rest of the proof is formally similar to the end of the proof of Theorem 3.5. Let us sketch it for completeness.

Thanks to our assumptions, we can apply (Bogachev et al., 2016, Theorem 1.1 and Remark 1.3) along with the data processing inequality, giving us that

$$\mathrm{KL}\left(\overleftarrow{\mu}_\delta^\theta | \overleftarrow{\mu}_\delta\right) \leqslant \mathrm{KL}\left(\gamma^d | \overrightarrow{\mu}_T\right) + \int_0^\delta\mathbb{E}\left[\left\|2\nabla\log\tilde{p}_{T-t}(\overleftarrow{X}_t^\theta) - s^\theta(T-t, \overleftarrow{X}_t^\theta)\right\|^2\right]\mathrm{d}t$$
$$\xrightarrow[\delta\to0^+]{} \mathrm{KL}\left(\gamma^d | \overrightarrow{\mu}_T\right),$$

where the convergence follows from the dominated convergence theorem and Assumption 4.1.

Finally, we note that, for all $t \in [0, T]$, we have

$$\bar{\eta}_T(t) := \exp\left\{-\int_t^T\frac{\mathrm{d}s}{\rho_{T-s}}\right\} = \sqrt{\frac{\rho_0}{\rho_0 + e^{2(T-t)} - 1}}.$$

This concludes the proof by setting $\bar{\lambda}(t) := \bar{\eta}_T(T - t)$. □

### B.6.2. PROOF OF COROLLARY 4.6

*Proof.* (of Corollary 4.6) The beginning of the proof follows the same line as the proof of Theorem 4.3 until the inequality,

$$\frac{\mathrm{d}}{\mathrm{d}t}\mathrm{KL}\left(\overleftarrow{\mu}_t^\theta|\overleftarrow{\mu}_t\right) \leqslant -\frac{1}{\rho_{T-t}}\mathrm{KL}\left(\overleftarrow{\mu}_t^\theta|\overleftarrow{\mu}_t\right) + \frac{1}{2}\mathbb{E}\left[\left\|2\nabla\log\tilde{p}_{T-t}(\overleftarrow{X}_t^\theta) - s^\theta(T-t,\overleftarrow{X}_t^\theta)\right\|^2\right],$$

with $\bar{\rho}_t := e^{-2t}\rho_0 + 1 - e^{-2t}$. We now apply Donsker-Varadhan's formula on the second term to obtain that, for any $C > 0$, we have (note that the statement is true even when the right-hand side is infinite),

$$\frac{\mathrm{d}}{\mathrm{d}t}\mathrm{KL}\left(\overleftarrow{\mu}_t^\theta|\overleftarrow{\mu}_t\right) \leqslant \left(-\frac{1}{\rho_{T-t}} + \frac{1}{C}\right)\mathrm{KL}\left(\overleftarrow{\mu}_t^\theta|\overleftarrow{\mu}_t\right) + \frac{1}{C}\log\mathbb{E}\left[e^{\frac{C}{2}\left\|2\nabla\log\tilde{p}_{T-t}(\overleftarrow{X}_t) - s^\theta(T-t,\overleftarrow{X}_t)\right\|^2}\right].$$

By choosing $C := 2\rho_{T-t}$, we obtain

$$\frac{\mathrm{d}}{\mathrm{d}t}\mathrm{KL}\left(\overleftarrow{\mu}_t^\theta|\overleftarrow{\mu}_t\right) \leqslant -\frac{1}{2\rho_{T-t}}\mathrm{KL}\left(\overleftarrow{\mu}_t^\theta|\overleftarrow{\mu}_t\right) + \frac{1}{2\rho_{T-t}}\log\mathbb{E}\left[e^{\rho_{T-t}\left\|2\nabla\log\tilde{p}_{T-t}(\overleftarrow{X}_t) - s^\theta(T-t,\overleftarrow{X}_t)\right\|^2}\right].$$

By Grönwall's lemma and the same arguments as in the end of Theorem 4.3, we obtain

$$\mathrm{KL}\left(\overleftarrow{\mu}_T^\theta|\mu\right) \leqslant \tilde{\eta}_T(0)\mathrm{KL}\left(\gamma^d|\overrightarrow{\mu}_T\right) + \int_0^T \frac{\tilde{\eta}_T(t)}{2\rho_{T-t}}\log\mathbb{E}\left[e^{\rho_{T-t}\left\|2\nabla\log\tilde{p}_{T-t}(\overleftarrow{X}_t) - s^\theta(T-t,\overleftarrow{X}_t)\right\|^2}\right]\mathrm{d}t,$$

with

$$\tilde{\eta}_T(t) := \exp\left\{-\int_t^T \frac{\mathrm{d}s}{2\rho_{T-s}}\right\} = \left(\frac{\rho_0}{\rho_0 + e^{2(T-t)} - 1}\right)^{\frac{1}{4}}.$$

This concludes the proof by setting $\tilde{\lambda}_T(t) := \tilde{\eta}_T(T-t)$. $\qquad\square$

## C. Omitted proofs of Section 4.2

In this section, we present the proofs of Theorem 4.8 and Corollary 4.9, which we split in several lemmas and propositions.

Let $b_t(x) = s(t, x) - x$ and $b_t^\theta(x) = s^\theta(t, x) - x$. Since we wish to derive contractions in the case where the bias terms are not equal, we must make a technical but important modification to the proof. In particular, the handling of what happens when the processes collide. The point of collision is the precise point where the reflection coupling is not well-defined and the application of Itô's lemma in Euclidean norm is not valid. In the case of (Eberle, 2016), the authors handle this using an optional stopping time, as well as the fact that the processes stick together after collision.

In the present case, we deal with it differently, borrowing the methodology of (Durmus et al., 2020). We couple the processes using a reflection coupling at distance and with a synchronous coupling when close. We define the coupling as follows:

$$\mathrm{d}\overleftarrow{X}_t = b_{T-t}(\overleftarrow{X}_t)\mathrm{d}t + \sqrt{2}\phi_r^\delta(Z_t)\mathrm{d}\mathrm{B}_t + \sqrt{2}\phi_s^\delta(Z_t)\mathrm{d}\tilde{\mathrm{B}}_t, \tag{41}$$

$$\mathrm{d}\overleftarrow{X}_t^\theta = b_{T-t}^\theta(\overleftarrow{X}_t^\theta)\mathrm{d}t + \sqrt{2}\phi_r^\delta(Z_t)(I - 2e_t e_t^T)\mathrm{d}\mathrm{B}_t + \sqrt{2}\phi_s^\delta(Z_t)\mathrm{d}\tilde{\mathrm{B}}_t, \tag{42}$$

$$Z_t = \overleftarrow{X}_t - \overleftarrow{X}_t^\theta,$$

where $e_t = (\overleftarrow{X}_t - \overleftarrow{X}_t^\theta)/\|\overleftarrow{X}_t - \overleftarrow{X}_t^\theta\|$ and for any $\delta > 0$, we define the functions $\phi_r^\delta : \mathbb{R}^d \to \mathbb{R}_+$ and $\phi_s^\delta : \mathbb{R}^d \to \mathbb{R}_+$ as any Lipschitz continuous functions satisfying,

$$\phi_r^\delta(z)^2 + \phi_s^\delta(z)^2 = 1, \qquad \phi_r^\delta(z) = \begin{cases} 1, & \|z\| \geqslant \delta, \\ 0, & \|z\| \leqslant \delta/2. \end{cases}$$

**Lemma C.1.** *Suppose that*

$$\frac{\langle Z_t, b_{T-t}(\overleftarrow{X}_t) - b_{T-t}^\theta(\overleftarrow{X}_t^\theta)\rangle}{\|Z_t\|}$$

*is integrable in $[0,T]$ almost surely, then we have that for all $t \in [0,T]$*

$$d\|Z_t\| = \mathbb{1}_{\|Z_t\|>0} \frac{\langle Z_t, b_{T-t}(\overleftarrow{X}_t) - b^\theta_{T-t}(\overleftarrow{X}^\theta_t)\rangle}{\|Z_t\|} dt + 2\sqrt{2}\phi^\delta_r(Z_t)dB_t \ ,$$

*almost surely.*

*Proof.* By Itô's lemma, we have that,

$$d\|Z_t\|^2 = 2\langle Z_t, b_{T-t}(\overleftarrow{X}_t) - b^\theta_{T-t}(\overleftarrow{X}^\theta_t)\rangle dt + 8\phi^\delta_r(Z_t)^2 dt + 4\sqrt{2}\phi^\delta_r(Z_t)\langle Z_t, e_t\rangle e^T_t dB_t \ .$$

For any $a > 0$, define the function $\psi_a(r) = (r+a)^{1/2}$ which is twice differentiable for all $r \geqslant 0$. Then, by Itô's lemma, we have that,

$$\begin{aligned} d\psi_a(\|Z_t\|^2) &= 2\psi'_a(\|Z_t\|^2)\langle Z_t, b_{T-t}(\overleftarrow{X}_t) - b^\theta_{T-t}(\overleftarrow{X}^\theta_t)\rangle dt \\ &\quad + \phi^\delta_r(Z_t)^2(8\psi'_a(\|Z_t\|^2) + 16\psi''_a(\|Z_t\|^2)\|Z_t\|^2)dt \\ &\quad + 4\sqrt{2}\psi'_a(\|Z_t\|^2)\phi^\delta_r(Z_t)\langle Z_t, e_t\rangle e^T_t dB_t \ . \end{aligned}$$

Given that $\psi'_a(r) = \frac{1}{2}(r+a)^{-1/2}$, we have that $r\psi'_a(r^2) \to \frac{1}{2}\mathbb{1}_{r\neq 0}$ pointwise as $a \to 0^+$. Since we also have that $r\psi'_a(r^2) \leqslant 1/2$ for all $a > 0, r \geqslant 0$, we can apply dominated convergence to obtain that,

$$\lim_{a\to 0}\int_0^T 2\psi'_a(\|Z_t\|^2)\langle Z_t, b_{T-t}(\overleftarrow{X}_t) - b^\theta_{T-t}(\overleftarrow{X}^\theta_t)\rangle dt = \int_0^T \mathbb{1}_{\|Z_t\|>0}\frac{\langle Z_t, b_{T-t}(\overleftarrow{X}_t) - b^\theta_{T-t}(\overleftarrow{X}^\theta_t)\rangle}{\|Z_t\|} dt \ .$$

The Brownian motion term is controlled similarly using the stochastic counterpart of the dominated convergence theorem (Revuz & Yor, 1999, Theorem 2.12, Chapter 4) to obtain,

$$\lim_{a\to 0}\int_0^T 4\sqrt{2}\psi'_a(\|Z_t\|^2)\phi^\delta_r(Z_t)\langle Z_t, e_t\rangle e^T_t dB_t = 2\sqrt{2}\int_0^T \mathbb{1}_{\|Z_t\|>0}\phi^\delta_r(Z_t)e^T_t dB_t \ .$$

Finally, since $8\psi'_a(r^2) + 16\psi''_a(r^2)r^2 = 4a/(r^2+a)^{3/2}$ and $\phi^\delta_r(Z_t)^2 = 0$ when $\|Z_t\| \leqslant \delta/2$, we can once again apply dominated convergence to obtain that,

$$\lim_{a\to 0}\int_0^T \phi^\delta_r(Z_t)^2(8\psi'_a(\|Z_t\|^2) + 16\psi''_a(\|Z_t\|^2)\|Z_t\|^2)dt = 0 \ .$$

This concludes the proof. $\qquad\square$

The following proposition completes the proof of Theorem 4.8.

**Proposition C.2.** *Suppose that there is a continuously differentiable non-increasing function, $L_t : [0,T] \to \mathbb{R}_+$, so that*

$$\frac{\langle x - y, b^\theta_t(x) - b^\theta_t(y)\rangle}{\|x-y\|^2} \leqslant \begin{cases} L_t, & \|x-y\| \leqslant R \ , \\ -K, & \|x-y\| \geqslant R \ , \end{cases}$$

*and that, almost surely,*

$$\int_0^T \mathbb{E}\left[\|s(t, \overrightarrow{X}_t) - s^\theta(t, \overrightarrow{X}_t)\|\right] dt < \infty \ .$$

*Then, with $f := f_{R,L_0,K}$, we have that,*

$$\mathscr{W}_f(\mu, \overleftarrow{\mu}^\theta_T) \leqslant \int_0^T C_t\|s(t, \overrightarrow{X}_t) - s^\theta(t, \overrightarrow{X}_t)\|dt + \exp\left(-\int_0^T c_t dt - T\right)\mathscr{W}_1(\mu, \gamma^d) \ ,$$

*where $c_t := c_{R,L_t,K}$ and,*

$$C_t := \exp\left(-\int_0^t c_s ds\right) \ .$$

*Proof.* We begin from Lemma C.1 coupling the processes $\overleftarrow{X}_t$ and $\overleftarrow{X}_t^\theta$ according to Equation (41) while also requiring that $\overleftarrow{X}_0$ and $\overleftarrow{X}_0^\theta$ take the optimal Kantorovich coupling. Using the notation $r_t = \|\overleftarrow{X}_t - \overleftarrow{X}_t^\theta\|$, we obtain from Lemma C.1 that,

$$dr_t = \mathbb{1}_{r_t>0} \frac{\langle \overleftarrow{X}_t - \overleftarrow{X}_t^\theta, b_{T-t}(\overleftarrow{X}_t) - b_{T-t}^\theta(\overleftarrow{X}_t^\theta)\rangle}{\left\|\overleftarrow{X}_t - \overleftarrow{X}_t^\theta\right\|}dt + 2\sqrt{2}\phi_r^\delta(Z_t)dB_t,$$

for some 1-dimensional Brownian motion $B_t$. Setting $f_t := f_{R,L_{T-t},K}$, since $f_t$ is twice-differentiable in both $t$ and $r$, it follows from Itô's lemma that,

$$df_t(r_t) = \frac{\partial f_t}{\partial t}(r_t)dt + \frac{\partial f_t}{\partial r}(r_t)\mathbb{1}_{r_t>0}\frac{\langle \overleftarrow{X}_t - \overleftarrow{X}_t^\theta, b_{T-t}(\overleftarrow{X}_t) - b_{T-t}^\theta(\overleftarrow{X}_t^\theta)\rangle}{\left\|\overleftarrow{X}_t - \overleftarrow{X}_t^\theta\right\|}dt + 4\phi_r^\delta(Z_t)^2\frac{\partial^2 f_t}{\partial r^2}(r_t)dt$$

$$+ 2\sqrt{2}\frac{\partial f_t}{\partial r}(r_t)\phi_r^\delta(Z_t)dB_t$$

$$= \frac{\partial f_t}{\partial t}(r_t)dt + \mathbb{1}_{r_t>0}\beta_t dt + \mathbb{1}_{r_t>0}\frac{\partial f_t}{\partial r}(r_t)\Delta_t^\theta dt - 4\mathbb{1}_{r_t>0}\phi_s^\delta(Z_t)^2\frac{\partial^2 f_t}{\partial r^2}(r_t)dt + 2\sqrt{2}\frac{\partial f_t}{\partial r}(r_t)\phi_r^\delta(Z_t)dB_t,$$

where we define the following terms:

$$\beta_t = \frac{\partial f_t}{\partial r}(r_t)\frac{\langle \overleftarrow{X}_t - \overleftarrow{X}_t^\theta, b_{T-t}^\theta(\overleftarrow{X}_t) - b_{T-t}^\theta(\overleftarrow{X}_t^\theta)\rangle}{\left\|\overleftarrow{X}_t - \overleftarrow{X}_t^\theta\right\|} + 4\frac{\partial^2 f_t}{\partial r^2}(r_t), \quad \Delta_t^\theta = \frac{\langle \overleftarrow{X}_t - \overleftarrow{X}_t^\theta, b_{T-t}(\overleftarrow{X}_t) - b_{T-t}^\theta(\overleftarrow{X}_t)\rangle}{\left\|\overleftarrow{X}_t - \overleftarrow{X}_t^\theta\right\|}.$$

Setting $\kappa_t = \kappa_{R,L_{T-t},K}$, it follows by construction that,

$$\beta_t \leqslant 4\frac{\partial^2 f_t}{\partial r^2}(r_t) - r_t\kappa_t(r_t)\frac{\partial f_t}{\partial r}(r_t)$$

$$\leqslant -c_{T-t}f_t(r_t).$$

Furthermore, using the Cauchy-Schwarz inequality, we have that,

$$\Delta_t^\theta \leqslant \frac{\partial f_t}{\partial r}(r_t)\left\|s(T-t, \overleftarrow{X}_t) - s^\theta(T-t, \overleftarrow{X}_t)\right\|$$

$$\leqslant \left\|s(T-t, \overleftarrow{X}_t) - s^\theta(T-t, \overleftarrow{X}_t)\right\|.$$

Furthermore, we can control the nuisance term from the modified coupling by noting that,

$$-4\mathbb{1}_{r_t>0}\phi_s^\delta(Z_t)^2\frac{\partial^2 f_t}{\partial r^2}(r_t) \leqslant 4\mathbb{1}_{r_t\in(0,\delta)}\left|\frac{\partial^2 f_t}{\partial r^2}(r_t)\right|.$$

Using Lemma A.6, we have that whenever $\delta \leqslant \sup_{t\in[0,T]}\delta_{R,L_t,K} > 0$, for any $t \in [0,T]$,

$$-4\mathbb{1}_{r_t>0}\phi_s^\delta(Z_t)^2\frac{\partial^2 f_t}{\partial r^2}(r_t) \leqslant C\delta,$$

where $C = \sup_{t\in[0,T]} C_{R,L_t,K}$. We also use Lemma A.5 to obtain that $f_t$ is everywhere non-increasing in $t$. Therefore, we have that,

$$\frac{d}{dt}\mathbb{E}\left[f(r_t)\right] \leqslant -c_{T-t}\mathbb{E}\left[f(r_t)\right] + \mathbb{E}\left[\left\|s(T-t, \overleftarrow{X}_t) - s^\theta(T-t, \overleftarrow{X}_t)\right\|\right] + C\delta.$$

Thus, from Grönwall's lemma, follows the bound,

$$\mathbb{E}\left[f_T(r_T)\right] \leqslant \int_0^T \mathbb{E}\left[\left\|s(T-t, \overleftarrow{X}_t) - s^\theta(T-t, \overleftarrow{X}_t)\right\|\right]\exp\left(-\int_t^T c_{T-u}du\right)dt + C\delta T$$

$$+ \mathbb{E}\left[f_t(r_0)\right]\exp\left(-\int_0^t c_{T-s}ds\right).$$

By a change of variables in the time integral, by definition of $\mathscr{W}_f$, and by taking $\delta \to 0^+$, we can simplify this to get that

$$\mathscr{W}_f(\mu, \overleftarrow{\mu}_T^\theta) \leq \int_0^T \mathbb{E}\left[\|s(t, \overrightarrow{X}_t) - s^\theta(t, \overrightarrow{X}_t)\|\right] \exp\left(-\int_0^t c_s \mathrm{d}s\right) \mathrm{d}t$$
$$+ \mathbb{E}[f_0(r_0)] \exp\left(-\int_0^T c_t \mathrm{d}t\right) .$$

We conclude the proof by noting that, $f_0(r_0) \leq r_0$ (by Lemma A.7), and hence, due to the choice of coupling, $\mathbb{E}[f_0(r_0)] \leq \mathscr{W}_1(\overrightarrow{\mu}_T, \gamma^d)$. By a standard mean-square argument, we obtain the contraction,

$$\mathscr{W}_1(\overrightarrow{\mu}_T, \gamma^d) \leq \exp(-T)\mathscr{W}_1(\mu, \gamma^d) .$$

This completes the proof. $\qquad\square$

The next corollary is a bound with respect to the Wasserstein's distance $\mathscr{W}_1$, that we can deduce from Lemma A.7. It completes the proof of Corollary 4.9.

**Corollary C.3.** *Under the same conditions as Proposition C.2, with $f := f_{R,L_0,K}$, we have that,*

$$\mathscr{W}_1(\mu, \overleftarrow{\mu}_T^\theta) \leq 2e^{\frac{L_0 R_0^2}{8}} \int_0^T C_t \|s(t, \overrightarrow{X}_t) - s^\theta(t, \overrightarrow{X}_t)\| \mathrm{d}t + \exp\left(-\int_0^T c_t \mathrm{d}t - T\right) \mathscr{W}_1(\mu, \gamma^d) ,$$

*where $c_t := c_{R_t, L_t, K_t}$ and,*

$$C_t := \exp\left(-\int_0^t c_s \mathrm{d}s\right) .$$

*Proof.* By Lemma A.7, we have that

$$\mathscr{W}_1 \leq 2e^{\frac{L_0 R_0^2}{8}} \mathscr{W}_f .$$

Then, the statement immediately follows from Proposition C.2. $\qquad\square$

The argument is completely symmetric, but for consistency with previous sections, we state the result for the forward case.

**Proposition C.4.** *Suppose that there is a continuous non-increasing function, $L_t : [0,T] \to \mathbb{R}_+$, so that*

$$\frac{\langle x - y, b_t(x) - b_t(y)\rangle}{\|x - y\|^2} \leq \begin{cases} L_t, & \|x - y\| \leq R_t, \\ -K_t, & \|x - y\| \geq R_t, \end{cases}$$

*and that, almost surely,*

$$\int_0^T \|s(t, \overleftarrow{X}_{T-t}^\theta) - s^\theta(t, \overleftarrow{X}_{T-t}^\theta)\| \mathrm{d}t < \infty.$$

*Then, with $f := f_{R,L_0,K}$, we have that,*

$$\mathscr{W}_f(\mu, \overleftarrow{\mu}_T^\theta) \leq \int_0^T C_t \mathbb{E}\left[\|s(t, \overleftarrow{X}_{T-t}^\theta) - s^\theta(t, \overleftarrow{X}_{T-t}^\theta)\|\right] \mathrm{d}t + \exp\left(-\int_0^T c_t \mathrm{d}t - T\right) \mathscr{W}_1(\mu, \gamma^d),$$

*where $c_t := c_{R,L_t,K}$ and,*

$$C_t := \exp\left(-\int_0^t c_s \mathrm{d}s\right) .$$

## D. Additional Results: Time-Uniform Generalization bounds for Diffusion Models

Recently, Dupuis et al. (2025); Farghly et al. (2026) proposed an algorithm- and data-dependent analysis of $\mathbb{E}[\varepsilon_t^\theta(\overleftarrow{X}_t)]$ in (1). We briefly recall their setup here. DMs are trained to minimize over $[0, T]$ the denoising score matching loss defined for $z \in \mathbb{R}^d$ and $t \in [0, T]$ as

$$\ell_t(\theta, z) := \mathbb{E}\left[\left\|s^\theta(t, \overrightarrow{X}_t^z) - 2\nabla \log \tilde{p}_{t|0}(\overrightarrow{X}_t^z | z)\right\|^2\right] .$$

In practice, as $\mu$ is unknown, we use a finite dataset $\mathbf{Z}^{(n)} := (Z_1, \ldots, Z_n) \sim \mu^{\otimes n}$ sampled from $\mu$. Then, what is minimized in practice by the learning algorithm (e.g., SGD, ADAM, ...) is the *empirical denoising score matching loss*,

$$\mathscr{L}_{\mathrm{DSM}}^{(n)}(\theta, t) := \frac{1}{n} \sum_{1 \leqslant i \leqslant n} \ell_t(\theta, Z_i)$$

These authors define the *generalization error* as the difference between $\mathscr{L}_{\mathrm{DSM}}^{(n)}$ and its population version, *i.e.*,

$$\mathscr{G}^{(n)}(\mathbf{Z}^{(n)}, \theta, t) := \int \ell_t(\theta, z)\mathrm{d}\mu(z) - \mathscr{L}_{\mathrm{DSM}}^{(n)}(\theta, t) , \quad \theta \in \Theta .$$

Let $\widehat{\mu}_n := n^{-1} \sum_{1 \leqslant i \leqslant n} \delta_{Z_i}$ be the empirical data distribution and $\overrightarrow{X}_t^{(n)}$ the process given by Equation (3) initialized at $\overrightarrow{X}_0 \sim \widehat{\mu}_n$. Finally, we denote by $\theta^{(n)}$ the (random) parameter learned by a learning algorithm (e.g., SGD, ADAM) optimizing $\mathscr{L}_{\mathrm{DSM}}^{(n)}$. The following lemma is particular case of the results of Farghly et al. (2026); Dupuis et al. (2025).

**Lemma D.1** (Expected decomposition). *For all $t \in [0, T]$,*

$$\mathbb{E}\left[\varepsilon_t^{\theta^{(n)}}(\overrightarrow{X}_t)\right] \leqslant \mathbb{E}\left[\mathscr{L}_{\mathrm{ESM}}^{(n)}(\theta^{(n)}, t) + \mathscr{G}^{(n)}(\theta^{(n)}, t)\right] ,$$

*where $\mathscr{L}_{\mathrm{ESM}}^{(n)}$ is the empirical explicit score matching loss at time t, given by*

$$\mathscr{L}_{\mathrm{ESM}}^{(n)}(\theta, t) := \mathbb{E}\left[\left\|s^\theta(t, \overrightarrow{X}_t^{(n)}) - 2\nabla \log \tilde{p}_t(\overrightarrow{X}_t^{(n)})\right\|^2\right] .$$

The proof of this lemma can be found for instance in (Dupuis et al., 2025, Lemma 3.2) (take $\lambda := \delta_t$ in their proof).

**Corollary D.2.** *With the same conditions and notation as Theorem 3.7, we have*

$$\mathbb{E}\left[\mathrm{KL}\left(\mu | \overleftarrow{\mu}_T^{\theta^{(n)}}\right)\right] \leqslant \mathrm{K}_T + \mathbb{E}\left[\int_0^T \frac{\lambda_T(t)}{2}\left(\mathscr{L}_{\mathrm{ESM}}^{(n)}(\theta^{(n)}, t) + \mathscr{G}^{(n)}(\theta^{(n)}, t)\right)\mathrm{d}t\right] ,$$

*Proof.* This corollary is an immediate consequence of Theorem 3.7, Lemma D.1 and Tonelli's theorem. $\square$

This corollary relates the performance of DMs to the generalization error $\mathscr{G}^{(n)}(\theta^{(n)}, t)$ of the DSM loss with a better dependence on $T$ compared to (Dupuis et al., 2025; Farghly et al., 2026). It shows that the overall generation performance is mainly impacted by the generalization error associated with small noise levels (*i.e.*, values of $t$ close to $T$ in the above equation). As we have seen above, this generalization of our results is only based on Lemma D.1 and a direct application of our bounds. Therefore, it is clear that some of our other results may be generalized similarly. In particular, we could extend our Wasserstein bounds of Section 4.2 to the generalization error framework, which is new to the best of our knowledge.

## E. Experiments details

### E.1. Toy setting

We consider a simple toy distribution consisting of a uniform distribution on a circle embedded in 2-dimensional Euclidean space. We use a simple training set consisting of 8 points spread equidistant on the circle and we train a diffusion model based on a minimal feed-forward neural network. The implementation is based on a blog that can be found here and is similar to what is considered in (Farghly et al., 2025). Each tick on the x-axis of the plots in Figure 7 represents 1000 epochs.

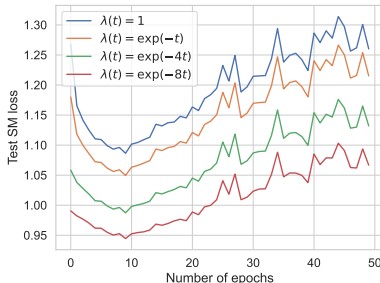

*Figure 7.* Weighted test score matching loss plotted against number of epochs during training.

### E.2. CIFAR-10

We also consider CIFAR-10, an implementation of the DDPM model from (Song et al., 2021b). We use the configuration titled `vp.ddpm.cifar10_continuous` which implements the DDPM model of (Ho et al., 2020) but for the continuous-time variance preserving setting. The architecture is a U-Net (Ronneberger et al., 2015) with the encoder and decoder each consisting of four resolution levels $(32 \times 32, 16 \times 16, 8 \times 8, 4 \times 4)$, with two residual blocks per level and utilizes self-attention at a resolution of $16 \times 16$. The model is conditioned on time, with the timestep encoded via a sinusoidal embedding and injected into each residual block.

For Figure 4, we choose $x, y$ by randomly choosing a test data point $z_0$, taking a sample $z \sim \tilde{p}_{t|0}(dz|z_0)$ and then setting $x = z + r_x \xi_x, y = z + r_y \xi_y$, where $\xi_x, \xi_y$ are standard multivariate Gaussians, and $r_x, r_y$ are Weibull distributed scalars.

For Figures 6 we use the implementation of the DDIM sampler with the same initial Gaussian noise across values of $t_p$. For Figure 5, we calculate the change in KL divergence by instead computing the change in log-likelihood using the `bpd` implementation in the codebase. Since this is a stochastic estimate, we compute 5 batches of log-likelihood values and plot the mean and the standard deviation.

