# OpenReview forum: "Tightening the Score Matching Gap for Diffusion Models"
_ICML.cc/2026/Conference — ICML 2026 regular_

### Official Review · Reviewer_Deji · 2026-03-13

**Soundness:** 3
**Presentation:** 3
**Significance:** 2
**Originality:** 3
**Overall Recommendation:** 4
**Confidence:** 3

**Summary:**

This paper aims to address the discrepancy between the theoretically intractable KL divergence (sample quality) and the tractable score matching loss (the surrogate optimized during training) in diffusion models. Because the tractable score matching loss used during training is merely an upper bound on the true KL divergence, meaning a lower score matching loss does not strictly guarantee better sample quality. To address this, the authors exploit the contraction properties of backward processes using entropy flows, Logarithmic Sobolev Inequalities (LSI), and reflection couplings. They derive tighter, time-weighted bounds for the Forward KL divergence, Reverse KL divergence, and Wasserstein distance. A central finding is the formalization of time-decaying weightings (e.g., $\lambda_T(t)$), which mathematically demonstrate that the quality of the score approximation at low noise scales (small $t$) disproportionately impacts the final distribution gap.

**Compliance With Llm Reviewing Policy:**

Affirmed.

**Key Questions For Authors:**

1. Are there theoretical pathways to relax the $\rho_0$-LSI assumption on the data distribution $\mu$ to a weaker condition (such as a Poincaré inequality) to make the Reverse KL bounds more applicable to multimodal, real-world datasets?

**Limitations:**

Yes.

**Strengths And Weaknesses:**

## Strengths

- The paper is technically sound and well-structured.

- They unite several advanced techniques from stochastic analysis. The adaptation of entropy flows to bound the forward and reverse KL divergences and the use of Eberle’s reflection coupling to tighten the time-dependence of Wasserstein bounds are interesting.


## Weaknesses

- The tighter Forward KL bounds (Theorems 3.8 and 3.10) heavily rely on the learned score network $s^\theta$ satisfying a global pseudo-Lipschitz condition and one-point dissipativity. However, modern highly non-linear architectures (e.g., Transformers, deep U-Nets) rarely satisfy global Lipschitz constraints without performance-degrading architectural interventions (like spectral normalization).

- The paper assumes the target data distribution $\mu$ satisfies a $\rho_0$-LSI. Real-world image distributions often lie on disconnected, low-dimensional manifolds and generally do not satisfy an LSI.

- In Corollary 4.2, the authors use a Donsker-Varadhan change of measure to translate the Reverse KL bound (which relies on the untractable approximated path measure) into an expectation over the true forward process. This results in an exponential term: $\log \mathbb{E}[e^{\rho_t \varepsilon_t^\theta(\vec{X}_t)^2}]$. In high-dimensional settings, empirical estimators of exponential expectations suffer from notoriously high variance, which may limit the practical utility of this bound as an evaluation tool.

---

> ### Author Rebuttal · Authors · 2026-03-30
>
> We thank the reviewer for their insightful review and the positive assessment of our paper.
>
> *The tighter Forward KL bounds (Theorems 3.8 and 3.10) heavily rely on the learned score network  satisfying a global pseudo-Lipschitz condition and one-point dissipativity. [...]*
>
> It is a pertinent remark from the reviewer that pseudo-Lipschitz and dissipativity conditions might not always be satisfied by modern architectures.
> Note that, even when such assumptions are not satisfied, theorem 3.7 shows that our proof technique still leads to improved bounds compared to the classical results.
> Moreover, theorem 3.8 shows that we improve over existing bounds even when the Lipschitz constant diverges as $t\to 0$.
>
> Moreover, we highlight that not all our results require such regularity assumptions. In particular, the reversed KL bounds do not rely on restrictive regularity assumptions on the score network, which is one of our motivations for extending our results to this topology.
>
> To make this discussion clearer, we provide additional experiments where we estimate the value of $M_t$ (thm 3.8) in a practically relevant setting (https://ibb.co/FQqfg2M, Fig 1).
>
>
> *The paper assumes the target data distribution  satisfies a -LSI. Real-world image distributions often lie on disconnected, low-dimensional manifolds and generally do not satisfy an LSI.*
> *Are there theoretical pathways to relax the $\rho_0$-LSI assumption on the data distribution  to a weaker condition (such as a Poincaré inequality) to make the Reverse KL bounds more applicable to multimodal, real-world datasets?*
>
> First, let us highlight that only the reverse KL bounds (theorem 4.1 and corollary 4.2) require the data distribution to satisfy a LSI.
> In particular, our forward KL bounds and our Wasserstein bounds do not require this assumption.
>
> Moreover, as we detail in the answer to Reviewer a3Jw, we can show that our assumptions can be relaxed so that we *do not need to assume* that the data distribution has a finite Fisher information or relative entropy with respect to the invariant measure. Therefore, most of results (forward KL and Wasserstein) are compatible with real-world distributions that lie on low-dimensional manifolds.
> We will make this clear in the next version and we thank the reviewer for this pertinent question.
>
> We investigated different ways to weaken the log-Sobolev assumption in the reverse KL bounds and we can answer *yes* to your question.
> The distribution of the forward process at a given time is a weighted convolution of the data distribution and the standard Gaussian. Recent works show that such distributions satisfy the log-Sobolev inequality if the data distribution is subgaussian with a large enough subgaussian norm, see for instance [1] below.
> In particular, this is true if the data distribution has compact support which is the case for any bounded dataset, like image distributions.
> We will add a theoretical result in Section 4 to handle this case precisely.
> We thank the reviewer for this question that enabled us to significantly strengthen our paper.
>
> *In Corollary 4.2, the authors use a Donsker-Varadhan change of measure...*
>
> This is an interesting remark from the reviewer. It is correct that this exponential term might be hard to evaluate in practice.
> Note that the main goal of this corollary is mainly to enhance the theoretical relevance of theorem 4.1. It is not a main result of the paper and is not meant for empirical applications.
> That being said, it might be possible to improve Corollary 4.2 by using different change of measure techniques, which is an interesting research direction which we leave for future work.
> In the next version, we will further discuss the practicality of this corollary.
>
> [1] Feng-Yu Wang. Jian Wang. "Functional inequalities for convolution probability measures." Ann. Inst. H. Poincaré Probab. Statist. 52 (2) 898 - 914, May 2016. https://doi.org/10.1214/14-AIHP659

---

> > ### Author Rebuttal · Reviewer_Deji · 2026-04-04
> >
> > Thank you for the detailed response and the additional experiments. Regarding my concern with Corollary 4.2, simply stating that the corollary is meant only for 'theoretical relevance' rather than empirical application does not fully resolve my concern. In high-dimensional settings, an expectation with such notoriously high variance means the bound is effectively vacuous as a practical evaluation tool. While I understand this may be left for future work, conceding its lack of practical utility significantly weakens the impact of the result. For this reason, I will maintain my score of 4.

---

> > > ### Author Response · Authors · 2026-04-04
> > >
> > > We thank you for your quick response. We agree with the reviewer that Corollary 4.2 is primarily of theoretical interest and is out of place in that respect. We will change it to a remark in the next version, to not allow this minor result to distract from the practical importance of the theorems.
> > >
> > > We hope that the reviewer agrees that, even with the limitations of this corollary, our work makes several contributions (tighter forward KL bounds, reverse KL bounds requiring only data regularity, Wasserstein bounds via reflection couplings, and new empirically-validated evaluation weightings) that the community can build upon.
> > >
> > > We also note that during this rebuttal, we have strengthened the paper beyond its initial submission: relaxing the Fisher information assumption to finite second moments, showing that the LSI condition can be replaced by compact support for reverse KL bounds, and providing new CIFAR-10 experiments validating our theory. We hope the reviewer might weigh these improvements, alongside the technical merits they identified in their initial review, when considering their final score.
> > >
> > > We thank the reviewer for their constructive engagement which has helped improve the paper.

---

### Official Review · Reviewer_oMHT · 2026-03-13

**Soundness:** 3
**Presentation:** 3
**Significance:** 3
**Originality:** 3
**Overall Recommendation:** 5
**Confidence:** 4

**Summary:**

Diffusion models are trained by minimizing a score-matching loss, but in some sense the actual loss of interest is the Kullback-Leibler divergence (KLD) between the trained distribution and the true distribution. These are related via an ELBO-like inequality, but it is somewhat unclear to what extent this inequality is tight. Prior work suggests it is in fact generically not tight, and (as the authors of this paper point out) this has empirical consequences: to get an understanding of whether a model is good or bad, we need to go beyond the (test) score-matching loss, and consider measures of sample quality like FID. This would not be necessary if the score-matching loss faithfully reflected the true KLD.

What can we say mathematically about the so-called 'score-matching gap', i.e., the size of the difference between the KLD and the score-matching loss? Is there a way to make it tighter? The authors prove a variety of results that address this question, each using somewhat different mathematical tricks. The inequalities obtained generalize the familiar one in a variety of interesting ways.

**Compliance With Llm Reviewing Policy:**

Affirmed.

**Final Justification:**

The authors mostly addressed my concerns, and I think their work is well-scoped and valuable to the community. I think it should be accepted.

**Key Questions For Authors:**

1. When might a practitioner be interested in each type of bound? Is there one bound to rule them all, or is each bound separately informative in not-totally-overlapping ways?

2. How do the authors reconcile their work with work on generalization? As mentioned in the "time decay analysis" section, it's possible that apparently 'suboptimal' time-weighting functions are used because they help avoid overfitting at small noise scales. If something like this is true, what do we learn from some of these bounds?

**Limitations:**

There is a limitations section, but it doesn't mention some of the important limitations, like the relationship between generalization and the authors' bounds. Also, some bounds may be hard to study empirically (e.g., Theorem 3.10, which is fairly formal due to its dependence on a non-explicit function C).

**Strengths And Weaknesses:**

## Strengths

Overall, I like the paper. Its introduction is well-written and the problem is well-scoped. There is a good balance between presenting the actual math results the authors proved, and providing intuition for how those results were proved (without overwhelming the reader with technical detail). I was impressed by the authors' command of the relevant literature.

The math looks carefully done and correct, although I did not verify the details.

The authors touch on an important problem, and a systematic treatment of the associated mathematics will likely benefit the community of people thinking about it. Their work suggests (although they do not discuss this much; but see "Time decay analysis" for some discussion) alternative training objectives for diffusion models which could be interesting to study empirically.

## Weaknesses

My main issue with the paper is probably that the reader is likely to get overwhelmed with technical details in the middle of it---or, at least, this is what happened to me. As I understand it, the authors' goal is to present a variety of generalizations of the familiar KL vs score-matching upper bound, each associated with different proof techniques and underlying logic. In principle, this helps us understand the many possible ways we could tighten the usual bound. (Which is good!)

I'm not totally sure of what would be helpful. One thing that is already present (and that is helpful) is a separation between some of these results; one kind of bound is in Sec. 3, while other kinds are in Sec. 4. Maybe what would be helpful is a little more discussion of (i) the motivation for each type of bound, and (ii) what we specifically learn from them. For example, the "Time decay analysis" section compares the result of Theorem 3.8 to the standard practice of having some time-weighting function. If there are other analogues, it would be helpful to discuss them for the later theorems; if there aren't, it may be helpful still to briefly comment on what they would imply about a 'good' training objective.

A simple possible helpful change would be to "name" theorems, e.g., "LSI-based bound" or similar.

Maybe I missed this, but it would also be helpful to see some discussion of which of the several bounds proved is most 'informative' in practice. If I care about a bound which is as tight as possible, which one should I look at? Or do they all have their pros and cons? (If so, when do I use one versus another?)

More broadly, it would be helpful if the authors guided the reader more gently through some of the intermediate results, discussing motivation and what we learn as non-technically as possible.

Two minor nitpicks. First, the name of $\bar{\lambda}$ seems to be different in the text than in Fig. 2 and its caption (no bar in Fig. 2). Also, the paper uses "time-continuous", "time-discrete", and "time-uniform"; I am used to seeing, e.g., "continuous-time" and "discrete-time", but it's possible the authors are just using a convention I wasn't aware of.

There are some typos throughout. Please look through and fix these and ones I didn't catch:

line 100: left apostrophe surrounding \chi^2 should be `` , not '' .

line 161: "Browwian" -> "Brownian"

line 175: "in generality" -> "in general"

line 200: "motivatesus" -> "motivates us"

line 310: "mprove" -> "improves" and "tighten" -> "tightens"

line 314: left quotation mark is wrong way

line 315, left column: "mentionned" -> "mentioned"

---

> ### Author Rebuttal · Authors · 2026-03-30
>
> We thank the reviewer for their thoughtful review and for their kind assessment of the presentation of our paper and the significance of the problem we have identified.
> We believe we have addressed the raised concerns and kindly ask whether the reviewer could increase their score based on our answer.
>
> **"Technically overwhelming paper and presentation", "which of the several bounds proved is most 'informative'"**
>
> We thank the reviewer for raising the point that the variety of our results may render the exposition dense. We will improve the clarity of the paper, as detailed below.
>
> The most informative and general KL bound is Theorem 3.7, which shows that the time decay of the forward KL bound is directly related to the LSI constants. Theorems 3.8 and 3.10 that follow are an application to the setting of pseudo-Lipschitz and dissipative score functions. The results that follow in Section 4 are an extension of our analysis in Section 3 to two alternative topologies, showing that different topologies require different regularity assumptions.
>
> Since section 3 contains the most significant result, we will simplify the exposition by moving Assumption 3.5 and Proposition 3.6 to section 4.1, labeling some of the propositions before Theorem 3.7 as 'lemma'. By having section 3 contain fewer propositions and theorems, we believe this will direct the eye of the reader who is less familiar with the area more easily.
>
> In the next version, we will also make this discussion clearer by highlighting the most informative results in the introduction. As you suggested, we will add additional discussion on the motivation of each bound and implement the naming of the theorems.
>
> We thank the reviewer for these remarks that helped us improve the clarity of our exposition.
>
> **"If there are other analogues, it would be helpful to discuss them", "do they all have their pros and cons?"**
>
> All our bounds suggest that the score approximation is more critical for the performance for small noise scales.
> Therefore, the discussion of the ``time analysis paragraph'' is applicable to all our bounds.
> As discussed, the reversed KL bounds take the expectation over the estimated process, which might make them less relevant for certain applications.
> We will make these distinctions clear.
>
> We thank the reviewer for catching some typos that we have now corrected.
>
>
> **Q1** We believe that practitioners will be particularly interested in the forward KL bounds, which tighten the usual ELBO.
>
> However, the reverse KL and Wasserstein bounds are valuable in complementary settings. Reverse KL bounds (Thm 4.1) are preferable when network regularity is hard to verify, as they use assumptions on the data distribution. Wasserstein bounds (Thm 4.5) are relevant when closeness in sample space rather than likelihood is of interest.
>
> While our bounds each rely on different regularity conditions and proof techniques, they all lead to the introduction of a time decay term in the bounds with similar qualitative behaviors, as we can see in Figures 1 and 2.
>
> **Q2** *briefly comment on what they would imply about a 'good' training objective.*
>
> This is a pertinent remark from the reviewer. Our new proof techniques can be used to obtain tighter generalization or convergence bounds, with improved time-dependence.
> We make a step in this direction in Appendix D, where we relate our bounds to a recently proposed generalization framework, leading to improved generalization bounds. In the next version, we will emphasize these aspects more.
>
> Our work suggests an important distinction between the train and evaluation weightings.
> As discussed in Section 5, this might lead to better evaluation metrics by providing a tighter upper bound on the KL divergence objective.
> As noted by the reviewer, we discuss in the time decay paragraph that a bad score approximation is particularly harmful for small noise scales. Thus, one might want to use small weightings at small noise scales to avoid potentially harmful overfitting.
>
> As mentioned to Reviewer a3Jw, we have now performed new experiments in this anonymized [link](https://ibb.co/FQqfg2M) to support our theory, which we will include in the next version.
>
> **Limitations**
> We thank the reviewer for suggesting a deeper discussion of the link with generalization. We will implement it by (1) highlighting more the link with generalization bounds presented in Section D and (2) making the practical utility of the bounds clearer to practitioners, as described above.
> We agree that Theorem 3.10 has a mostly theoretical impact. To make the empirical study of our bounds clearer, we provide additional experiments in this anonymized [link](https://ibb.co/FQqfg2M) (fig. 1), where we numerically estimate the parameters appearing in Theorem 3.8 in a practically relevant setting.

---

> > ### Author Rebuttal · Reviewer_oMHT · 2026-04-04
> >
> > I thank the authors for addressing my concerns. I think the paper is a valuable contribution. I will increase my score.

---

> > > ### Author Response · Authors · 2026-04-04
> > >
> > > We warmly thank you for your feedback, which will greatly enhance the next version of our work, the positive assessment as well as the score upgrade.

---

### Official Review · Reviewer_TX5p · 2026-03-14

**Soundness:** 3
**Presentation:** 3
**Significance:** 2
**Originality:** 2
**Overall Recommendation:** 3
**Confidence:** 5

**Summary:**

This work studies score matching gap, produced by the the difference between distribution estimation error and the score matching loss. The authors establish tighter bound for three metrics: KL divergence, reverse KL divergence, and Wasserstein distance.  To obtain these bounds, contraction properties of the backward processes are exploited. In particular, the analysis relies on entropy flows, logarithmic Sobolev inequalities and reflection couplings.

**Compliance With Llm Reviewing Policy:**

Affirmed.

**Final Justification:**

I believe my evaluation was fair.

**Key Questions For Authors:**

I have several questions:

1. I find Proposition 3.1 somewhat unconvincing as motivation. Showing the existence of a bad score estimator for which the classical bound is loose does not by itself say much about realistic estimators produced by practical training procedures. As I already mentioned in the weaknesses, the authors should clarify the meaning of "score matching gap" in this work. This proposition does not present as a supporting evidence for gap.
2. Could you explain why you care about reverse KL bound. While mathematically interesting and valid, it is not clear that this quantity provides insights to practical diffusion models analysis.

**Limitations:**

yes

**Strengths And Weaknesses:**

This paper establish tighter bound for three metrics: KL divergence, reverse KL divergence, and Wasserstein distance by exploiting the contraction properties of the backward processes in diffusion models. The presentation is clear and easy to follow. Although this is a theoretically interesting paper, I have some concerns:

1. My first concern is on the meaning of "score matching gap". To me, it sounds like the score matching loss cannot be optimally minimized. However, many papers such as [1] shows that score matching loss enjoys minimax optimal statistical estimation rate; so is distribution estimation error. The contribution in this work is better understood as a different proof and decomposition framework, not as the discovery of a fundamentally new deficiency in score matching.
2. The second concern is the presentation of bounds. The paper’s main claims are presented mostly in decomposition form including Theorem 3.7, 3.10, 4.1 and 4.5, which makes it difficult to compare directly with prior work. The theorems show that distribution estimation error can be bounded by weighted score matching errors plus initial distribution approximation error. Nevertheless, the paper does not show explicit end-to-end rates under concrete estimation assumptions. As a result, it is not actually clear whether the bounds are indeed tighter than existing ones in prior work.
3. I also do see the technical difficulties of time-inhomogeneous weight function while the authors emphasize a lot in the manuscript. I believe the existing techniques and analysis in prior can be applied to this setup with only minor modifications. The authors should clarify why this is an important contribution under the proposed new analytical framework.



[1] Oko, Kazusato, Shunta Akiyama, and Taiji Suzuki. "Diffusion models are minimax optimal distribution estimators." International Conference on Machine Learning. PMLR, 2023.

---

> ### Author Rebuttal · Authors · 2026-03-30
>
> We thank the reviewer for their time and detailed feedback.
>
> We clarify below some potential confusions and address the raised concerns. We hope that the reviewer could reconsider their score in case they are addressed.
>
> **Weakness 1**
>
> *Score matching gap*
>
> What we mean by the score matching gap (line 70 l.) is the (often significant) difference between the KL divergence (between generated and target distributions), and the ELBO, commonly invoked to derive the score matching loss.
> The presence of this gap does **not** mean that the score matching loss cannot be optimally minimized but that it might not be the best quantity to evaluate the performance of diffusion models in practice, which is what we want to address.
>
> *Contributions*
>
> We agree that our toolbox, if used for minimax rates derivation (as in Oko et al.), would only yield minor improvements.
> However, we insist that our focus is **not** on minimax rates or sample complexity. These studies are learning algorithm agnostic and cannot precisely explain or improve practical performances.
>
> In this work, we take a complementary approach and identify cases where the score matching gap can be tightened, suggesting better ways to weight the loss in practice.
> Precisely, the classical ELBO relies on a data processing inequality and Girsanov's theorem, yielding linear dependence in $T$, our core technical contribution is to improve the time-dependence of these bounds.
>
> By tightening existing results on KL and Wasserstein, our theory could also improve generalization bounds, a link that we made formal in *Appendix D*. In this algorithm-dependent case, we do not improve the sample complexity, however, we obtain a significant improvement in terms of time dependency.
>
> Thus, our results do not to provide 'different proof and decomposition framework', but an improvement of existing results.
> We will add a dedicated paragraph on this matter to improve clarity.
>
> **Weakness 2**
>
> Indeed, our bounds sum an initialization term and a score approximation term. This allows comparison with existing bounds such as Eq. (1).
> By analyzing such decompositions, we observe that our technique provides tighter bounds for all terms that appear.
> Our work may also be compared with the convergence bounds literature (in a slightly different setting), such as Conforti et al., where the results are also in a decomposition form.
> In this case, we also observe that our theory leads to improved time dependence under similar estimation assumptions, by avoiding the use of the data processing inequality.
>
> As explained above, the goal of our paper is not to compare with sample complexity bounds, but is a complementary approach, providing algorithmic insights on the performance of diffusion models by tightening the score matching gap.
>
> **Weakness 3**
>
> Our theorems involve inhomogeneous time functions as a direct consequence of our analysis that strictly improves the classical ELBO.
> We argue that this is an important contribution for two reasons: (i) it improves the time dependence of existing bounds and (ii) it suggests new evaluation (and train) losses that better capture the performance (see experiments).
> Thus, rather than a new analytical framework, we provide new bounds that are tighter than existing ones.
> These results are significantly different from existing works, as we successfully avoided the data processing and Girsanov theorem.
>
>
> **Q1**
>
> We suspect a misunderstanding here.
> Proposition 3.1 does not show the existence of a bad score estimator for which the classical bound is loose. On the contrary, it shows that the classical ELBO bound is tight in the worst case, ie, there exist (bad) score estimators for which it is almost tight.
> This motivates the following question: "when can we tighten the score matching gap (as defined above)?" And, more importantly, can we tighten it for practically trained estimators?
>
> Prop 3.1 shows that tightening this ELBO requires additional assumption on the score estimator, which we do in Section 3.2 and 4.2.
>
> Therefore, this result supports the existence of a score matching gap, as it is unclear when the classical ELBO is not tight.
>
> **Q2**
>
> We have several motivations for reverse KL bounds.
> While the forward KL bounds rely on the regularity of the score network, the reversed KL analysis relies on assumptions on the data distribution (which we have now weakened, see the answer to Reviewer Deji). In some settings, verifying regularity of the data distribution (e.g., log-concavity or LSI) may be easier than verifying regularity of a trained neural network.
>
> The reviewer is correct that the reverse KL is not often considered in the literature, but we believe it is because it is hard to estimate. Our results make it possible to more closely estimate the reverse KL using a weighted score matching loss.
>
> The qualitative conclusions from the reverse and forward KL bounds are similar, showing consistency across topologies, which is an additional motivation.

---

> > ### Author Rebuttal · Reviewer_TX5p · 2026-04-04
> >
> > Thank you for your response. As I mentioned, ELBO is not a theoretically sound way to derive score matching loss while KL divergence and Girsanov's theorem is. I am still confused why you focus on the difference of this two. In terms of the algorithm-dependent result, [1] provides the first result on gradient-based analysis of score matching and their framework also leads to a minimax optimal sample complexity. Thus, the argument from this perspective does not make sense. Finally, time-dependent or say time-inhomogeneous analysis were established in the literature. Therefore, my impression is that this manuscript provides marginal contributions after reading your rebuttal.
> >
> >
> >
> > [1] Han, Yinbin, Meisam Razaviyayn, and Renyuan Xu. "Neural Network-Based Score Estimation in Diffusion Models: Optimization and Generalization." The Twelfth International Conference on Learning Representations.

---

> > > ### Author Response · Authors · 2026-04-04
> > >
> > > We thank the reviewer for their answer. We believe there are misunderstandings regarding our contributions. We hope to clarify the concerns of the reviewer below.
> > >
> > > **``ELBO is not a theoretically sound way to derive score matching loss while KL divergence and Girsanov's theorem is''**
> > >
> > > We believe there is a misunderstanding here that we would like to clarify. The classical ELBO bound (Equation 1 and Song et al. (2021, Theorem 1)) is derived *using Girsanov's theorem*-- these are not two separate approaches. The looseness in the bound comes from the data processing inequality step used before applying Girsanov, which bounds the KL between marginals by the KL between path measures. Our contribution is precisely to avoid this data processing step, yielding strictly tighter bounds. We are not comparing two different derivations, we are improving the standard one by introducing a decaying time weighting. We apologize if there is a confusion on the terms that are used here.
> > >
> > > The KL divergence between the data and model distributions cannot be directly computed, whereas the score matching loss can. Therefore defining the score matching gap makes sense as it measures the discrepancy between the theoretical quantity of interest and the upper bound which suggests the DSM loss as an empirical proxy. By tightening the score matching gap, our theory suggests new losses to evaluate diffusion models. In the paper and in the new experiments (see reviewer a3JW), we show the practical relevance of the new losses suggested by our theory.
> > >
> > > **``[Han et al. (2024)] provides the first result on gradient-based analysis of score matching [...] Thus, the argument from this perspective does not make sense''**
> > >
> > > We thank the reviewer for this reference but we would like to stress that it addresses a different question: it establishes that score matching achieves minimax optimal estimation rates.
> > > We would like to emphasize that the primary goal of our paper is not to achieve minimax optimality or obtain convergence bounds. A minimax optimal estimator can still have a loose relationship between its score matching loss and the resulting KL divergence -- the rate tells you how fast the loss decreases with samples, not how faithfully the loss reflects sample quality at any given point in training. This latter question is what we address, and our time-decaying weightings provide a finer-grained answer compared to the classical analysis.
> > >
> > > While we appreciate the reviewer's interest in the statistical theory of diffusion models, we note that much of the diffusion model literature -- from improved noise schedules (Karras et al., 2022) to architectural choices (Dhariwal and Nichol, 2021) -- proposes changes that improve practical performance without necessarily improving minimax rates. Judging contributions solely by whether they improve sample complexity would exclude a large portion of impactful work in this area.
> > >
> > >
> > > **``time-inhomogeneous analysis were established in the literature''**
> > >
> > > The goal of our paper is not to analyze the score matching loss with a time-inhomogeneous weighting. Instead, our work derives a weighting that refines and tightens the classical Girsanov-based upper bound. These time factors stem from a careful analysis of the backward distribution, based on recent works on perturbation analysis of log-Sobolev inequalities and reflection couplings (among others). To the best of our knowledge, no such analysis exists in the literature. If we have misunderstood what the reviewer meant here, and indeed they understood the purpose of our work, we kindly ask them to point us to the specific prior results they have in mind so that we can discuss the comparison.
> > >
> > > **``Therefore, my impression is that this manuscript provides marginal contributions after reading your rebuttal.''**
> > >
> > > We respectfully disagree that our contributions are marginal. We provide strictly tighter bounds than the classical ELBO, using novel technical tools (entropy flows, reflection couplings for mismatched drifts, perturbation analysis of LSIs). During this rebuttal, we have further strengthened the paper by relaxing assumptions to finite second moments and compact support, and by providing new CIFAR-10 experiments validating our theory.
> > >
> > > We thank the reviewer for their time and help with improving the paper and we hope that they will reconsider their assessment in light of these clarifications and improvements.

---

### Official Review · Reviewer_a3Jw · 2026-03-16

**Soundness:** 3
**Presentation:** 3
**Significance:** 2
**Originality:** 2
**Overall Recommendation:** 4
**Confidence:** 3

**Summary:**

This paper investigates the gap between the Kullback-Leibler (KL) divergence of sample distribution and the score matching training loss. It is discovered that without specifying a particular score estimator, the score matching gap cannot be further tightened. Moreover, under several regularity conditions, including a time-dependent LSI condition and a pseudo-Lipschitz condition, the paper derives new upper bounds for the forward KL divergence. In addition, the results are extended to other topologies measured by the Wasserstein distance and the reserve KL divergence.

**Compliance With Llm Reviewing Policy:**

Affirmed.

**Final Justification:**

This paper studies  the gap between the KL divergence of sample distribution and the score matching training loss. Overall, my final recommendation is 4. Compared to my initial review, my assessment has slightly improved after considering the authors' rebuttal, as the authors have addressed my concern about assumptions and applications.

**Key Questions For Authors:**

1.	In Figure 3, what does the number of epochs mean? In addition, why both the KL divergence and the reverse KL divergence increase as the number of epochs increases?

2.	As stated in weakness part, could the theoretical result derived in this paper be applied to improve the sample quality in practice?

3.	It seems that the regularity conditions presented in Section 3 is different from those presented in Section 4. Could the authors clarify why this difference is necessary?

**Limitations:**

yes

**Strengths And Weaknesses:**

Strengths:

This paper derives tighter bounds for the score matching loss, and extends results to alternative topologies. The theoretical analysis is technically sound, and the presentation is generally clear and well written.

Weakness:

Weakness: Although the paper derives new bounds, it remains unclear how to apply these bounds to improve practical performance in diffusion models. First, it is unknown whether the assumptions required in the analysis are satisfied on real-world datasets, such as Assumptions 3.2, 3.3 and regularity conditions. In particular, this paper does not discuss how to verify these conditions in practice.

In addition, it is unclear whether the new bounds lead to improvements in sample quality. The authors provide numerical validation on a toy example to show the correctness of theoretical results. However, no empirical evidence is provided to show that the new bounds could contribute to improved sample quality on large-scale real-world datasets such as ImageNet.

Finally, how to determine the regularity parameters in Theorem 3.7-3.10, such as $\gamma$, $M$ remains unknown. Additional discussion on the choice of these parameters in practice would improve the applicability of theoretical results.

---

> ### Author Rebuttal · Authors · 2026-03-30
>
> We thank the reviewer for acknowledging the technical soundness and clarity of presentation. We address each concern below.
>
> **Are the assumptions required in the analysis satisfied on real-world datasets, such as Assumptions 3.2, 3.3 ?**
>
> *Assumption 3.2:* This is a minor regularity condition that ensures the estimated backward density is well-defined. This is required for any analysis of the backwards SDE and is standard (but sometimes not explicitly stated) in the literature. E.g. a sufficient condition is that the score satisfies a linear growth bound, which DNNs are indeed known to satisfy (see 3.3 of [2] for weaker conditions). In the next version, we will include a detailed remark discussing this.
>
> *Assumption 3.3:* Thank you for spotting this. Upon careful analysis, the finite Fisher condition can be largely relaxed.
> Indeed, our bounds can be adapted by replacing $e^{-2T}\mathrm{KL}(\mu | \gamma^d)$ by $e^{-2(T-\epsilon)} \mathrm{KL}(\mu_\epsilon | \gamma^d)$, which, by the results of [1] is in $O(\epsilon^{-1})$ as soon as $\mu$ has finite second-order moment. Therefore, a tuning of $\epsilon$ provides an exponentially decaying initialization term, *without* assumption 3.3.
>
> **Verifying the regularity assumptions in practice and determining the regularity parameters in Theorem 3.7-3.10, such as $(M,\gamma)$**
>
> $\gamma$ in Thm 3.7 is the LSI constant that is estimated in 3.8 and 3.10 based on regularity assumptions on the score network (e.g. $M_t$).
>
> $M_t$ and the other regularity parameters are a property of the family of score networks. To illustrate, we conducted experiments on a U-Net trained on CIFAR-10 and provided results in Figure 1 of the document [here](https://ibb.co/FQqfg2M). By randomly sampling points, we estimate the pseudo-Lipschitz constant $M_t$ appearing in Thm 3.8. As expected, the regularity constant has small values for big enough $t$ but can become larger as $t\to 0$. Note that this is compatible with Thm 3.8, where $M_t$ might diverge as $t\to 0$, still yielding improved bounds compared to the literature.
>
> **How to apply these bounds to improve practical performance? Do the new bounds improve sample quality on a large-scale real dataset ?**
>
> While our contributions are mostly theoretical, we believe that they have concrete practical applications. Sample-based evaluation metrics, such as FID or KL divergence, are costly to estimate in practice and often the score matching loss is used in its place. Our theory suggests new ways to weight the score matching loss in practice to make it a more faithful proxy for such metrics which could lead to more accurate evaluation methods for diffusion models.
>
> To substantiate this claim, we conducted new experiments on CIFAR-10 showing that our theoretically-motivated weighting is the only one to accurately track the true KL divergence, while standard validation losses fail at small times. This experiment is available at this anonymized [link](https://ibb.co/FQqfg2M) (Figures 2 & 3). Starting with a trained U-Net, we add a Gaussian perturbation on a small time window ('perturbation time' on the figure) and study the impact of this perturbation on sample quality. We compute the KL divergence and the SM loss with the ELBO weighting, train weighting and our exponential weighting (from the theory in favorable cases). We observe that our weighting is the only one to accurately reproduce the shape of true KL divergence, while the usually used validation loss fails to predict the explosion of the loss for small perturbation times. This shows that our theory might help create better evaluation metrics, which could help to improve sample quality.
>
> In addition, we discussed, in the time decay analysis paragraph (Section 3), that our theory aligns with some of the standard practices to train diffusion models. We will add these experiments in the revision.
>
> **Q1** In Figure 3, the number of epochs refers to the number of training epochs and the reported quantities all refer to test losses. Thus, it is expected that these losses start increasing as the model overfits the data and notably, the small-timestep SM loss tracks this overfitting transition most closely, which is precisely what our time-decaying bounds predict.
>
> **Q2** See the answer to the weaknesses above.
>
> **Q3** The assumptions are different because we rely on two completely different proof techniques---entropy flows and Wasserstein contractions. We have since found that the condition of Thm 3.8 with parameters $(\rho, L, 2R)$ implies Definition 4.3 with parameters $(L, R, \rho)$ and will consider using the same conditions for both results for simplicity.
>
> **We hope the new experiments, relaxation of assumptions, and clarifications address the concerns. We would be grateful if the reviewer considered increasing their score.**
>
> [1] Otto, Villani Comment on: “Hypercontractivity of Hamilton–Jacobi equations”, 2001
>
> [2] Pavliotis, “Stochastic Processes and Applications.”, 2014

---

> > ### Author Rebuttal · Reviewer_a3Jw · 2026-04-03
> >
> > Thank you for your response. I have one further question. You mention that " our weighting is the only one to accurately reproduce the shape of true KL divergence". Could you clarify what you mean by "reproduce the shape"? Once I understand your explanation, I will consider increasing my score.

---

> > > ### Author Response · Authors · 2026-04-03
> > >
> > > Thank you for the fast follow-up and for offering us the opportunity to clarify an important point from our rebuttal.
> > > We thank the reviewer for considering increasing their score and we hope that our response provides enough clarification.
> > >
> > > ## Context of the experiment
> > >
> > > The goal of our work is to provide a weighting for the score matching (SM) loss that more accurately reflects sample error, in particular, the KL divergence. In the additional experiment we provided (https://ibb.co/FQqfg2M, see **Figure 3**), we consider the traditional ELBO weighting of the SM loss, the weighting used by practitioners during training, as well as our proposed weighting and we compare all three against an estimate of the KL divergence.
> > > Note that all notions of loss we consider in this experiment are for the test / population loss.
> > >
> > > The way we perform our comparison is by adding a Gaussian perturbation to the score function that only acts at certain noise levels and we plot how much that notion of loss changes (y-axis) against the noise level where the perturbation was applied (x-axis).
> > >
> > > ## Response to your question
> > >
> > > We see from the plot that the KL is very sensitive to the perturbations that occur at small noise levels and very insensitive at large noise levels.
> > >
> > > When analyzing the ELBO-weighted SM loss (which we recall, is used as an estimate of the KL), we see that the loss changes an amount that is uniform across noise levels, and fails to capture this relationship between sensitivity of error and noise level. The train weighted loss is first increasing then decreasing, failing to capture how perturbations at small noise levels are highly influential, and failing as an upper bound on the KL divergence.
> > >
> > > **Our weighting is the only one that faithfully produces an upper bound on the KL divergence whilst also capturing this negative relationship between loss sensitivity and noise level.** This is what we mean by "accurately reproduc[ing] the shape of the KL divergence": our weighting correctly predicts the negative trend between KL sensitivity and noise scale.
> > >
> > > We are happy to answer any other question you may have.

---

### Decision · Program_Chairs · 2026-04-30

**Decision:**

Accept (regular)

**Comment:**

Error bounds for diffusion models typically go through the ELBO, bounding by KL between path measures which may be loose. The paper investigates this gap and develops tighter bounds in KL, reverse KL, and Wasserstein, which indicate the relative importance of lower noise scales. Reviewers found the paper insightful and bringing in techniques from stochastic analysis; the paper appears to be the first to focus on this particular problem, though the ultimate practical applicability was less clear.